# Learning consistent subcellular landmarks to quantify changes in multiplexed protein maps

Hannah Spitzer ®[1,7], Scott Berry ®[2,3,7], Mark Donoghoe ®[4], Lucas Pelkmans[2]✉ & Fabian J. Theis ®[1,5,6]✉

Highly multiplexed imaging holds enormous promise for understanding how spatial context shapes the activity of the genome and its products at multiple length scales. Here, we introduce a deep learning framework called CAMPA (Conditional Autoencoder for Multiplexed Pixel Analysis), which uses a conditional variational autoencoder to learn representations of molecular pixel profiles that are consistent across heterogeneous cell populations and experimental perturbations. Clustering these pixel-level representations identifies consistent subcellular landmarks, which can be quantitatively compared in terms of their size, shape, molecular composition and relative spatial organization. Using high-resolution multiplexed immunofluorescence, this reveals how subcellular organization changes upon perturbation of RNA synthesis, RNA processing or cell size, and uncovers links between the molecular composition of membraneless organelles and cell-to-cell variability in bulk RNA synthesis rates. By capturing interpretable cellular phenotypes, we anticipate that CAMPA will greatly accelerate the systematic mapping of multiscale atlases of biological organization to identify the rules by which context shapes physiology and disease.

The wide availability of single-cell omics techniques has rapidly advanced our understanding of cell biology in health and disease[1,2]. Currently, there is a rapidly growing range of spatially resolved omics methods, which can quantify tens to hundreds of molecular species in single cells across large populations of cells or tissues, and at the same time show how these molecular species are spatially organized from the multicellular to the subcellular scale[3–5]. This combination of quantitative and spatial information across multiple scales holds enormous promise for understanding biological systems.

Cells in different states (for example, distinct cell cycle positions or disease states) or experimental conditions show changes in the relative abundance and subcellular localization of proteins and RNAs. From an analysis perspective, the challenge is to identify and quantify these changes directly from multiplexed image-based datasets in an unbiased manner, and thereby facilitate their biological interpretation. Previously, pixel clustering of multiplexed image data has been used to identify subcellular regions via similarity of their molecular profiles[3,4]. These approaches weigh all channels equally in clustering, therefore,

[1]Institute of Computational Biology, Helmholtz Center Munich, Munich, Germany. [2]Department of Molecular Life Sciences, University of Zurich, Zurich, Switzerland. [3]EMBL Australia Node in Single Molecule Science, School of Biomedical Sciences, University of New South Wales, Sydney, New South Wales, Australia. [4]Stats Central, Mark Wainwright Analytical Centre, University of New South Wales, Sydney, New South Wales, Australia. [5]School of Computation, Information and Technology CIT, Technical University of Munich, Munich, Germany. [6]TUM School of Life Sciences Weihenstephan, Technical University of Munich, Munich, Germany. [7]These authors contributed equally: Hannah Spitzer, Scott Berry. ✉e-mail: lucas.pelkmans@imls.uzh.ch; fabian.theis@helmholtz-munich.de

when applied across cells from different experimental conditions they typically result in pixels from different conditions being identified as distinct[4], even though they may represent the same subcellular region. As an extreme example, if an experimental treatment eliminates a single target protein (Fig. 1a), the reduction in intensity of the corresponding channel may be the largest difference between the high-dimensional pixel profiles of the two conditions. In this case, direct pixel clustering would identify independent sets of pixel clusters for each condition (Fig. 1b). Although this may be useful for qualitative identification of differences between conditions[4], it does not enable quantification of changes in the internal organization of cells because it is difficult to compare the different sets of subcellular regions found in each condition (Supplementary Note 1).

Recently, deep learning-based segmentation models were used to segment cells and nuclei from multi-channel fluorescence microscopy images[6,7]. However, adapting these supervised methods to generate consistent segmentations of subcellular structures would require annotated training data from all conditions. Although self-supervised approaches alleviate the need for this time-consuming manual labeling[8,9], they do not account for changing localizations of molecular species across perturbations nor do they enable quantification of these changes. To facilitate high-throughput quantitative analysis of subcellular organization, we therefore need approaches that can identify consistent subcellular landmarks despite condition-dependent, and possibly unanticipated, changes to abundance and/or relative localization of measured proteins and RNAs.

To achieve this, we have developed CAMPA (Conditional Autoencoder for Multiplexed Pixel Analysis), a deep learning framework based on conditional variational autoencoders (cVAEs)[10]. CAMPA uses a cVAE for unsupervised learning of condition-independent molecular profile representations to identify consistent subcellular landmarks (CSLs), that is, pixel clusters that are conserved across conditions. Using these landmarks to measure changes in molecular composition and spatial organization at the subcellular scale, CAMPA enables an interpretable comparison of conditions (Fig. 1c). CAMPA is an open-source python package with strong links to the single-cell transcriptomics analysis software, scanpy[11], and its spatial extension, squidpy[12]. It enables high-throughput analysis of high-resolution multiplexed imaging datasets with GPU (graphics processing unit)-accelerated assignment of pixels to CSLs.

Here, we use CAMPA to derive a detailed map of subnuclear organization across different perturbations, directly from high-resolution iterative indirect immunofluorescence imaging (4i) (ref. 4) data. This shows how key proteins and protein states (for example, phosphoproteins and histone post-translational modifications) involved in transcription, chromatin, mRNA processing and nuclear export, as well as subnuclear organelles, change at the cellular and subcellular scale upon perturbation of various stages of messenger RNA metabolism. We find that the three aspects of cellular phenotypic information captured

by CAMPA (cellular intensities, subcellular protein localizations and subcellular spatial organization) contribute unique information to characterize perturbations, indicating that CAMPA will be a powerful approach for cellular phenotypic screening. Finally, by capturing and quantifying interpretable cellular phenotypes at multiple scales, we demonstrate that the combination of 4i and CAMPA can uncover quantitative relationships across scales, from cell populations to subcellular organelles.

## Results

### CAMPA identifies consistent subcellular landmarks

In highly multiplexed image datasets, each pixel is represented as a multiplexed pixel profile: a one-dimensional vector containing the intensity of each marker at that spatial location. We developed CAMPA to identify consistent types of pixel profiles across different experimental conditions, even when some of the underlying channels change. CAMPA first learns a local, condition-independent representation of multiplexed pixel profiles and subsequently clusters the learned representations into CSLs (Fig. 1c). To learn a latent representation $z$, a cVAE is trained on an $n \times n$ neighborhood of the multiplexed pixel profiles $x$, together with a set of condition labels $c$ for each pixel profile. Pixels are then grouped together by applying the Leiden algorithm[13] on a k-nearest neighbor graph of the learned latent (pixel) representations. Because the cVAE model learns a conditional generative distribution $p_\theta(x|z, c)$ for the pixel profiles, the model is optimized to encode variation such as subcellular differences in intensity that occur across all conditions (and omit condition-specific information) in the latent representation $z$[10,14], which results in less condition-dependent clustering of $z$ (Fig. 1h,i). Within CAMPA, identified CSLs can be quantitatively compared in terms of their size, shape, molecular composition and relative spatial organization.

A key goal of perturbation experiments is to identify and quantify induced changes in cellular phenotypes. Here, we focus on how perturbation of various stages of RNA metabolism affects subcellular organization, by collecting a high-resolution (pixel size, 108 nm × 108 nm) 44-plex image dataset of 11,848 human epithelial cells (184A1) across six chemical perturbations, using 4i (ref. 4) (Fig. 1d). The perturbations target different pathways involved in RNA production and processing (histone deacetylation, trichostatin A (TSA); polymerase (Pol) I transcription, CX5461 (ref. 15); Pol II transcription initiation, triptolide[16]; Pol II transcription activation, AZD4573 (ref. 17); and mRNA splicing, meayamycin[18]). The proteins and post-translational modifications imaged (Supplementary Table 1) either play roles in RNA metabolism or are molecular markers of subcellular organelles (for example, nuclear speckles) or cellular states (for example, cell cycle stage, cell crowding). We observed changes in overall protein state abundances across all perturbations (Fig. 1e), confirming previous observations in other cell lines[19]. However, we also noticed perturbation-induced changes in the composition and relative spatial organization of membraneless

---

**Fig. 1 | CAMPA enables unsupervised learning of CSLs using a cVAE.**
**a**, Schematic showing perturbation-induced changes in channel intensity. **b**, Schematic of direct pixel clustering across experimental conditions leading to condition-dependent clusters. **c**, Schematic of CAMPA, showing how a cVAE conditioned on perturbation can learn a perturbation-independent latent space. Clustering this latent space identifies CSLs, enabling quantitative comparisons. **d**, Schematic of the 4i experiment and dataset dimensions. **e**, Fold-change in nuclear mean intensity in different perturbations compared with unperturbed cells, for all proteins with nuclear localization. $P$ values show the significance of the perturbation effect on mean intensity, as determined using a mixed-effect model (Wald test, multiple testing correction using Benjamini–Yekutieli method). 5-EU represents 5-ethynyl uridine pulse labeling of nascent RNA (Methods). **f**, UMAP representation of pixels using either multiplexed pixel profiles (left) or cVAE latent space (right). Pixels from unperturbed cells, trichostatin A (TSA)-treated and triptolide-treated cells colored by perturbation. Data shown are the

subset of pixel profiles used to derive the clustering (see Methods). **g**, Comparison of perturbation dependence of multiplexed pixel profiles, and VAE/cVAE latent space coordinates. Plots show balanced accuracy scores of binary logistic regression classifiers predicting perturbation from normalized multiplexed pixel profiles or latent representations. Accuracy of 0.5 indicates random chance (perturbation information absent from data). **h**, Example cells from each perturbation colored by clusters, along with a pie chart of relative abundance of clusters per perturbation. Left: Direct pixel intensity clustering (Leiden resolution, 1.2). Right: cVAE latent space clustering (CSLs) (Leiden resolution, 0.5). **i**, Comparison of perturbation dependence of direct clustering at different Leiden resolutions, and VAE and cVAE latent space clustering (CSLs). Plots show the coefficient of variation of the fraction of pixels assigned to each cluster in each perturbation. The boxplot summarizes results for all clusters with the number of clusters $n$ shown above. Center line, median; box limits, upper and lower quartiles; whiskers, 1.5-fold the interquartile range; points, all data points.

nuclear organelles involved in RNA metabolism, such as nuclear speckles, promyelocytic leukemia (PML) bodies and the nucleolus. This dataset therefore provides an ideal use-case for the CAMPA framework to generate novel insights into relationships between RNA metabolism and subcellular organization.

To quantify these changes, we initially focused on analyzing the approximately 100 million nuclear pixels for the 34 markers that localized to the nucleus (Extended Data Fig. 1 and Supplementary Tables 2, 3). We applied CAMPA cVAE training and clustering to these data using

cell cycle stage (labeled independently of CAMPA[19,20]) and perturbation condition as categorical condition labels. As expected, we found that multiplexed pixel profiles were highly perturbation dependent when plotted using UMAP (uniform manifold approximation and projection) embedding[21], while the cVAE latent representations appeared to have overlapping distributions (Fig. 1f). To verify the condition independence of the latent representation, we used binary linear classifiers trained to distinguish pixels from perturbed and unperturbed cells based on their latent representations. These classifiers were often

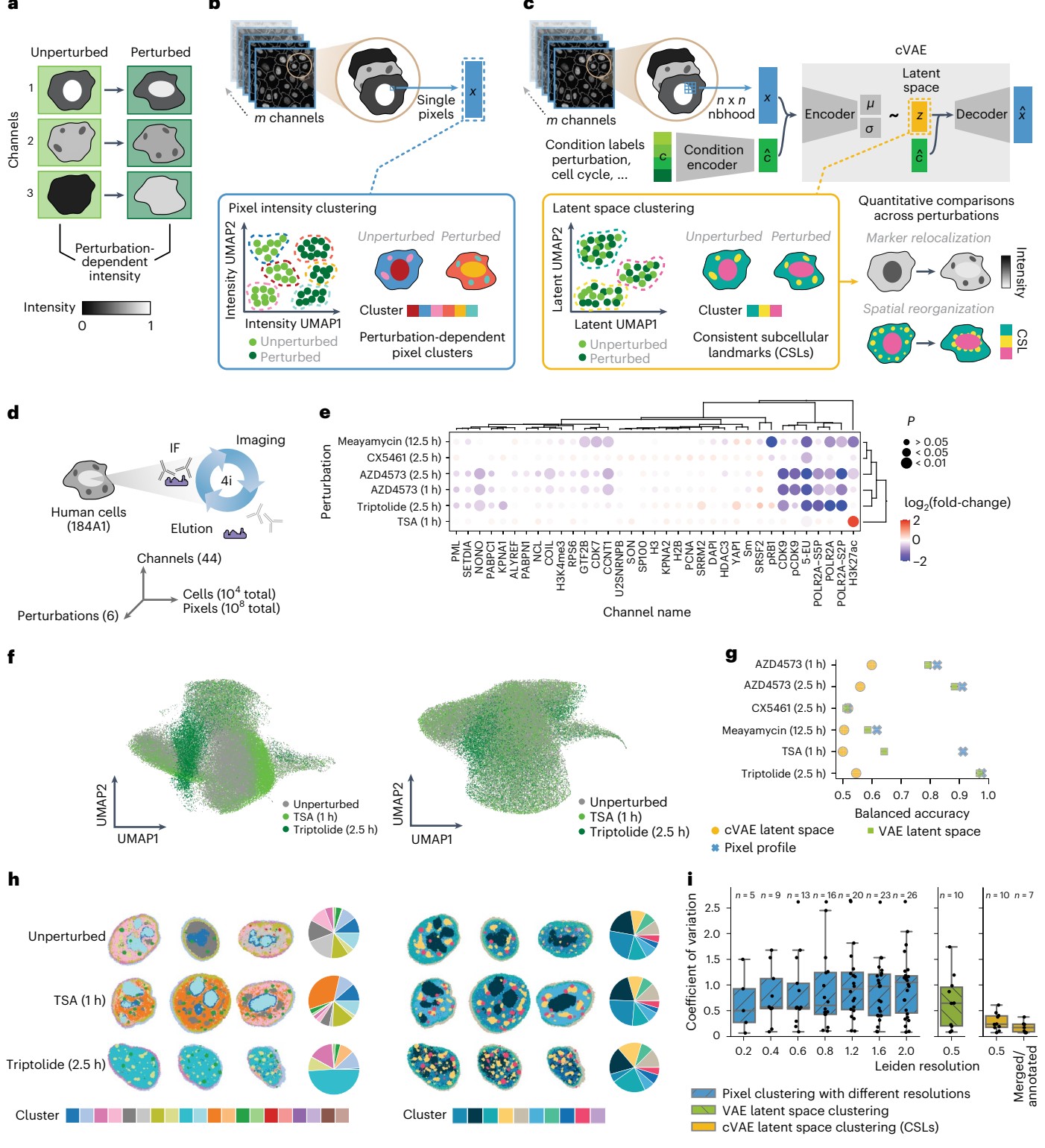

not better than random chance (median accuracy, 0.53; minimum, 0.50; maximum, 0.60). In contrast, classifiers based on multiplexed pixel profiles reached a median accuracy of 0.87 (minimum, 0.52; maximum, 0.98). The VAE model without conditioning was not able to generate condition-independent latent spaces (median accuracy, 0.72; minimum, 0.51; maximum, 0.97), indicating that explicit use of conditioning is necessary in CAMPA (Fig. 1g and Extended Data Fig. 2a). To investigate the importance that the cVAE places on the condition, we used integrated gradients[22], which showed that for channels with perturbation-specific intensity changes, the cVAE places increased importance on the condition input (as opposed to the latent representation) for modeling these channels (Extended Data Fig. 4d,e). We also optimized the input neighborhood size to improve cVAE latent space robustness to single-pixel noise, which often occurs in microscopy imaging. For our data, a $3 \times 3$ neighborhood was optimal (Supplementary Fig. 1d).

To accelerate latent space clustering and to enable interactive clustering on a standard workstation, we clustered a subsample of pixels (150,000 pixels) and then projected resulting clusters to all pixels using the 15 nearest neighbors. This resulted in 10 clusters. Cluster stability was not significantly influenced by a different random subsample nor by increasing or decreasing the number of samples used for the clustering by a factor of two (Supplementary Fig. 1a,b). Because all conditions are considered together, any cluster instability does not affect the ability to quantitatively compare cells across conditions. For comparison with previous approaches, we also directly clustered pixels using their multiplexed pixel profiles[4]. Whereas intensity space clusters were enriched in different perturbations (Fig. 1h and Extended Data Fig. 2b), latent space clusters were evenly distributed across perturbations (Fig. 1h and Extended Data Fig. 2c). To quantify the perturbation specificity of clusters, we computed the median coefficient of variation of the fraction of pixels assigned to each cluster across perturbations. The median coefficient of variation of the latent space clustering is 0.24 (minimum, 0.08; maximum, 0.61), indicating that clusters have a similar relative abundance in different perturbations, whereas direct pixel clustering at similar resolution results in a median coefficient of variation of 0.57 (minimum, 0.09; maximum, 2.62) (Fig. 1i). In addition, despite differences in intensities of some 4i markers across different cell cycle phases (for example, PCNA (proliferating cell nuclear antigen), pRB1), the inclusion of cell cycle as a condition in CAMPA reduced the cell cycle dependence of the latent representations (median accuracy of pairwise binary classifiers of latent space/pixel profiles, 0.58/0.67), which resulted in latent space clusters being assigned consistently across cell cycle stages (median coefficient of variation across cell cycle stages of latent space clustering/direct pixel clustering, 0.11/0.21) (Extended Data Fig. 3). We therefore name these cVAE latent space clusters 'consistent subcellular landmarks' (CSLs) and use them in the following to analyze the impact of perturbations on subcellular organization.

To enable biological interpretability of quantitative comparisons between cells, we annotated CSLs with the names of known subcellular structures (see Methods) (Fig. 2a). To facilitate this optional step in the CAMPA workflow and to avoid mis-annotations, automated annotation proposals can be obtained by querying the Human Protein Atlas (https://www.proteinatlas.org/)[23] database. The annotation resulted in assignment of the 10 original CSLs to seven annotated CSLs (Nucleolus, Nuclear speckles, PML bodies, Cajal bodies, Nucleoplasm, Nuclear periphery and Extra-nuclear (outside the nucleus)) (Fig. 2d–i), by merging four original CSLs into the Nucleoplasm CSL (Extended Data Fig. 4a). These annotations are consistent with automatic annotations proposed by the Human Protein Atlas database (Extended Data Fig. 4b). In the following we refer to these annotated CSLs simply as CSLs. To quantitatively validate CSL annotations, we performed two manual segmentations of nuclear speckles and two manual segmentations of PML bodies using state-of-the-art pixel classifiers[24] (Extended Data Fig. 5). These were

based only on single-channel intensities of canonical markers for these membraneless organelles (SON and SRRM2 for nuclear speckles and SP100 and PML for PML bodies). We quantitatively compared these manual segmentations with their respective CSLs using the F1-score (a measure of similarity) and found that CSL-derived nuclear speckles were as similar to the manual segmentations ($F_{1(CSL|SON)} = 0.963 \pm 0.006$, $F_{1(CSL|SRRM2)} = 0.967 \pm 0.006$, mean ± s.d. between conditions) as the different manual segmentations are to one another ($F_{1(SRRM2|SON)} = 0.964 \pm 0.007$) (Extended Data Fig. 5). $F_1$-scores were similarly high for PML bodies.

We therefore conclude that CAMPA enables consistent identification and annotation of subcellular landmarks across perturbations and cell cycle stages. This contrasts with previous direct pixel clustering approaches, which often identify different clusters for the same subcellular organelle in different conditions or cell cycle stages. Unlike for manual segmentation of subcellular structures, when using CAMPA to identify CSLs there is no need to pre-define markers of certain landmarks in advance, because the cVAE uses all channels that are consistent across perturbations to define the latent space. This may ultimately enable identification of novel landmarks defined by higher-dimensional combinations of different channels. Importantly, the cVAE learns to remove condition-specific information from channels that show characteristic changes in intensity between conditions when generating the latent space and the CSLs. Naturally, as shown in the following, these channels can then be used to compare the effects of, and differences between, perturbations when aggregated on the CSLs.

## Uncovering perturbation-induced subcellular landmark changes

To quantify subcellular changes in abundance of markers across the six perturbations, we calculated the mean intensity of each marker in each CSL per cell. We then computed the fold-change for a particular condition compared with unperturbed cells, across all CSL–channel combinations, as well as the fold-changes in the size (number of pixels) of each CSL (Supplementary Fig. 2a,b). Unlike direct pixel clustering approaches[3,4], in which conditions are compared by identifying pixel classes that change abundance between conditions (Extended Data Fig. 6 and Supplementary Note 1), CAMPA compares molecular abundances across landmarks that are consistently found in both conditions (CSLs). This naturally extends traditional quantification of overall cellular abundance changes (Fig. 1d) to the subcellular scale. Focusing on meayamycin, which perturbs mRNA splicing[18], CAMPA identified a set of markers that were uniformly depleted across the nucleus, and an overall increase in the size of nuclear speckles (Supplementary Fig. 2b). To investigate relocalization of proteins (rather than overall changes in abundance), we normalized intensity fold-changes in each CSL by their corresponding whole-nucleus fold-changes (Fig. 3a and Supplementary Fig. 2c). This showed that the relative size of nuclear speckles increases upon meayamycin treatment, and that their molecular composition changes: they become significantly enriched in cytoplasmic poly(A) binding protein 1 (PABPC1) (Fig. 3d) and depleted in POLR2A-S2P (a marker of actively transcribing RNA polymerase II) (Fig. 3e). PABPC1 relocalization to nuclear speckles was observed previously[25]. POLR2A-S2P is typically distributed throughout the nucleoplasm with slight enrichment in nuclear speckles (Fig. 2c)[26]. However, upon inhibition of mRNA splicing, POLR2A-S2P is reduced in overall abundance (Supplementary Fig. 2b) and is specifically excluded from nuclear speckles (Fig. 3a,e). These changes in POLR2A-S2P were mirrored by a reduction in bulk RNA production upon meayamycin treatment, as measured using 5-ethynyl uridine pulse labeling (Fig. 1d and Methods). Many mRNA splicing factors are located in nuclear speckles, and transcription and splicing has been reported to occur more efficiently in their vicinity[27–29]. Moreover, Ser2-phosphorylation of POLR2A is important for coupling of mRNA splicing and transcriptional

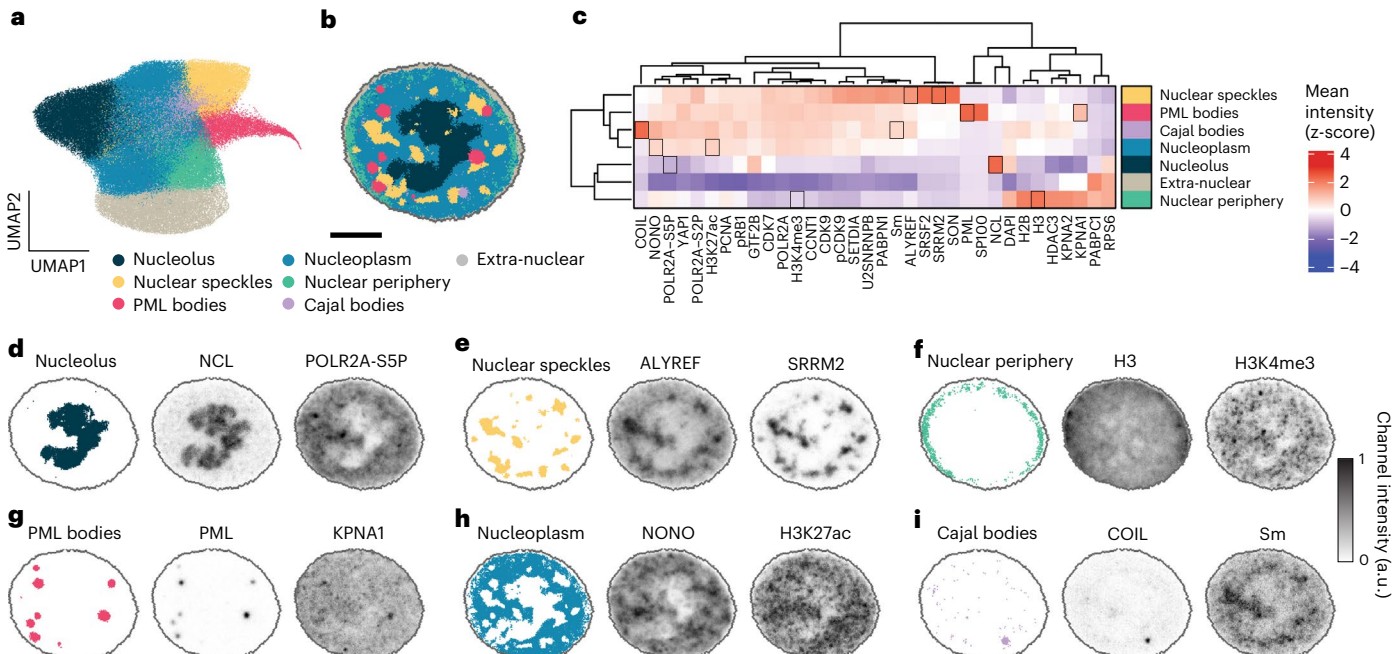

**Fig. 2 | CSLs represent known subnuclear structures. a**, UMAP representation of pixels using their cVAE latent representations generated in CAMPA, colored by CSL. **b**, Example nucleus showing the spatial distribution of CSLs. **c**, Relative mean intensity of each channel in each annotated CSL (see Extended Data Fig. 4a for all 10 Leiden clusters). Heatmap z-scored by column to show the relative localization of each channel across CSLs. The black-outlined boxes are highlighted in **d**–**i**. **d**–**i**, Example 4i channels that are enriched or depleted in the identified CSLs, shown together with CSLs. See **c** for the distribution of channels across the CSLs. Scale bar, 5 μm.

elongation[30]. However, our analysis shows that the relative abundance of CDK9 (the kinase predominantly responsible for POLR2A-S2P) increases in nuclear speckles at the same time (Fig. 3a). This indicates that inhibition of splicing affects overall transcription rates, and either causes relocalization of transcribing Pol II (POLR2A-S2P) further away from nuclear speckles or preferentially affects transcription of genes that are normally transcribed in the vicinity of nuclear speckles. These findings are in agreement with a model in which splicing and transcription are functionally and kinetically coupled[31].

To analyze changes in spatial arrangement of the identified CSLs upon meayamycin treatment, we computed the pairwise spatial co-occurrence between all CSLs (Fig. 3f). Spatial co-occurrence[12,32] captures the relative probability that two CSLs are found within a given distance interval from one another (Fig. 3g and Supplementary Fig. 3). At short distances, co-occurrence scores from a structure to itself (auto-co-occurrence) are typically high, reflecting the fact that pixels in close spatial proximity are likely to be from the same CSL. We found that spatial auto-co-occurrence of nuclear speckles remains high at larger distances in meayamycin-treated cells than in unperturbed cells. This indicates that the average size of nuclear speckles increases in this perturbation, which we confirmed (Fig. 3h). Examining the co-occurrence between CSLs, we found that co-occurrence of PML bodies and nuclear speckles increases at short distances in meayamycin-treated cells compared with unperturbed cells (Fig. 3g), indicating that PML bodies are more likely to be found in close proximity to nuclear speckles. The opposite effect was observed between nuclear speckles and the nucleolus (Fig. 3g). Re-examination of images of CAMPA-derived subcellular segmentations showed that, upon meayamycin treatment, PML bodies indeed appear to coalesce onto nuclear speckles, and the nucleolus and nuclear speckles appear to move further from one another (Fig. 3b). To our knowledge, neither of these observations has been previously reported. PML bodies have been reported to juxtapose with Cajal bodies[33] and some PML isoforms (produced through alternative splicing) localize to the nucleolar periphery[34]. Notably,

all of these compartments, including nuclear speckles, are thought to form through liquid–liquid phase separation[35], therefore relocalization of PML bodies to contact nuclear speckles could represent surface-wetting between these distinct condensates[36].

CSLs can thus be used to identify and statistically quantify both absolute and relative changes in molecular abundance in different cellular structures and to quantify changes in the size, morphological properties and the high-dimensional subcellular spatial organization of thousands of cells.

**Comparing multiple perturbations**

So far, we have considered comparisons of each perturbation to unperturbed controls. Here, we extend these analyses and show how CAMPA can be used to compare multiple perturbations with one another. To do this, we generated a feature vector for each cell containing the mean intensity of each channel in each CSL (Fig. 4b). We used this as a representation of the specific subcellular-localized abundance of each channel. In a similar way, we represented the spatial organization of the nucleus as a feature vector containing the pairwise spatial co-occurrence scores (Fig. 4c). Finally, we used a baseline feature vector of mean nuclear intensities of all channels to represent the information available without subcellular resolution (Fig. 4a). To determine how these distinct aspects of cellular organization change across all perturbations, we quantified differences between perturbations with all three per-cell representations using pairwise silhouette scores (Fig. 4d–f). Using mean nuclear intensity features, perturbations targeting Pol II transcription (AZD4573, triptolide) showed low pairwise silhouette scores, indicating common changes in overall nuclear abundance of the proteins and protein states measured (Fig. 4d). In almost all of the cases, pairwise silhouette scores were higher when considering per-CSL intensities (Fig. 4e) instead of whole-nucleus intensities. This indicates that per-CSL intensities provide a more fine-grained characterization of the cellular phenotype and are therefore better able to distinguish perturbations. In contrast, we found that spatial co-occurrence

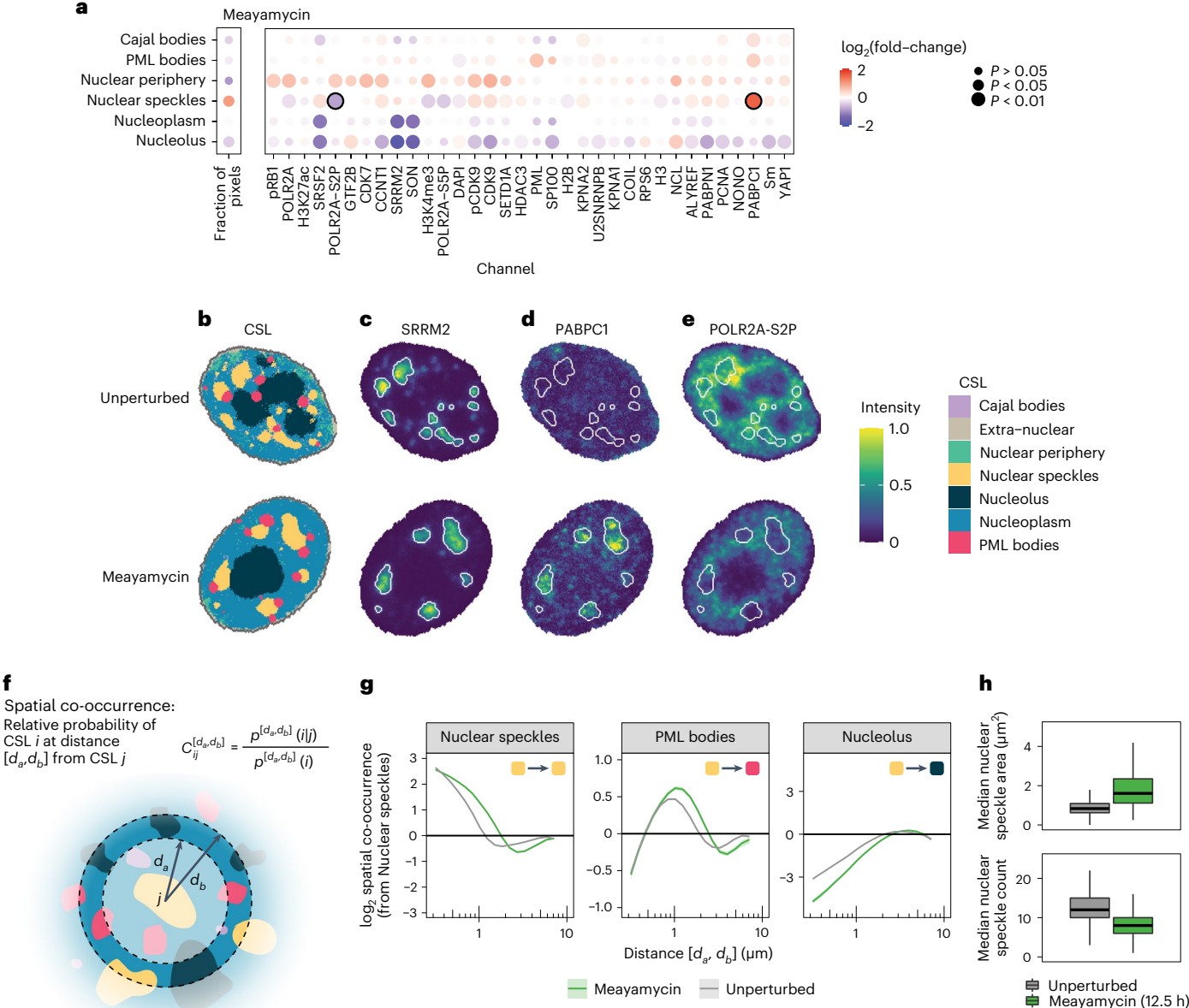

**Fig. 3 | Molecular composition and spatial organization of subcellular landmarks change upon inhibition of mRNA splicing. a,** log₂ fold-change of mean intensities for each channel in each CSL, or number of pixels in each CSL, when comparing meayamycin with unperturbed cells. Values shown are normalized to overall (whole-nucleus) changes in intensity. *P* values show the significance of meayamycin treatment on intensity of each channel and CSL combination, compared with the change observed for the whole nucleus, as determined from the mixed-effect model (Wald test, multiple testing correction using the Benjamini–Yekutieli method). **b,** Example unperturbed (top) and meayamycin-treated (bottom) cells, colored by CSL. **c–e,** Example cell from **b** with pixels colored by SRRM2 intensity (**c**), PABPC1 intensity (**d**) and POLR2A-S2P intensity

(**e**). **f,** Schematic showing calculation of spatial co-occurrence. **g,** Mean log₂ spatial co-occurrence from Nuclear speckles to Nuclear speckles (auto-co-occurrence), Nucleolus and PML bodies, as a function of distance (minimum of the distance interval; on log scale) in meayamycin-treated and unperturbed cells. Shaded regions indicate 95% confidence intervals for the mean. See Supplementary Fig. 3b for all co-occurrence plots of meayamycin-treated and unperturbed cells. **h,** Median area of individual nuclear speckles, and number of nuclear speckles per cell. Boxplots summarize distributions over meayamycin-treated and unperturbed cells. Center line, median; box limits, upper and lower quartiles; whiskers, 1.5-fold the interquartile range; outliers omitted for clarity. Unperturbed, *n* = 3,680; meayamycin, *n* = 755 (see Supplementary Table 3 for details).

scores alone were generally less able to distinguish perturbations than mean nuclear intensities (lower silhouette scores). For example, cells treated with the histone deacetylase inhibitor trichostatin A, were distinct from unperturbed cells when using whole-nucleus intensities but highly similar when using co-occurrence scores (Fig. 4d,f). This indicates a limited change in spatial organization of the nucleus upon histone deacetylase inhibition (for the 4i markers quantified in our experiment), despite hyperacetylation of histones (Fig. 1d). One notable exception was the RNA Pol I inhibitor in CX5461-treated and

unperturbed cells. Here, spatial information was significantly more informative than molecular abundance information when distinguishing perturbations both at the whole-nucleus and CSL levels (Fig. 4d,f). To pinpoint how their spatial organization differs, we compared all CSL spatial co-occurrences between CX5461-treated cells and unperturbed controls. This showed that the major difference was in the relative spatial distribution of the nucleolus CSL, compared with itself and with other CSLs (Fig. 4g and Extended Data Fig. 7). In particular, the nucleolus had higher spatial auto-co-occurrence at short distances

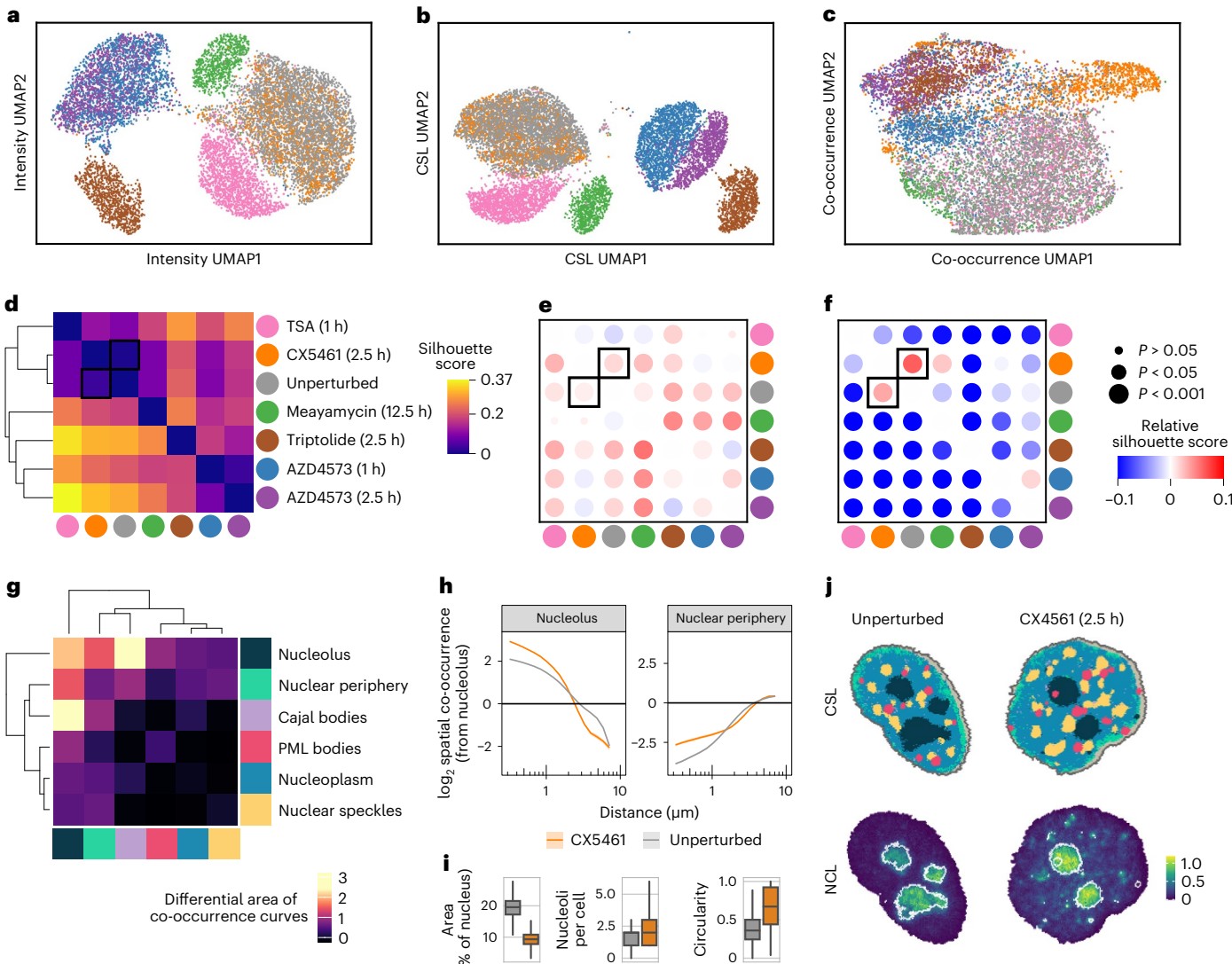

**Fig. 4 | CAMPA-derived cell features enable comparisons of spatial and molecular differences across multiple perturbations. a–c**, UMAP embedding of cells using per-nucleus mean intensity (**a**), per-CSL mean intensity (**b**) and pairwise CSL spatial co-occurrence scores (**c**). Points are colored by perturbation, and UMAP outliers are manually excluded for clarity (Supplementary Fig. 4). **d**, Pairwise differences between perturbations measured by silhouette score using per-cell mean intensity values. Higher silhouette scores indicate less overlap between perturbations. **e**, Change in silhouette score when considering per-CSL intensities. Negative values indicate decreased silhouette scores compared with per-cell intensity silhouette scores; positive values indicate increased silhouette scores. $P$ values obtained using the two-sided Wilcoxon signed-rank test and were adjusted for multiple testing using Bonferroni correction. **f**, As in **e**, for the change in silhouette score when considering pairwise CSL spatial co-occurrence scores. **g**, Comparison of pairwise spatial co-occurrences for different CSLs in

CX5461-treated cells and unperturbed cells quantified as the area between spatial co-occurrences curves (computed using log-transformed distances). **h**, Mean $\log_2$ spatial co-occurrence from Nucleolus to Nucleolus (auto-co-occurrence) or Nuclear periphery, as a function of distance (on log scale) in CX5461-treated and unperturbed cells. Shaded regions indicate 95% confidence intervals for the mean. **i**, Total physical area of nucleolus (as a fraction of the nuclear area), number of nucleoli per cell and median nucleolus circularity per cell. Boxplots summarize distributions across CX5461-treated and unperturbed cells. Center line, median; box limits, upper and lower quartiles; whiskers, 1.5-fold the interquartile range; outliers omitted for clarity. Before obtaining counts and circularity per cell, small objects were removed (Methods). Unperturbed, $n = 3,680$; CX5461, $n = 1,152$ (see Supplementary Table 3 for details). **j**, Top: example CX5461-treated and unperturbed cells with pixels colored by CSL. Bottom: nucleolin (NCL) intensity from the same cells with nucleolus CSL outlines overlaid.

and lower spatial auto-co-occurrence at longer distances, indicating that the nucleolus adopts a more compact and spatially coherent conformation in CX5461-treated cells (Fig. 4h). Moreover, pixels assigned to the nucleolus were more likely to be found close to the nuclear periphery. On examination of example images we found that CX5461 treatment results in a circularization and shrinking (Fig. 4i,j) of the nucleolus and fragmentation into smaller regions enriched in the nucleolar marker NCL. Given that CX5461 inhibits synthesis of ribosomal RNA, changes in the morphology of the nucleolus (the site of rRNA transcription) in CX5461-treated cells are not unexpected. Nonetheless, it shows that CAMPA can rapidly identify that the nucleolus

is the primary site of activity of this compound, despite the antibody panel not having a marker for the directly targeted protein (RNA polymerase I). This points to the exciting future possibility of applying CAMPA in a chemical compound screening format to provide clues to subcellular locations that are relevant for the activity of a particular molecule. Overall, this analysis shows that cellular representations obtained through CAMPA can be used to compare cells from several perturbations at once, at the level of subcellular localization or spatial organization. These are rich and readily interpretable sources of information, which are complementary to one another and to overall protein abundance measurements.

## Revealing subcellular reorganization upon cell size change

Having developed CAMPA on a 34-plex dataset focused on cell nuclei, we next applied it to whole cell images to demonstrate its potential to identify a larger number of cellular landmarks from higher-dimensional image data. Here, we examined HeLa cells in which expression of *SBF2* (SET binding factor 2) is reduced by treatment with short interfering RNA, which results in an approximate twofold increase of cell volume and an approximately threefold increase in cell area[19] compared with control cells transfected with scrambled siRNA (Supplementary Tables 4 and 5). We applied CAMPA on 43 channels comprising both nuclear and cytoplasmic 4i stains, using perturbation (SBF2 or scrambled siRNA) and cell cycle stage (G1, S, G2) as conditions (Supplementary Table 6). This resulted in 21 CSLs (Extended Data Fig. 8a), some of which were manually merged, including two distinct P-body CSLs that correspond to the center and periphery, respectively (Extended Data Fig. 8b). This resulted in 16 distinct cytoplasmic and nuclear annotated CSLs (Fig. 5). These comprise all major compartments marked by the antibodies in the panel, including all previously identified nuclear CSLs (for comparison see Extended Data Fig. 8d,e) as well as cytoplasmic landmarks such as perinuclear and peripheral endoplasmic reticulum and mitochondria (HSPD1/CALR), Golgi apparatus (GOLGA2) cell–cell contacts (CTNNB1), focal adhesions (PXN) and P-bodies (DDX6) (Fig. 5b,c). Our manual annotation is consistent with the automated annotation, but is more detailed (Extended Data Fig. 8c). Comparison of the per-CSL mean intensities of each marker between conditions showed several uniform differences across the whole nucleus or cytoplasm (Supplementary Fig. 5a,c). The more striking changes were differences in the relative size of CSLs (Fig. 5d). This indicates that the doubling of cell volume induced by *SBF2* knockdown is associated with disproportionate changes in size of different subcellular compartments, however, we cannot exclude other effects of *SBF2* knockdown that are independent of cell size changes. Focusing on membraneless organelles, we found that the markers of the nucleolus and Cajal bodies (NCL and COIL, respectively) both increased their molecular abundance in larger *SBF2* knockdown cells (Supplementary Fig. 5e,f). However, the size of the nucleolus in *SBF2* knockdown cells was similar to that of controls (Fig. 5f). Because nuclear area also increases with cell volume upon *SBF2* knockdown[19], the size of the nucleolus as a fraction of the nucleus decreases. In contrast, Cajal bodies increased their combined size by approximately fivefold, a larger increase than the increase in nuclear or cell area (Fig. 5e). This was predominantly achieved by increasing the size of the individual Cajal bodies rather than by increasing their number per cell (Fig. 5e). In contrast, we found that P-bodies, a cytoplasmic membraneless organelle involved in RNA processing[37], increased in number per cell rather than by increasing the size of individual P-bodies (Fig. 5g). When we binned cells by cell size (total protein content), we found that the number of P-bodies in each cell is closely related to cell size, independent of the genetic perturbation (Fig. 5h).

This analysis shows that CAMPA generalizes to a higher level of multiplexing and can identify CSLs not only across conditions with different molecular profiles but also across different CSL sizes. Morphological properties of CSLs on a per-cell basis such as count and area can be used to compare and interpret changes in scaling behavior between conditions.

## Linking cellular heterogeneity to subcellular reorganization

Finally, we use CAMPA to study how subcellular properties vary within cell populations, to examine its potential in uncovering links between subcellular properties and cellular states. Rates of RNA production are heterogeneous in cell populations[19,38] and can be measured by RNA metabolic labeling with 5-ethynyl uridine[39]. Nuclear 5-ethynyl uridine intensity quantifies the amount of nascent RNA synthesized during a 5-ethynyl uridine pulse at the single-cell level (Fig. 6a). To examine how differences in bulk RNA production are related to subcellular changes, we considered control cells (scrambled siRNA) from the CAMPA model trained on entire HeLa cells (Fig. 5) and binned these into either 'low' (lower quartile) or 'high' (upper quartile) RNA synthesis, using mean nuclear 5-ethynyl uridine intensity (Extended Data Fig. 9a). Examination of intensity fold-changes for each channel–CSL combination between these groups revealed changes in overall nuclear concentration of POLR2A and other proteins and protein states related to RNA synthesis (Extended Data Fig. 9b), as previously observed[19]. Focusing on the subcellular level, we observed that PML bodies showed a change in the relative molecular composition of PML and SP100, the two markers of PML bodies used in this experiment. In cells with low RNA synthesis, PML bodies were enriched in PML, while in cells with high RNA synthesis, PML bodies were enriched for SP100. These changes are difficult to observe in overall (all) or whole-nucleus (Nucleus (combined)) CSLs, demonstrating the importance of quantifying this at the subcellular scale. These trends were recapitulated across the full range of 5-ethynyl uridine intensities (Fig. 6b), and were observed in the G1, S and G2 phases of the cell cycle (Extended Data Fig. 9c). PML bodies have previously been implicated in transcriptional regulation[40], however, their molecular composition has not been linked to global changes in transcriptional output of single cells. Examining images directly revealed heterogeneity in PML body composition, both between and in cells (Fig. 6c). Specifically, cells with low RNA synthesis had PML bodies lacking SP100, while high RNA synthesis cells had PML bodies lacking PML. Classically, these bodies are defined as having both SP100 and PML[41]. Detection of these nuclear bodies based only on PML or on SP100 (univariate) would have not assigned all these pixels as PML bodies, highlighting a key difference between CAMPA and univariate approaches. It is important to note, however, that, given that we did not use 5-ethynyl uridine intensity as a condition in the cVAE training, we would expect to see these unique pixel combinations annotated as different CSLs at higher clustering resolution (Extended Data Fig. 9d).

**Fig. 5 | Subcellular landmarks reveal coordination of organelle and cell size.** **a**, CSLs identified using CAMPA from 43-plex 4i data of HeLa cells transfected with scrambled siRNA (top) or *SBF2* siRNA (bottom). **b**, Relative mean intensity of each channel in each CSL, omitting the Antibody Aggregate CSL (see Extended Data Fig. 8 for all 21 cVAE latent space Leiden clusters). Heatmap z-scored by column to show the relative localization of each channel across CSLs. ER, endoplasmic reticulum. **c**, Example 4i images in the example *SBF2* knockdown cell for comparison with identified CSLs. **d**, $\log_2$ fold-changes of number of pixels per cell assigned to each CSL when comparing *SBF2* knockdown with control cells (scrambled siRNA). *P* values show the significance of the effect of *SBF2* knockdown on the abundance of each CSL, as determined from the mixed-effect model. *P* values are corrected for multiple hypothesis testing using the Benjamini–Yekutieli method. Left panels show non-normalized changes in CSL sizes, right panels show changes normalized to the nuclear (upper) or cytoplasmic (lower) size changes, respectively. **e**, Upper: number of Cajal bodies per cell and their per-cell median areas. Before obtaining counts and areas per cell, small objects were removed (Methods). Lower: Cajal body area as a percentage of nuclear area or as un-normalized. Boxplots summarize distributions across cells (center line, median; box limits, upper and lower quartiles; whiskers, 1.5-fold the interquartile range; outliers omitted for clarity). Scrambled, *n* = 2,301; SBF2, *n* = 430 (see Supplementary Table 5 for details). **f**, As in **e** for NCL and Nucleolus. **g**, As in **e** for DDX6 and P-bodies. **h**, Cells binned by cell size (total protein content). The upper panel shows the fraction of cells in each bin per condition. The middle panel shows the mean number of P-bodies per cell for each bin. The lower panel shows the average size of individual P-bodies (mean of median P-body area per cell). Bins with less than 10 cells per genotype were omitted. Error bars show 95% confidence intervals for mean (obtained using bootstrapping; n = 500). Fit lines show LOESS (locally estimated scatterplot smoothing) regression of binned data with the shaded region representing the 95% confidence interval. Before obtaining counts per cell, small objects were removed (Methods). Scale bars: **a**, 20 μm; **e–g**, 20 μm.

These results demonstrate that CAMPA can be used not only to reveal changes between perturbations but also to uncover links between global properties of cells and their subcellular organization.

## Discussion

Quantifying changes in subcellular organization across perturbations in an automated manner is a central goal in highly multiplexed imaging. This has so far been difficult because perturbation-induced changes or heterogeneity in cell populations has prevented the consistent annotation of subcellular structures. In CAMPA, we use a cVAE to learn robust

perturbation- and cell state-independent latent representations of pixels that enables the identification of CSLs, found across perturbations and cell states. This differs from previous approaches based on direct clustering of multiplexed pixel profiles, which aim to identify pixel combinations that are unique or enriched in different experimental conditions or cell states. In contrast, CAMPA quantifies changes in all markers with respect to consistently identified landmarks. This leads to a more interpretable and quantitative assessment of changes between conditions that directly provides insights into changes in subcellular protein abundance and localization, and the relative positioning of

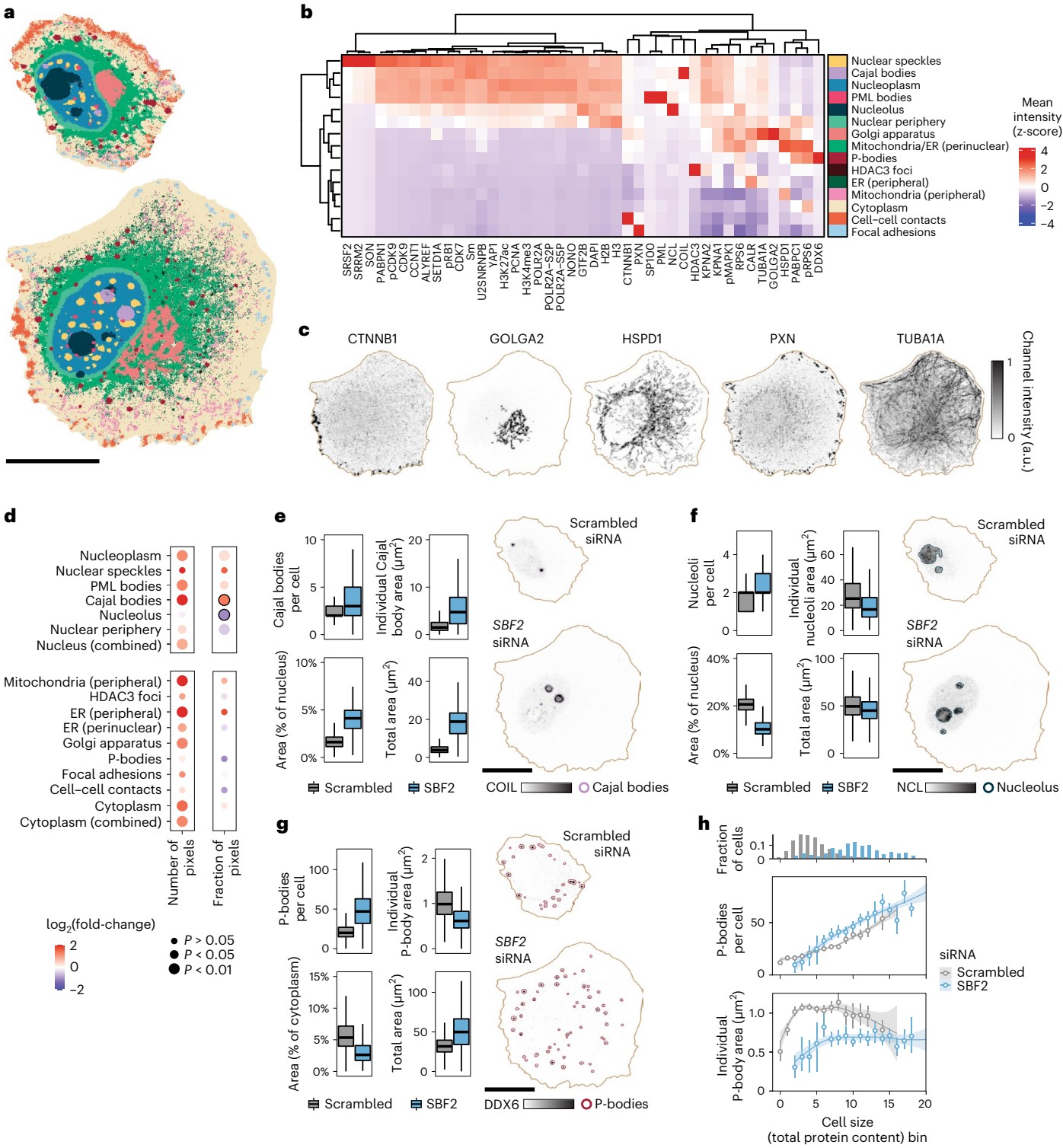

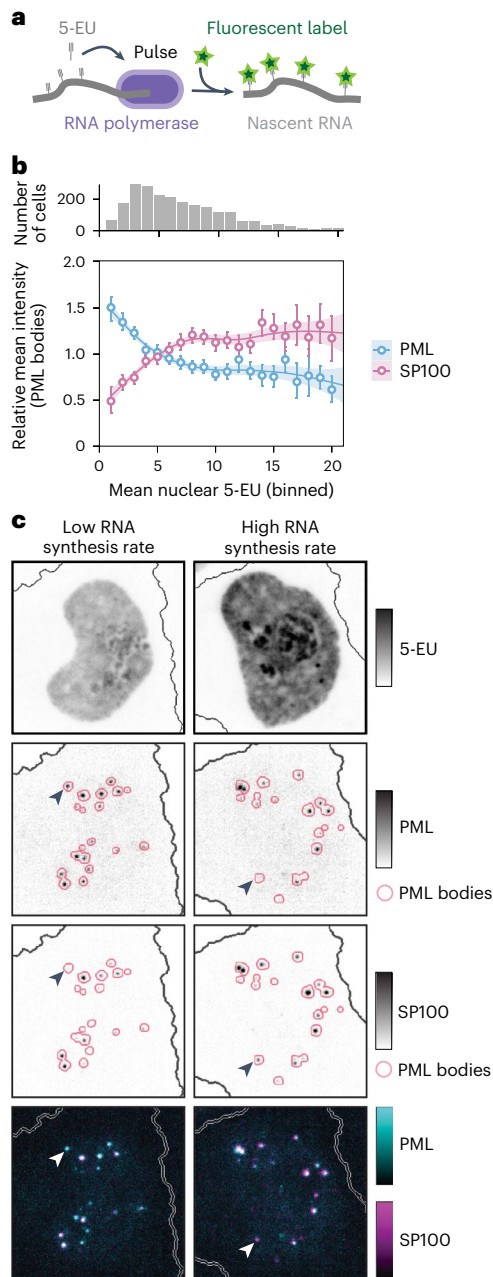

**Fig. 6 | Cellular RNA synthesis rates are associated with altered molecular composition of PML bodies. a**, Schematic of RNA metabolic pulse labeling with 5-EU[39]. **b**, Mean intensity of PML and SP100 in PML bodies as a function of mean nuclear 5-EU intensity (RNA synthesis rate) relative to the mean across all cells. All cells binned by 5-EU. Bins with less than 10 cells were omitted. Error bars show 95% confidence intervals for mean (obtained using bootstrapping; $n = 500$). Fit lines show LOESS regression of binned data with the shaded region representing the 95% confidence interval. The upper panel shows the number of cells in each bin (total $n = 2,301$ (scrambled siRNA, see Supplementary Table 5)). **c**, Example images comparing two S-phase cells in states of high and low RNA synthesis. Arrows highlight a PML-only PML body (low RNA synthesis, left) and an SP100-only PML body (high RNA synthesis, right). The outlines of PML bodies derived from PML body CSL were dilated by 9 pixels for visualization purposes. Scale bar, 10 μm.

organizational units in the cell at subcellular length scales. Compared with direct pixel clustering, CAMPA also scales more readily to compare large numbers of perturbations, because the number of CSLs that needs to be considered does not necessarily increase with the number of different perturbation conditions studied.

Cellular representations based on CAMPA-derived features can be used to compare multiple perturbations with one another simultaneously. We found that different sources of information (spatial versus intensity based) were complementary at distinguishing perturbations. Unlike other deep learning-based approaches for generating cellular representations, CAMPA-derived cellular representations are highly interpretable. For example, the observation that CX5461-treated cells are distinguishable from unperturbed cells using spatial representations leads readily to the identification of a change in nucleolar morphology in this perturbation. Because both 4i and CAMPA can be applied in high throughput, this approach has enormous potential for screening applications. We envisage that CAMPA-derived cellular representations could be used as interpretable fingerprints to characterize and compare perturbations in terms of their subcellular phenotypes.

Here, we focused on subcellular imaging of proteins using 4i, however, we anticipate that CAMPA could readily be applied to other modalities such as multiplexed RNA fluorescence in situ hybridization[42] or integrated spatial genomics[3] (RNA, proteins and DNA in the same cells), that is, technologies that have not yet been used to study perturbations at the subcellular scale. Currently, one limitation of CAMPA (and all previous pixel clustering approaches) is that pixels are assigned only to one cluster type. Pixel types therefore compete for allocation, with markers that show characteristic, sparse distributions in cells preferentially being used to define cellular landmarks. Limited optical resolution means that proteins that do not occupy the same physical space in the cell are nonetheless visualized in the same pixels. In our data, the number of structures visualized was appropriate for the optical resolution used, as evidenced by the limited overlap between defining channels of CSLs, however, as we further increase the number of structures simultaneously visualized, this problem will become more pronounced. In CAMPA, this may be addressed in the future by using mixture models[43] or approaches from fuzzy clustering[44] on the latent space, to enable pixels to be simultaneously assigned to multiple different CSLs.

CAMPA uses a cVAE to generate consistent latent representations of multiplexed pixel profiles across multiple conditions, which is computationally similar to approaches for integrating and clustering single-cell transcriptomics data[14,45]. Extensions and enhancements to the cVAE framework developed in this related field could easily be leveraged by CAMPA in the future. One example of this would be an adversarial loss to enforce strict disentangling of more complex condition effects and latent representation or 'architecture surgery'[46] to enable integration of new data to already learned representations. In this way, CAMPA could contribute to building a queryable atlas of intracellular variation, onto which novel observations from different experimentalists could be projected to not only annotate CSLs, but also to compare with reference atlases. Altogether this will render CAMPA applicable to an even wider range of data and conditions and thus contribute to uncovering the rules by which spatial context shapes the activity of our genome across multiple scales.

## Online content

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

## Methods

### Cell lines and culture conditions

HeLa Kyoto (female) cell populations were derived from a single-cell clone and were tested for identity by karyotyping[47]. HeLa cells were cultured in high glucose DMEM supplemented with 10% FBS and 1% GlutaMAX. Cells with low passage number (2–6) were used for all experiments.

184A1 (human female breast epithelial) cell populations were derived from a single-cell clone, and were used at low passage number (2–6) for all experiments. 184A1 cells were cultured in DMEM/F12 media supplemented with 5% horse serum, 20 ng ml$^{-1}$ epidermal growth factor, 10 µg ml$^{-1}$ insulin, 0.5 µg ml$^{-1}$ hydrocortisone and 10 ng ml$^{-1}$ cholera toxin.

For all experiments, cells were grown and imaged in uncoated Greiner µClear plastic-bottom 394-well plates.

### Chemical treatments

A total of 1,250 184A1 cells were plated 72 h before chemical treatment. RNA polymerase I inhibitor, CX5461 (ref. 15) was dissolved in 5 mN HCl at a concentration of 5 mM and used at 2 µM. XPB (TFIIH) inhibitor, triptolide[16] was dissolved in dimethylsulfoxide (DMSO) at a concentration of 10 mM and used at 2 µM. CDK9 inhibitor AZD4573 (ref. 17) was dissolved in DMSO at a concentration of 10 mM and used at 0.1 µM. Splicing factor 3b subunit 1 (SF3b1) inhibitor, meayamycin[18], was dissolved in DMSO at a concentration of 10 µM and used at 10 nM. When applicable, the final DMSO concentration was 0.1%. Duration of chemical treatment is noted throughout the text and figures.

### siRNA transfection

Transfection with siRNA was performed as previously described[25]. In brief, 700 HeLa cells were plated per well in 384-well plates for reverse transfection onto a mixture of pooled siRNAs (5 nM final concentration) and lipofectamine RNAiMAX (0.08 µl per well in OptiMEM) according to the manufacturer's specifications. Cells were subsequently grown for 72 h at 37 °C in a final volume of 50 µl growth media, to establish efficient knockdown of the targeted genes[19]. *SBF2* knockdown was validated previously[19].

### Image acquisition

Imaging was performed on an automated spinning-disk microscope (CellVoyager 7000, Yokogawa, software vR1.17.05), equipped with four excitation lasers (405, 488, 568 and 647 nm) and two Neo sCMOS cameras (Andor), using a ×60/NA (numerical aperture) 1.27 water-immersion objective lens. Bandpass emission filters centered on 445, 525, 590 and 675 nm were used for detection. The pixel dimensions of images are 108 × 108 nm, with a theoretical lateral resolution of 214, 252, 283 and 324 nm (for emission at 445, 525, 590 and 675 nm, respectively). Images were acquired with a z-spacing of 0.8 µm, and were maximum-projected during acquisition.

### In situ metabolic labeling of nascent RNA

Cells were pulsed with 5-ethynyl uridine for 30 min before fixation. Nascent RNA was visualized using the Click-iT RNA Alexa Fluor 488 Imaging Kit (Invitrogen), following the manufacturer's instructions except for the substitution of Alexa Fluor 488 azide with Alexa Fluor 647 azide (Invitrogen).

### Iterative indirect immunofluorescence imaging

4i was performed as previously described[4] with two modifications: intercept blocking buffer (LI-COR Biosciences) was used for all blocking, primary and secondary antibody incubations, and 50 mM HEPES (Sigma) was included in imaging buffer, which was adjusted to a pH of 7.4. To detect primary antibodies, goat anti-rabbit IgG Alexa Fluor 568 (Thermo Scientific) was combined with either goat anti-mouse IgG Alexa Fluor 488 (Thermo Scientific) or goat anti-rat IgG Alexa

Fluor 488 (Thermo Scientific), all at a dilution of 1:500. The first cycle included no primary antibodies, to quantify the background level of fluorescence in all cells. Before 4i experiments, all antibodies were tested for compatibility with elution buffer using the following criteria: similar staining on normal and elution buffer-treated cells, minimal residual signal after elution and re-staining with secondary antibody. The following proteins and protein post-translational modifications were measured: ALYREF, CALR, CCNT1, CDK7, CDK9, COIL, CTNNB1, DDX6, GOLGA2, GTF2B, H2B, H3, H3K27ac, H3K4me3, HDAC3, HSPD1, KPNA1, KPNA2, NCL, NONO, PABPC1, PABPN1, pCDK9, PCNA, pMAPK1, PML, POLR2A, POLR2A-S2P, POLR2A-S5P, pRB1, pRPS6, PXN, RPS6, SETD1A, Sm antigen, SON, SP100, SRRM2, SRSF2, TUBA1A, U2SNRNPB and YAP1 (Extended Data Fig. 1). Primary antibodies used are listed in Supplementary Table 1.

### DNA and total protein stain

In cycles 1–7, nuclear DNA was stained using 4′,6-diamidino-2-phenylindole dihydrochloride (DAPI) for 5–10 min at a final concentration of 0.4 µg ml$^{-1}$ in PBS. For cycles 8–22, nuclei were visualized with chicken anti-H2B primary antibody (1:1,000, Abcam) and Goat anti-Chicken IgY Alexa Fluor 405 (1:500, Abcam). Before the last imaging cycle, total protein was stained using Alexa Fluor 647 NHS Ester (succinimidyl ester) (Invitrogen) for 10 min at a final concentration of 0.2 µg ml$^{-1}$ in 50 mM carbonate-bicarbonate buffer, pH 9.2.

### Nuclear and cell segmentation

We typically perform nuclear and cell segmentation as described previously[48], however, this can result in segmentation artifacts when cells are irregularly shaped or highly crowded. To further improve this segmentation, we made use of additional information available in the multiplexed image data. Using DAPI, CALR (endoplasmic reticulum marker) and CTNNB1 (cell–cell contact marker) channels, we manually trained a pixel classifier in Ilastik (v1.3.3) to identify cell–cell boundaries (which were typically high in CTNNB1 and low in CALR). We refer to the probability map generated as 'cell outlines'. To segment nuclei, we first used these outlines to mask the DAPI channel and then thresholded and segmented these objects as 'primary' nuclei. These were then used as seeds on the original thresholded DAPI image to segment 'full' nuclei using propagation. To segment cells, we then summed the total protein and CALR channels and again masked the resulting image with the cell outlines mask to segment 'primary' cells. Finally, the primary cells were used as seeds to obtain the final cell segmentation using a thresholded sum of total protein, CTNNB1 and TUB1A1 channels.

### Data cleanup

After cell segmentation, border cells were excluded. Supervised machine learning models (support vector machines) were trained to exclude polynucleated cells and mitotic cells using the Tissue-MAPS framework (https://github.com/TissueMAPS), as previously described[19]. After this cleanup we noticed that there were still cells with extreme DNA content. These were removed using manually derived thresholds based on histograms of DNA content. Cells with nuclei that moved during image acquisition or which were incompletely acquired in any cycle were identified and removed by examining the correlation of DNA content at the single-cell level across cycles. The first imaging cycle used a secondary antibody only with no primary antibody. Any cells with excessive background in this staining cycle were also removed from analysis. Supplementary Tables 2 and 4 list the number of cells in each of these classes.

### Cell cycle classification

Cell cycle classification for 184A1 cells was performed using a machine learning approach with 5-ethynyl-2′-deoxyuridine (EdU) ground truth data, as previously described[19,20]. The balanced accuracy of the S-phase classifier was 0.97. For HeLa cells, EdU wells were not included for the

SBF2 condition, therefore no independent ground truth was available. In this case, S-phase cells were manually annotated using PCNA and DAPI texture features by iterative supervised support vector machine training in the TissueMAPS framework.

## Datasets for cVAE training

Two datasets were collected for training and evaluating cVAE models. Each dataset was split into training, validation and test cells (80%, 10%, 10%, respectively, for each dataset). Following the split, multiplexed pixel profiles from the cells were extracted together with their local $3 \times 3$ neighbors to make the cVAE latent representation more robust to noise. When one or more of the $3 \times 3$ neighbors of the pixel of interest were outside of the segmented region of the cell, the molecular profile of the missing neighbors was replaced with the mean multiplexed pixel profile inside the $3 \times 3$ window.

The first dataset consisted of 184A1 cells across six chemical treatments (Supplementary Table 3), using 34 channels localizing (at least partially) to the nucleus (ALYREF, CCNT1, CDK7, CDK9, COIL, DAPI, GTF2B, H2B, H3, H3K27ac, H3K4me3, HDAC3, KPNA1, KPNA2, NCL, NONO, PABPC1, PABPN1, pCDK9, PCNA, PML, POLR2A, POLR2A-S2P, POL2RA-S5P, pRB1, RPS6, SETD1A, Sm antigen, SON, SP100, SRRM2, SRSF2, U2SNRNPB, YAP1). For each nucleus in the training and validation split, 0.5% of all molecular profiles were extracted for cVAE training and validation. The second dataset consisted of control and *SBF2* knockdown HeLa cells (Supplementary Table 5), using 43 channels (including all of those used in the first dataset together with pRPS6, pMAPK1, CALR, CTNNB1, PXN, HSPD1, GOLGA2, TUBA1A, DDX6). For each cell in the training and validation split, 5% of all molecular profiles were extracted for cVAE training and validation. See Supplementary Table 6 for the exact number of cells and molecular profiles in each dataset.

## Preprocessing of datasets

Immunofluorescence background levels were determined in each imaging cycle from control wells stained with secondary antibodies (without primary antibodies). These values were subtracted from the molecular profiles. Molecular profiles were normalized using per-channel 98th quantile normalization $x_{norm} = x/q_{98}$.

This background subtraction and normalization was also applied to the multiplexed pixel profiles before obtaining a direct clustering.

## cVAE training

The cVAE models the pixel profiles as samples generated by a generative conditional distribution $p_\theta(x|z,c)$ (also named the probabilistic decoder), where z is a latent variable generated from a prior distribution $p_\theta(z|c)$, and c represents the condition labels (for example, perturbation and cell cycle state of the cell that the current pixel profile is coming from). For a given x, the latent variable z is inferred using a probabilistic encoder $q_\varphi(z|x,c)$, which approximates the intractable true posterior $p_\theta(z|x,c)$. Using variational inference, parameters θ and φ are jointly tuned by maximizing the evidence lower bound of the marginal log-likelihood $\log(p_\theta(x|c))$ (refs. [10],[49]):

$$L(x,c;\theta,\varphi) = E_{q_\varphi(z|x,c)}[\log p_\theta(x|z,c)] - D_{KL}(q_\varphi(z|x,c)||p_\theta(z|c)) \leq \log p_\theta(x|c)$$

With this formulation, the pixel profiles x are modeled by latent distribution z and condition labels, which encourages the model to encode non-condition specific variation (such as subcellular differences in intensity that occur across all conditions) in the latent distribution.

To improve training stability and samples from the decoder, we use σ-VAE[50] to learn the variance of the decoder, to produce calibrated decoders:

$$L(x,c;\theta,\varphi) = D\ln\sigma + \frac{D}{2\sigma^2}\mathrm{MSE}(\hat{x},x) + D_{KL}(q_\varphi(z|x,c)||p_\theta(z|c))$$

with $x \in R^D$ being the center pixel of the input and $\hat{x} = p_\theta(x|z,c)$ the VAE reconstruction of x. We use the analytical solution for the variance[50], which minimizes the (weighted) mean squared error loss (MSE) while also minimizing the logarithm of the variance:

$$\sigma^{*2} = \mathrm{MSE}(x,\mu)$$

where $\mu$ is the estimated latent mean for x. As prior distribution $p_\theta(z|c)$ we choose:

$$p_\theta(z|c) = p(z) = N(0,1).$$

The input to the model was a $3 \times 3$ local neighborhood around the pixel of interest, and the output was the reconstructed center pixel. The encoder consisted of an initial $1 \times 1 \times 32$ convolutional layer to mix the channels of individual pixel inputs, followed by three fully connected layers (32, 16, 16 nodes), and a linear decoder. Conditions were provided via a two-layer condition encoder (10, 10 nodes) to the encoder and decoder by concatenating the learned condition representations with pixel inputs and latent space, respectively. Before concatenating to the pixel inputs, condition representations were broadcast to match the shape of the input patch. The size of the latent representation was 16.

Training was done for 25 epochs with a batch size of 128 and a learning rate of 0.001 (0.0001 for the HeLa dataset). For the 184A1 dataset, the cVAE was trained using perturbation and cell cycle stage as conditions by concatenating one-hot encoded representations of both condition inputs. Note that although the control DMSO treatment and untreated cells were used as different conditions in the cVAE model, there was no significant difference in mean intensity between them and they are pooled together for the remainder of the analysis. Together these untreated and DMSO-treated cells are referred to as 'unperturbed'. For each quantitative comparison between conditions, we validated that DMSO and untreated cells showed no differences. These comparisons are shown in Supplementary Figs. 2a,c and 3a. For the HeLa dataset, the cVAE was similarly trained using siRNA condition and cell cycle stage as conditions.

## Clustering

For clustering, the dataset was subsampled to 150,000 (300,000 for the HeLa dataset) multiplexed pixel profiles. To obtain CSLs, a k-nearest neighbor graph (k = 15) of cVAE latent representations of the subsampled data was computed and partitioned with the Leiden algorithm[13] using a resolution of 0.5 (0.9 for the HeLa dataset). For comparison, the subsampled multiplexed pixel profiles were also directly clustered by applying the Leiden algorithm to the k-nearest neighbor graph of the multiplexed pixel profiles with varying resolutions of 0.2, 0.4, 0.6, 0.8, 1.2, 1.6 and 2.0. To project cluster assignments to the entire dataset, each data point was assigned to the most frequent cluster within 15 nearest neighbors of the subsampled, clustered set. Neighbors were found using approximate nearest neighbor search[51].

To assess the impact of subsampling the data before clustering, we varied the random initialization for the Leiden algorithm (five different initializations), the random seed for the subsampling (five different subsamples) and the size of the subsample, resulting in $5 \times 5$ alternative clusterings for each subsample size of 1,100, 2,300, 4,600, 9,300, 19,000, 37,000, 75,000, 150,000 and 300,000. The overlap of these clusterings with the final CSLs was computed using the adjusted mutual information (AMI)[52] and the adjusted Rand index (ARI)[53,54] (see Supplementary Fig. 1a,b).

Let $U = \{U_1, U_2, \ldots, U_c\}$ be the ground truth CSL clustering, and $V = \{V_1, V_2, \ldots, V_k\}$ any other clustering of n data points. We calculated AMI(U,V) and ARI(U,V) for all alternative clusterings V to compare clusterings to final CSLs. In addition, we computed the overlap of the resulting clusterings with the final annotated CSLs using the

homogeneity score[55] (see Supplementary Fig. 1c) $h = 1 - H(U|V)/H(U)$, with entropy $H(U) = -\sum_{i=1}^{c} \frac{|U_i|}{n} \log(\frac{|U_i|}{n})$.

Homogeneity was calculated for each individual CSL $i$ using a modified $\hat{U} = \{U_i, U/U_i\}$, which contained only the CSL $i$ and one other cluster grouping all other CSLs. This cluster instability analysis could be used to refine the antibody panel for future experiments, by indicating those CSLs for which additional channels might be needed.

To validate CSL pixel assignments, we compared CSLs with manual segmentations of the underlying subcellular structures obtained by training Ilastik[24] (v1.3.3) segmentation models on single-channel intensities of canonical markers for these membraneless organelles (compare with Extended Data Fig. 5). We quantitatively compared these manual segmentations with their respective CSLs using the $F_1$-score (a measurement of classification accuracy) using the manual segmentations as the ground truth: $F_1 = \frac{TP}{TP + 0.5(FP+FN)}$, where TP denotes the number of true positives, FP denotes the number of false positives, and FN, the false negatives.

### Annotation

To aid interpretability we manually annotated CSLs with biologically meaningful labels. CSLs corresponding to the same biological structure may be merged into the same annotated CSL. This annotation was done in an iterative fashion and considered the following factors: presence of canonical organelle markers in the top enriched channels in each CSL in unperturbed (control) cells (if no canonical markers were present, consider the CSL as 'background' (that is, nucleoplasm or cytoplasm)); spatial distribution of CSLs compared with the spatial distribution of canonical markers of organelles in unperturbed (control) cells; and Human Protein Atlas subcellular localization (https://www.proteinatlas.org)[23] of most enriched channels in each CSL, weighted by z-scored channel intensity.

To simplify the presentation of results, we merged those CSLs that, according to the above criteria, correspond to the same biological structure. The merged CSLs either corresponded to the same structure that displayed within-condition variation (for example, Nucleoplasm CSLs in Fig. 2, see Extended Data Fig. 4c) or to different spatial locations of the same biological structure (for example, P-body CSL in Fig. 5, see Extended Data Fig. 8b).

### Feature extraction using CSLs

For quantitative analysis of differences between conditions, several statistics using the CSLs were computed.

**Per-CSL mean intensity.** Per-CSL mean intensity values were calculated for each cell and CSL and averaged for each condition.

**CSL object features.** For each cell and CSL, connected components using 8-connectivity were calculated. To filter out noise and obtain more reliable estimates, only components consisting of more than 10 pixels were counted. In addition, we removed small components from each cell by sorting all components by size and removing the smallest components up to a cumulative area of <10% of the total area of the CSL in that cell (Supplementary Figs. 6 and 7). If no component was smaller than 10% of the total area, no components were removed from that cell.

After filtering, the number, mean or median area, and mean or median circularity of these components was extracted and median-averaged across cells for each condition.

Circularity $c$ was computed as $c = 4\pi a/p^2$ where $a$ is the area and $p$ the perimeter of the component.

**Spatial co-occurrence.** Spatial co-occurrence[12,32] $c_{ij}^{[d_a,d_b]}$ captures the relative probability that two CSLs $(i, j)$ are found within a distance interval $[d_a, d_b]$ from one another:

$$c_{ij} = p^{[d_a,d_b]}(j|i)/p^{[d_a,d_b]}(i)$$

Distance intervals were log-spaced to enable a focus on small-scale changes in spatial reorganization. For the 184A1 dataset, 19 log-spaced distance intervals between 2 and 80 were used. For the HeLa dataset, 27 log-spaced distance intervals between 2 and 320 were used. The maximum distance of 80 pixels (320 pixels) was chosen to be approximately the 99th quantile of the maximum radius of the nucleus (of the cell for the HeLa dataset).

### Statistical analysis of mean intensity and CSL abundance changes

To quantify the changes in channel intensities in CSLs, we estimated the fold-difference of each channel between treated and unperturbed control cells in the geometric mean of the per-CSL mean intensity. Specifically, if $Y_{ijk}$ denotes the mean intensity for CSL $k$ in cell $j$ of well $i$, we fit a hierarchical linear mixed-effects model:

$$\log(Y_{ijk}) = \mu_k + \gamma_k t_i + \beta_{ik} + \epsilon_{ijk}$$

where $\mu_k$ denotes the (log) geometric mean of CSL $k$ in the control group and $t_i$ is an indicator variable for condition ($t_i = 1$ for treated wells and 0 for unperturbed control wells), such that $\exp(\gamma_k)$ is the treatment effect on CSL $k$. To account for clustering, $\beta_i \sim N(0, \Sigma_w)$ is a multivariate normal well-specific random effect, with mean zero and general covariance matrix $\Sigma_w$, and $\epsilon_{ij}$ is a multivariate normal random error with mean zero and covariance matrix $\Sigma_{\epsilon(i)} = S_{t_i} R S_{t_i}$ where $S_{t_i}$ is a diagonal matrix of (condition-specific) standard deviations of the CSL-specific errors, and $R$ is an unstructured correlation matrix that captures the relationships between CSLs in a single cell. Before calculating fold-differences and hypothesis testing, we removed compartments of size zero.

For each CSL we tested the null hypothesis of no treatment effect ($\gamma_k = 0$) using a Wald test. To determine relative relocalization of proteins and protein states rather than overall changes in abundance, the fold-changes in each CSL were normalized by the whole-nucleus fold-changes. That is, if $k = 0$ denotes mean intensity across the whole nucleus, then $\exp(\gamma_k - \gamma_0)$ is the compartment-specific treatment effect for CSL $k$, and we similarly tested $\gamma_k = \gamma_0$ using a Wald test. CSL sizes were analyzed in the same way as mean channel intensities.

We used the nlme package[56] (v3.1–153) in R v3.6.3 (ref. [57]) to fit these models, and used emmeans[58] (v1.7.0) to extract estimates and perform the hypothesis tests of interest. For computational efficiency, we fitted a separate model for each CSL for each marker (using only the data from that CSL and the whole nucleus), and used the conservative 'containment' method[59] to determine the degrees of freedom of the Wald statistic in the analyses of CSL versus whole-nucleus differences. The false discovery rate was controlled across all combinations of CSLs and channels for each treatment using the Benjamini–Yekutieli method[60].

### Comparison of perturbations

To compare perturbations with respect to different aspects of cellular organization, we generated three separate cellular representations: mean nuclear intensities of all proteins; per-CSL mean intensities of all proteins; and pairwise spatial co-occurrence between CSLs.

To measure how well these different cellular representations separate cells from different perturbations, we calculated silhouette scores[61] (using L1 distance) $S(p,q)$ for each pair of perturbations $p, q$:

$$S(p, q) = \frac{1}{|p|} \sum_{i \in p} \frac{(d_q(i) - d_p(i))}{\max(d_p(i), d_q(i))}$$

With $d_p(i)$ being the mean $L1$ distance of $i$ to all elements in perturbation $p$:

$$d_p(i) = \frac{1}{|p|} \sum_{j \in p} L1(i,j)$$

### Reporting summary

Further information on research design is available in the Nature Portfolio Reporting Summary linked to this article.

### Data availability

The data used to generate all results and figures reported in this manuscript are available at https://doi.org/10.5281/zenodo.7299516 (ref. 62). Pre-trained models and clusterings reported in the manuscript are available at https://doi.org/10.5281/zenodo.7299750 (ref. 63). CSL-derived features from the 184A1 and the HeLa datasets are available at https://doi.org/10.6084/m9.figshare.19699651. The Human Protein Atlas, used to annotate CSLs, is available at www.proteinatlas.org.

### Code availability

Analysis was performed using CAMPA, which is available at https://github.com/theislab/campa with documentation at https://campa.readthedocs.io. All scripts necessary for reproducing the results and figures (except schematic figures Fig. 1a–c and Fig. 3f) can be found at https://github.com/theislab/campa_ana. Nuclear and cell segmentation, identification of border cells, and cell cycle classification was performed using TissueMAPS, an open-source project for high-throughput image analysis available at https://github.com/pelkmanslab/TissueMAPS. The TissueMaps analysis pipeline description with module files containing parameter settings used for the preprocessing of data in this paper is provided at https://github.com/theislab/campa_ana.

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

### Acknowledgements

The authors thank all members of L.P.'s and F.J.T.'s laboratories for discussions and manuscript comments. The authors also thank D. Spector for the anti-SNRPB2 and Sm antibodies; A. Fox for the NONO antibody; K. Weis for the KPNA1 and KPNA2 antibodies; and K. Koide for providing meayamycin. S.B. acknowledges support from a European Molecular Biology Organisation long-term fellowship (ALTF1175-2016), a Human Frontiers Science Programme long-term fellowship (LT000238/2017-L), an Australian Research Council Discovery Early Career Researcher Award (DE230100271) and the University of New South Wales. L.P. is supported by the European Research Council (ERC-2019-AdG-885579) and the Swiss National Science Foundation (SNSF grant 310030_192622). S.B. and L.P. acknowledge support from the University of Zurich. H.S. and F.J.T. acknowledge support from the German Federal Ministry of Education and Research (BMBF) (031L0210A) and from the Helmholtz Association's Initiative and Networking Fund through Helmholtz AI (ZT-I-PF-5-01).

### Author contributions

H.S., S.B., L.P. and F.J.T. conceived the study and wrote the manuscript; H.S. wrote the CAMPA code; S.B. performed experiments; H.S. and S.B. designed and performed the data analysis; M.D. designed and implemented statistical analysis of mean intensity and CSL abundance changes; L.P. and F.J.T. supervised the work; all authors read and corrected the final manuscript.

### Funding

### Competing interests

F.J.T. consults for Immunai Inc., Singularity Bio B.V., CytoReason Ltd and Omniscope Ltd, and has ownership interest in Dermagnostix GmbH and Cellarity. L.P. has filed a patent on the 4i technology, which is used in this manuscript (patent WO2019207004A1), and consults for Dewpoint Therapeutics and has ownership interest in Sagimet Biosciences and Apricot Therapeutics. All other authors have no competing interests.

### Additional information

**Extended data** are available for this paper at https://doi.org/10.1038/s41592-023-01894-z.

**Correspondence and requests for materials** should be addressed to Lucas Pelkmans or Fabian J. Theis.

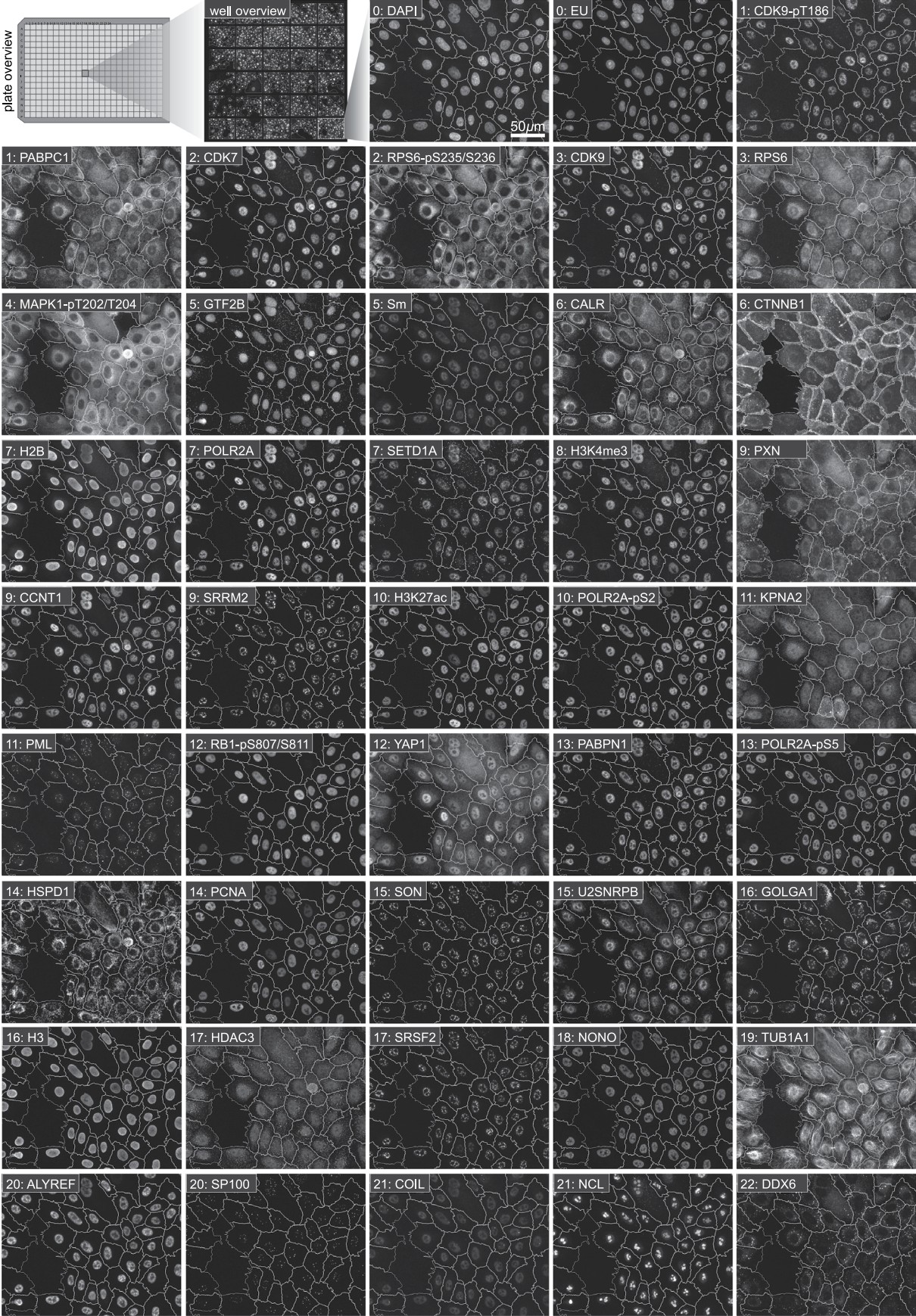

**Extended Data Fig. 1 | Iterative Indirect Immunofluorescence Imaging (4i) at high spatial resolution.** Example intensity images with overlaid cell segmentation of unperturbed 184A1 cells for each of the 43 channels measured by 4i (n=4 unperturbed replicate wells imaged; cell numbers in Supplementary Table 3).

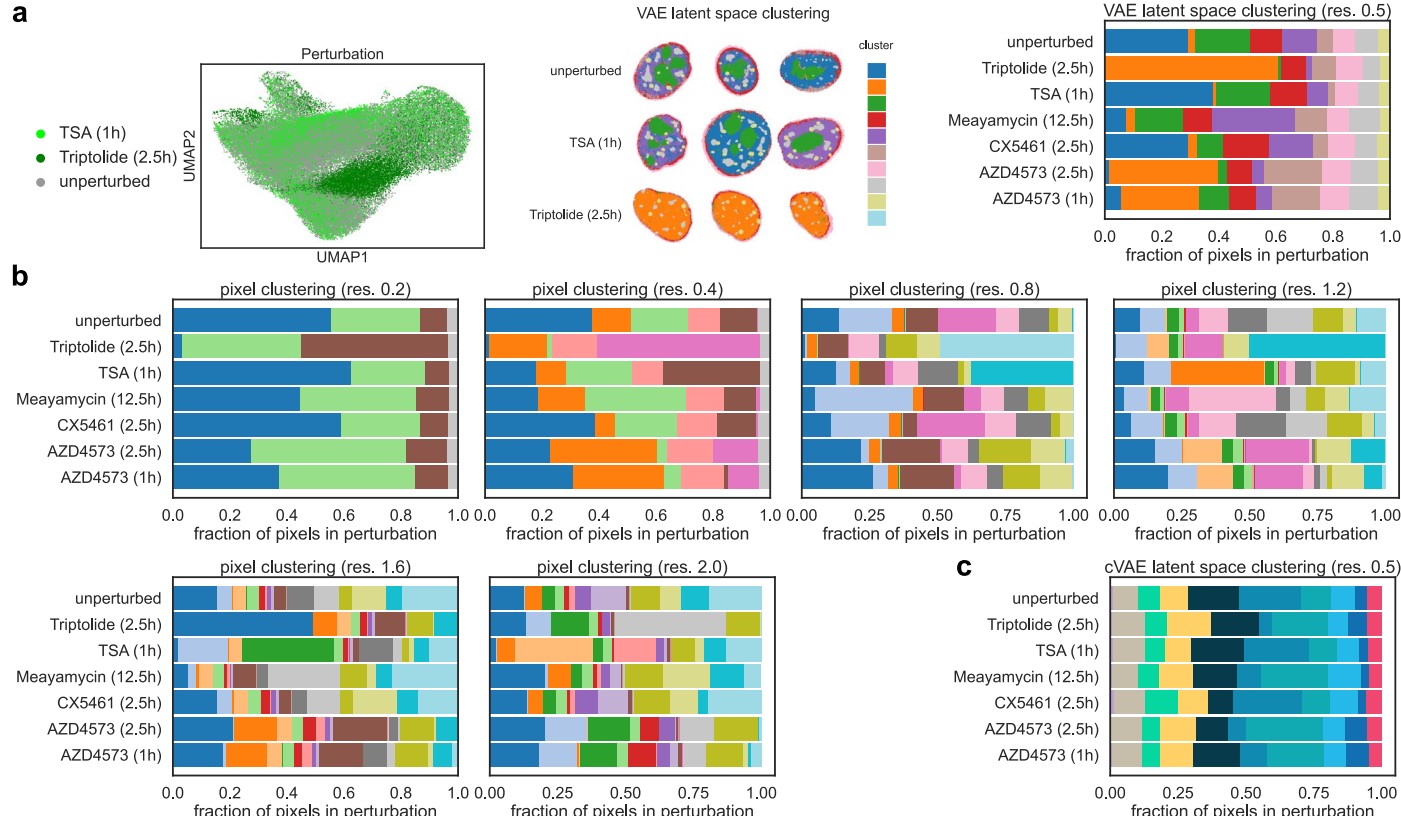

**Extended Data Fig. 2 | Direct pixel clustering and VAE latent space clustering are perturbation dependent at different Leiden clustering resolutions. a:** Left: UMAP representation of VAE latent space colored by perturbation. Middle: Example cells from each perturbation colored by VAE latent space clustering. Right: Fraction of pixels assigned to each cluster per perturbation colored by VAE latent space clustering. **b:** Fraction of pixels assigned to each cluster per perturbation colored by direct pixel clustering for Leiden resolutions 0.2, 0.4, 0.6, 0.8, 1.2, 1.6, 2.0 **c:** Fraction of pixels assigned to each cluster per perturbation colored by cVAE latent space clustering (CSLs) before annotation.

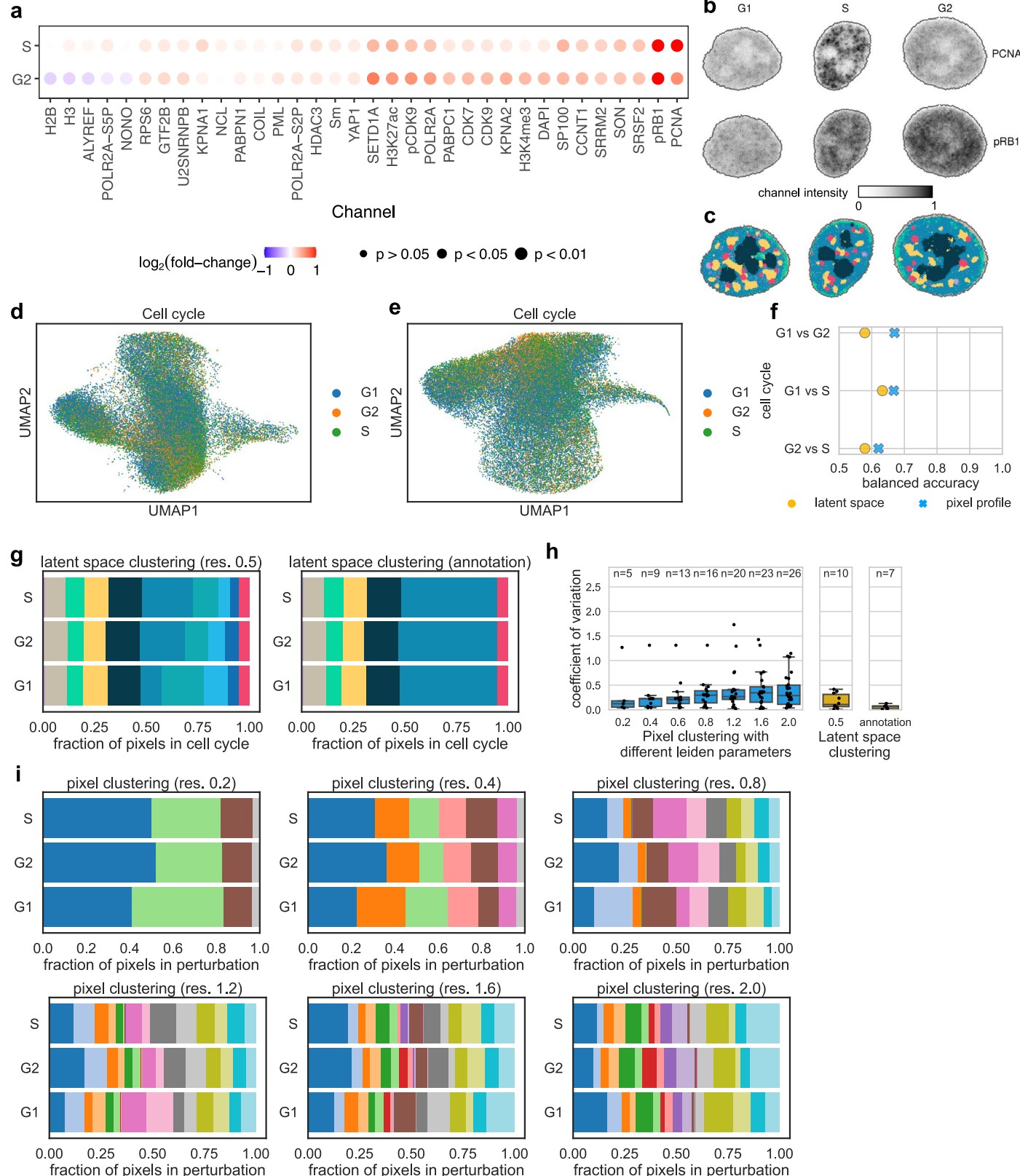

**Extended Data Fig. 3 | See next page for caption.**

**Extended Data Fig. 3 | Direct pixel clustering is cell cycle-dependent at different Leiden clustering resolutions. a:** Fold-change in nuclear mean intensity in S and G2 cells compared to G1 cells, for all proteins with nuclear localization. P-values show significance of perturbation effect on mean intensity, as determined from the mixed-effect model (Wald test, multiple testing correction using Benjamini–Yekutieli method). **b:** Example intensity images of cell cycle specific markers PCNA and pRB1 in cells from G1, S, and G2 cell cycle phases (d). **c:** Example cells in b, colored by CSL clusters. **d-e:** UMAP representation of multiplexed pixel profiles from unperturbed cells (d) and of corresponding cVAE latent space (e) colored by cell cycle. **f:** Comparison of cell cycle-specificity of 4i pixel profiles and cVAE latent space coordinates. Plots show balanced accuracy scores of pairwise binary logistic regression classifiers predicting cell cycle from normalized 4i pixel profiles or latent representations of

pixels. Accuracy values of 0.5 indicate random chance (perturbation information is not present in the data). Latent space contains less cell cycle information than pixel profiles. **g:** Fraction of pixels assigned to each cluster per cell-cycle stage colored by latent space clustering and annotation. **h:** Comparison of cell-cycle-specificity of direct clustering at different Leiden resolutions (0.2, 0.4, 0.6, 0.8, 1.2, 1.6, 2.0) with cVAE latent space clustering. Plots show coefficient of variation of the fraction of pixels in each cell-cycle stage assigned to each cluster. Boxplot summarises results for all clusters with resulting cluster number n shown above. Center line, median; box limits, upper and lower quartiles; whiskers, 1.5x interquartile range; points, all data points. **i:** Fraction of pixels assigned to each cluster per cell-cycle stage colored by pixel profile clustering for resolutions 0.2, 0.4, 0.8, 1.2, 1.6, 2.0.

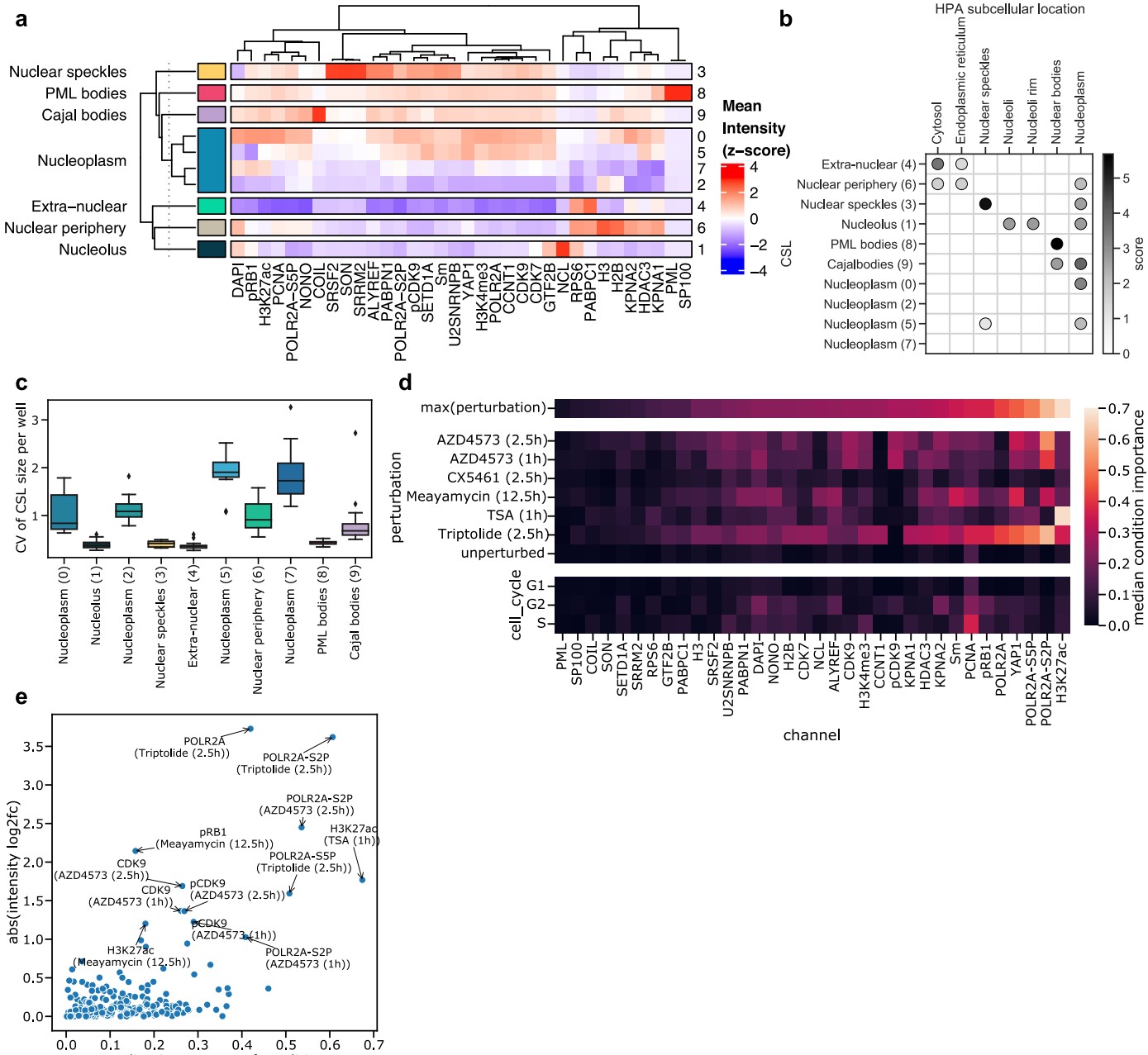

**Extended Data Fig. 4 | Details of cVAE clustering and annotation (184A1 cells). a:** Mean intensities of each channel across different clusters (unperturbed 184A1 cells). Original Leiden clusters obtained from cVAE latent space are shown on the right and their corresponding annotation is shown on the left. Four original clusters are manually merged into the 'Nucleoplasm' CSL, as described in the main text. Values z-scored by channel (compare with Figure 2c). **b:** Automated annotations from the Human Protein Atlas. For each CSL a maximum of 3 channels with a z-score >1 (from a) were used to query HPA for subcellular localization of these channels. Shown is the result of these queries for each CSL, weighted by the mean z-score of the channel (shown in a). **c:** Coefficient of variation (CV) of CSL sizes in each cell across all cells in a well. Original nucleoplasm clusters have a large CV (variable abundance between cells), and are merged to one annotated CSL (shown in a). Center line, median; box limits, upper and lower quartiles; whiskers, 1.5x interquartile range; points, outliers. n=21 wells from 9 perturbations. **d:** Importance of condition input for reconstructing each channel. Saliency scores are calculated with integrated

gradients[22] for every output channel with respect to the latent representation and condition input. The importance of the condition for each output channel is calculated as the fraction of absolute scores for the condition, normalized by the absolute sum of scores for the entire input (condition and latent representation). Shown is the maximum condition importance of each channel; the median condition importance for each perturbation and channel; and the median condition importance for each cell cycle stage and channel. Large values indicate that the condition information was important for correctly reconstructing this channel, and the cVAE focused less on the latent representation. **e:** Scatter plot of condition importance (shown in d) vs Log2 fold-change in overall cellular intensity for each channel and perturbation (shown in Fig. 1d). Channels that show an overall perturbation-dependent effect in intensity also have high condition-importances, meaning that the cVAE preferentially uses the condition label and not the latent representation to model the pixel intensities for these channels.

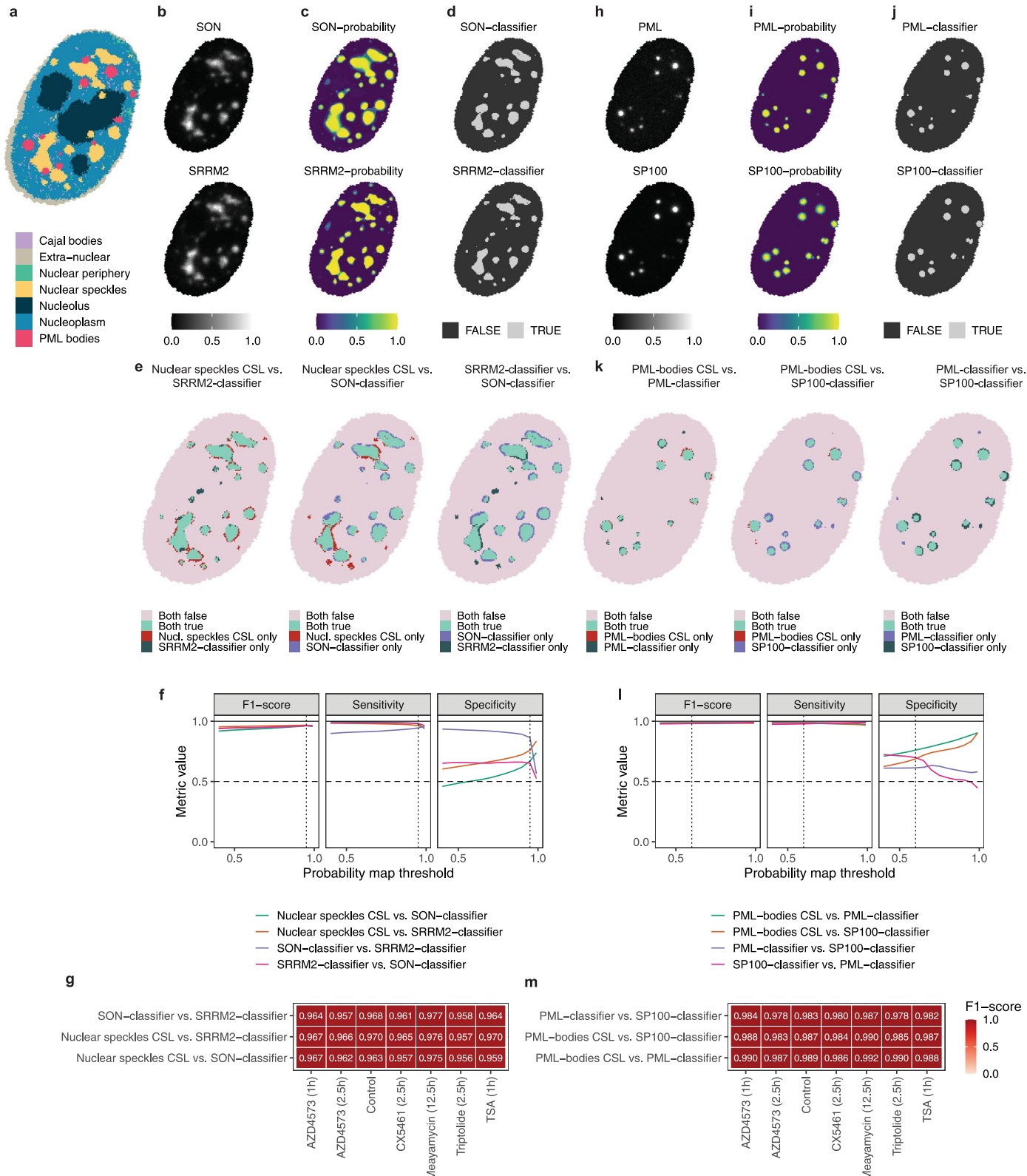

**Extended Data Fig. 5 | See next page for caption.**

**Extended Data Fig. 5 | Comparison of CSL-derived nuclear speckles/PML bodies with supervised segmentation of nuclear speckles/PML bodies.** **a:** CSLs generated using CAMPA for an example 184A1 nucleus. **b:** Measured intensities of canonical nuclear speckle markers SON and SRRM2. **c:** Probability maps of nuclear speckles generated through supervised pixel clustering of single-channel images in Ilastik. **d:** Nuclear speckle classifier derived from thresholding the probability map at P=0.95. **e:** Comparison of pixels assigned to nuclear speckles using supervised classifiers based on SON, SRRM2 with nuclear speckle CSL. **f:** Quantitative comparison of the different classification approaches using SON and SRRM2 classifiers as the ground truth. x-axis shows the value at which the probability map is thresholded to generate the segmentation. Dashed line at P=0.95 is the classification shown in d. Metrics computed on a random 10% subsample of all data. **g:** $F_1$-scores for each comparison for all perturbations individually. Data from all pixels in a single well for each condition. Probability map threshold at P=0.95. **h-j:** As in b-d for canonical PML body markers SP100 and PML. Classification in i is the probability map thresholded at P=0.6. **k:** As in e for PML bodies. **l:** As in f for PML bodies. Dashed line at P=0.6 is the classification shown in j. **m:** As in g for PML bodies. Probability maps thresholded at P=0.6.

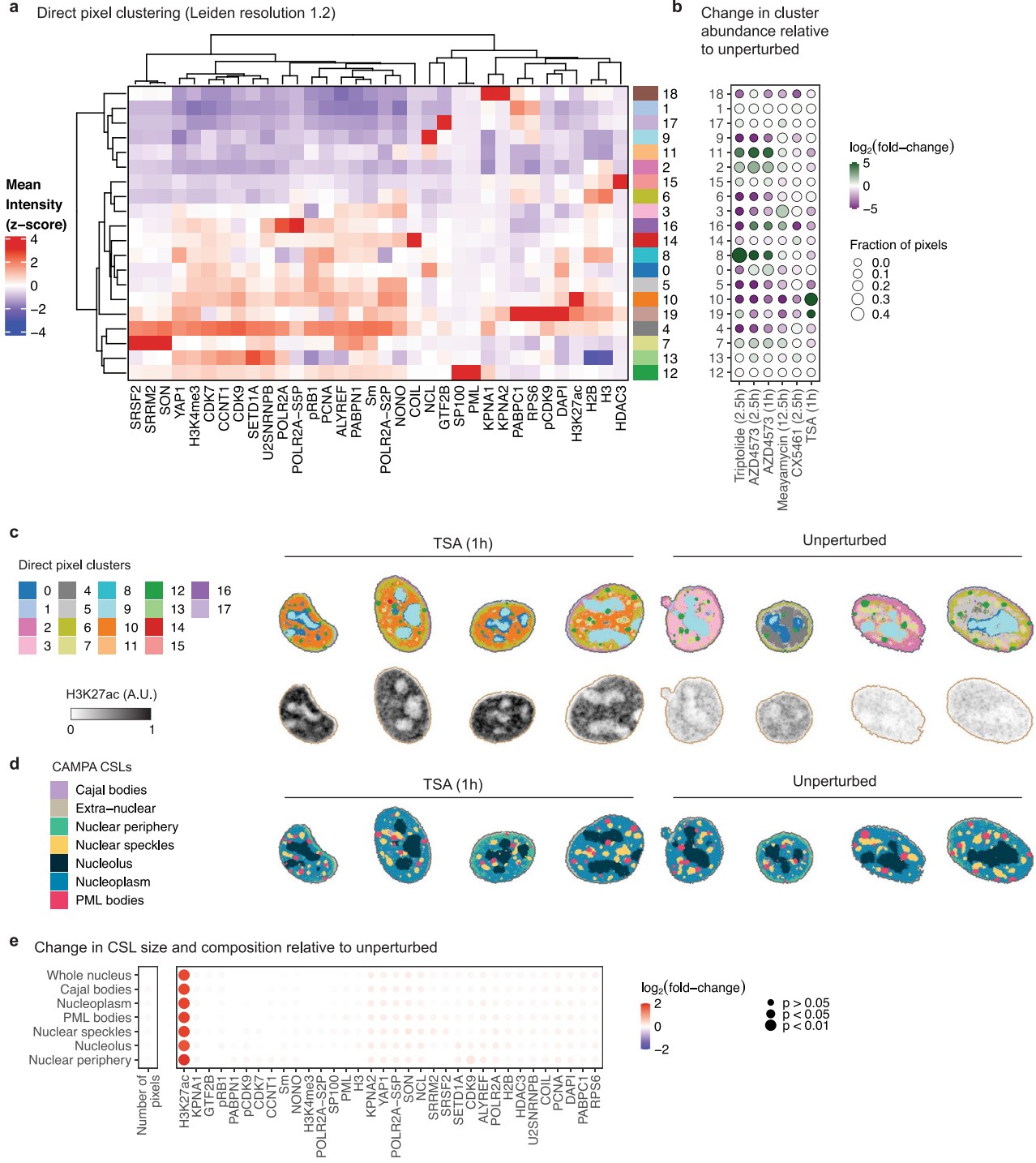

**Extended Data Fig. 6 | Detailed comparison of CAMPA and direct pixel clustering in perturbed 184A1 nuclei. a:** Mean intensities of each channel across different clusters. Clusters obtained from direct Leiden clustering (resolution 1.2) of pixel profiles for all 34 nuclear channels across all six experimental conditions. Clusters are those shown in Fig. 1h. Values z-scored by channel. **b:** Log2 fold-change in relative cluster abundance in perturbation conditions, compared to unperturbed cells. **c:** Example cells treated with TSA or unperturbed. Direct pixel clusters and H3K27ac levels shown. **d:** CAMPA-derived CSLs for the cells shown in c. **e:** Log2 fold-change of mean intensities for each channel in each CSL, or number of pixels in each CSL, when comparing TSA-treated with unperturbed control cells. P-values show significance of TSA treatment on levels for each channel/CSL, as determined from a mixed-effect model (Wald test, multiple testing correction using Benjamini–Yekutieli method).

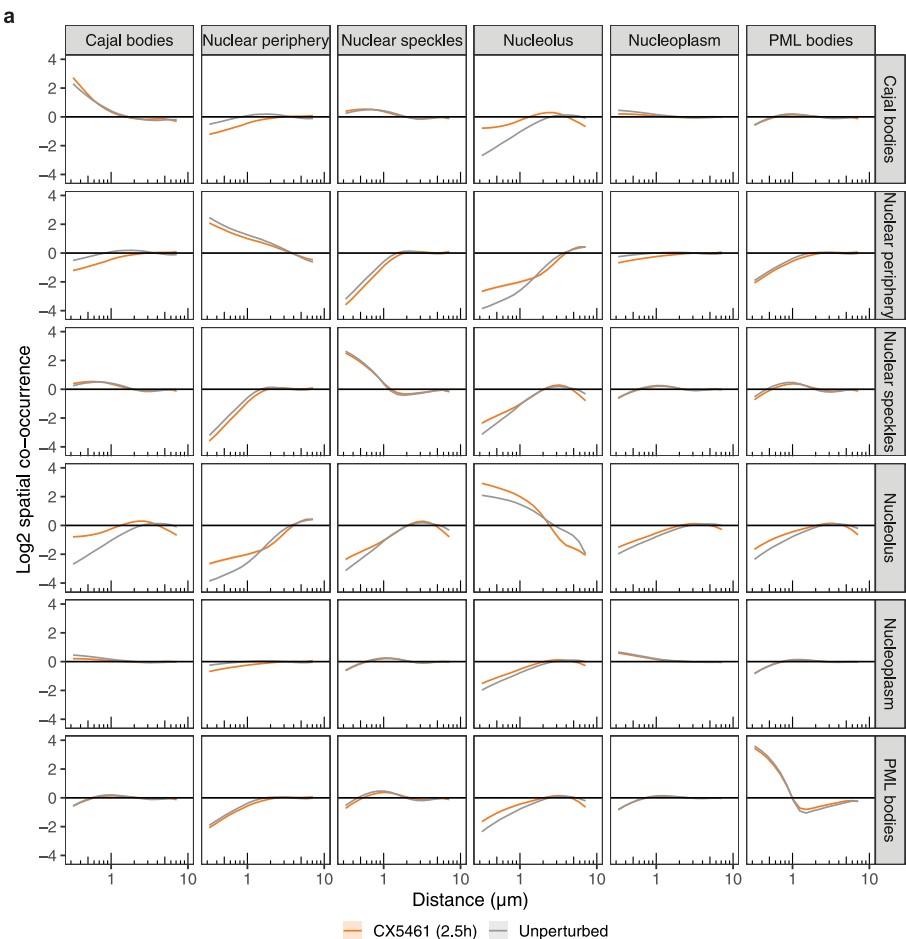

**Extended Data Fig. 7 | Pairwise co-occurrence scores between CSLs. a**: Pairwise mean log2 spatial co-occurrence between all CSLs as a function of distance on x-axis (minimum of distance interval; on log scale) for CX5461-treated and unperturbed cells. Shaded regions indicate 95% confidence intervals for the mean.

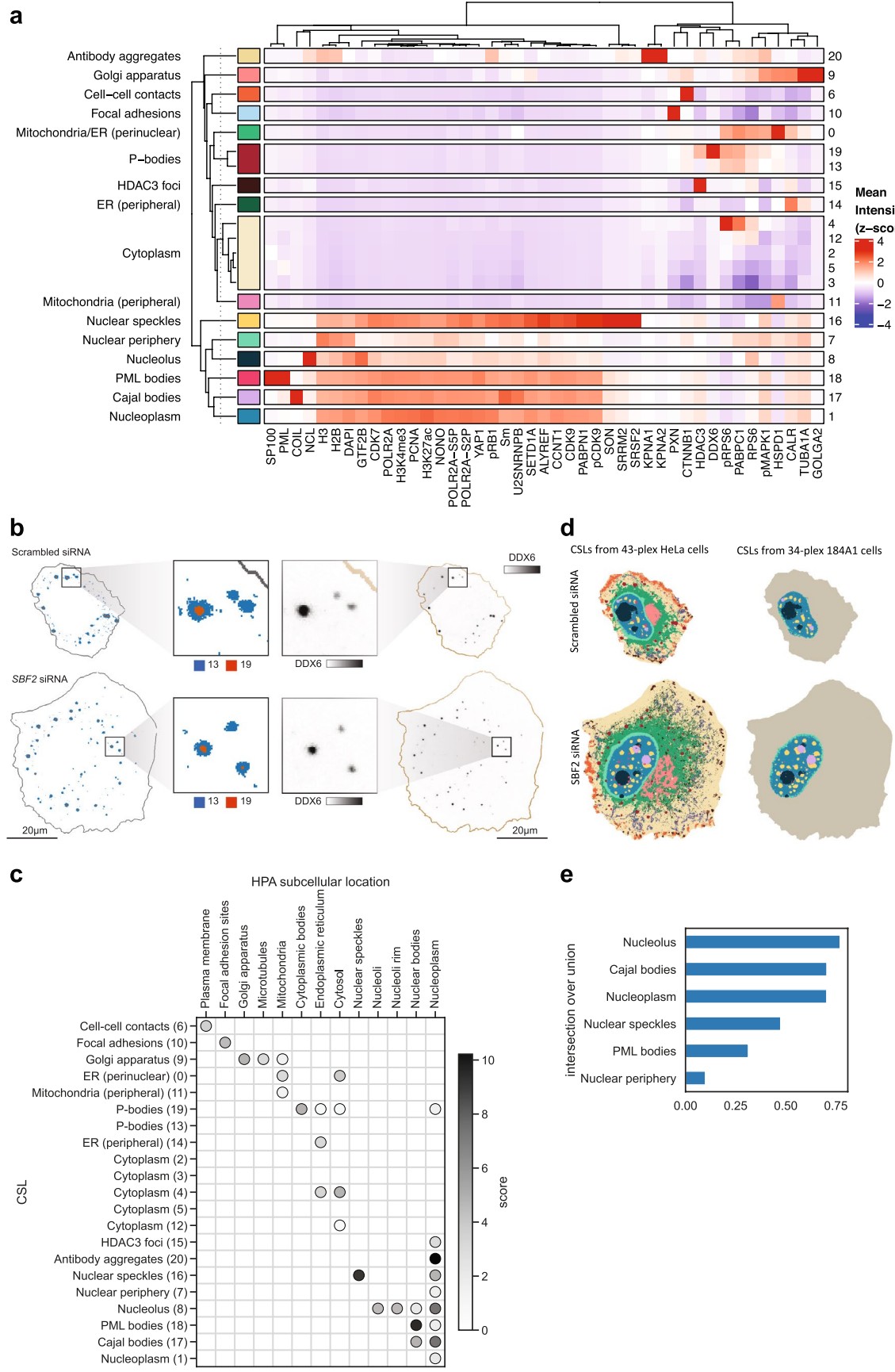

**Extended Data Fig. 8 | See next page for caption.**

**Extended Data Fig. 8 | Details of cVAE clustering and annotation (HeLa cells, scrambled siRNA). a:** Mean intensities of each channel across different clusters. Original Leiden clusters obtained from cVAE latent space are shown on the right and their corresponding manual annotation is shown on the left. Five original clusters are manually merged into the 'Cytoplasm' CSL. Two original clusters are manually merged into the 'P-bodies' cluster. Values z-scored by channel (c.f., Fig. 5b). **b:** Cluster 13 and 19, which are merged into the 'P-bodies' cluster, and DDX6 intensity visualized in two example cells from scrambled siRNA and SBF2 siRNA. Cluster 19 corresponds to P-body center with very high DDX6 intensity, cluster 13 corresponds to P-body periphery with lower DDX6 intensity. Both P-body clusters show identical behavior when comparing scrambled to SBF2 (Supplementary Fig. 5b, d). **c:** Automated annotations from the Human Protein

Atlas (HPA). For each CSL a maximum of 3 channels with a z-score >1 (from a) were used to query HPA for subcellular localization of these channels. Shown is the result of these queries for each CSL, weighted by the mean z-score of the channel (shown in a). **d:** Example HeLa cells colored by CSLs obtained from 43-plex HeLa cells (c.f., Fig. 5) and CSLs obtained from 34-plex 184A1 cells (nucleus only, c.f., Fig. 2). **e:** Comparison of CSLs obtained from 43-plex HeLa cells and CSLs obtained from 34-plex 184A1 cells on HeLa cells. Shown is the intersection over union (IOU) of the common CSLs from both clusterings. Nucleolus, Cajal bodies, Nucleoplasm and Nuclear speckles have high IOU scores. Nuclear speckles, PML bodies, and Nuclear periphery have lower IOU scores, indicating either cell-line specific differences, or that information from the additional channels in the 43-plex HeLa dataset results in a slightly different definition of these CSLs.

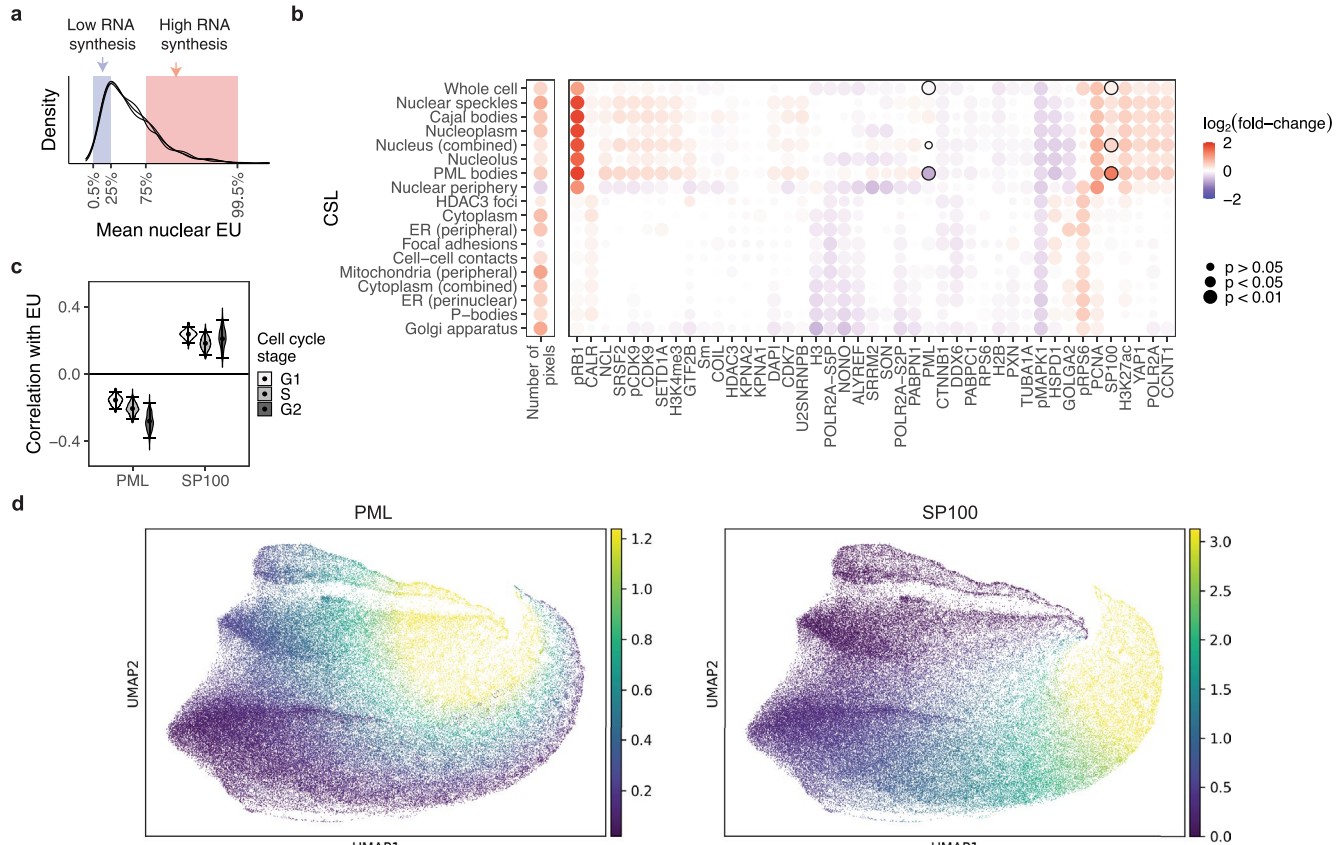

**Extended Data Fig. 9 | Cellular RNA synthesis rates are associated with altered molecular composition of PML bodies. a:** Distribution of RNA synthesis rates in the cell population, as quantified by mean nuclear EU incorporation. Cells are binned into upper and lower EU intensity quartiles. **b:** Log2 fold-change of mean intensities for each channel in each CSL, or number of pixels in each CSL, when comparing HeLa cells with high RNA synthesis to cells with low RNA synthesis. P-values show significance of the difference in mean intensity for each channel/CSL, as determined from the mixed-effect model (Wald test, multiple testing correction using Benjamini–Yekutieli method). **c:** Pearson correlation of mean PML body intensity of PML and SP100 with mean nuclear EU in each cell cycle phase. Violin plots show the distribution over 500 bootstrapped correlations with error bars representing 95% confidence intervals. Center dot, mean. **d:** UMAP of cVAE latent representation of 10% of all PML body pixels from scrambled HeLa cells. Pixels are colored by PML body markers SP100 (left) and PML (right). PML body pixels are heterogeneous, from PML-enriched to SP100-enriched, and the latent representation conserves this heterogeneity. At larger Leiden resolutions this CSL might be split up into several clusters.

# Reporting Summary

## Statistics

For all statistical analyses, confirm that the following items are present in the figure legend, table legend, main text, or Methods section.

| n/a | Confirmed | |
|---|---|---|
| ☐ | ☒ | The exact sample size ($n$) for each experimental group/condition, given as a discrete number and unit of measurement |
| ☐ | ☒ | A statement on whether measurements were taken from distinct samples or whether the same sample was measured repeatedly |
| ☐ | ☒ | The statistical test(s) used AND whether they are one- or two-sided<br>*Only common tests should be described solely by name; describe more complex techniques in the Methods section.* |
| ☒ | ☐ | A description of all covariates tested |
| ☐ | ☒ | A description of any assumptions or corrections, such as tests of normality and adjustment for multiple comparisons |
| ☐ | ☒ | A full description of the statistical parameters including central tendency (e.g. means) or other basic estimates (e.g. regression coefficient) AND variation (e.g. standard deviation) or associated estimates of uncertainty (e.g. confidence intervals) |
| ☐ | ☒ | For null hypothesis testing, the test statistic (e.g. $F$, $t$, $r$) with confidence intervals, effect sizes, degrees of freedom and $P$ value noted<br>*Give P values as exact values whenever suitable.* |
| ☒ | ☐ | For Bayesian analysis, information on the choice of priors and Markov chain Monte Carlo settings |
| ☐ | ☒ | For hierarchical and complex designs, identification of the appropriate level for tests and full reporting of outcomes |
| ☐ | ☒ | Estimates of effect sizes (e.g. Cohen's $d$, Pearson's $r$), indicating how they were calculated |

*Our web collection on statistics for biologists contains articles on many of the points above.*

## Software and code

Policy information about availability of computer code

| Data collection | Data were acquired using an automated spinning-disk microscope (CellVoyager 7000, Yokogawa), using the proprietary CV7000 software (version R1.17.05). |
|---|---|
| Data analysis | Analysis was performed using CAMPA which is available at https://github.com/theislab/campa with docs at https://campa.readthedocs.io. All scripts necessary for reproducing the results and figures  (except schematic figure panels Fig. 1a,b,c, Fig. 3f) can be found at https://github.com/theislab/campa_ana.<br>Nuclear and cell segmentation, identification of border cells, and cell-cycle classification was performed using TissueMAPS, an open-source project for high-throughput image analysis which is available at https://github.com/pelkmanslab/TissueMAPS. The TissueMaps analysis pipeline description with module files containing parameter settings used for the preprocessing of data in this paper is provided at https://github.com/theislab/campa_ana.<br>We used Ilastik (version 1.3.3) to train a model for nuclear and cell segmentation.<br>For statistical analysis of mean intensity and CSL abundance changes, we used the nlme package (version 3.1-153) in R version 3.6.3 to fit the mixed models and used emmeans (version 1.7.0) to extract estimates and perform hypothesis tests. |

For manuscripts utilizing custom algorithms or software that are central to the research but not yet described in published literature, software must be made available to editors and reviewers. We strongly encourage code deposition in a community repository (e.g. GitHub). See the Nature Portfolio guidelines for submitting code & software for further information.

## Data

Policy information about availability of data

All manuscripts must include a data availability statement. This statement should provide the following information, where applicable:

- Accession codes, unique identifiers, or web links for publicly available datasets
- A description of any restrictions on data availability
- For clinical datasets or third party data, please ensure that the statement adheres to our policy

The data used to generate all results and figures reported in this manuscript is available at https://doi.org/10.5281/zenodo.7299516. Pre-trained models and clusterings reported in the manuscript are available at https://doi.org/10.5281/zenodo.7299750.
CSL-derived features from the 184A1 and the HeLa datasets are available at https://doi.org/10.6084/m9.figshare.19699651.
The human protein atlas that was used to annotate CSLs is available at www.proteinatlas.org.

## Human research participants

Policy information about studies involving human research participants and Sex and Gender in Research.

| Reporting on sex and gender | N/A |
|---|---|
| Population characteristics | N/A |
| Recruitment | N/A |
| Ethics oversight | N/A |

Note that full information on the approval of the study protocol must also be provided in the manuscript.

# Field-specific reporting

Please select the one below that is the best fit for your research. If you are not sure, read the appropriate sections before making your selection.

☒ Life sciences          ☐ Behavioural & social sciences          ☐ Ecological, evolutionary & environmental sciences

For a reference copy of the document with all sections, see nature.com/documents/nr-reporting-summary-flat.pdf

# Life sciences study design

All studies must disclose on these points even when the disclosure is negative.

| | |
|---|---|
| Sample size | No sample-size calculation was performed. We sampled 1000-3000 cells per condition, across 2-4 replicate wells per condition (Supplementary Tables 3 and 5) based on experimental feasibility. This sample size enables statistical comparisons between conditions using mixed-effects models with random effects for each replicate well, as described in Methods. |
| Data exclusions | Mitotic and polynucleated cells were excluded during data cleanup, as described in Methods, and quantified in Supplementary Tables 2 and 4. In Figure 4a, a small subset of outlier cells were excluded from the UMAP plot. Justification of this exclusion is provided in Supplementary Figure 10, as indicated in Figure 4 legend. When analyzing CSL object features, small objects were removed as described in Methods, and indicated in Figure legends. |
| Replication | Data were from a single experiment. Replicate wells were included for all conditions. |
| Randomization | Plate layout was designed so that replicate wells were in different rows and columns of the plate, and were periodically interspersed with negative control wells with respect to image acquisition time. |
| Blinding | Investigators were not blinded during the experiments because we wanted to validate success of experimental perturbations (visually) before proceeding with several weeks of expensive immunofluorescence analysis. The analysis method is predominantly automated, from image acquisition to unsupervised training of machine learning models, to statistical analysis of quantitative data. An exception to this is the supervised training of machine-learning models as described in "Data exclusions". |

# Reporting for specific materials, systems and methods

We require information from authors about some types of materials, experimental systems and methods used in many studies. Here, indicate whether each material, system or method listed is relevant to your study. If you are not sure if a list item applies to your research, read the appropriate section before selecting a response.

## Materials & experimental systems

| n/a | Involved in the study |
|-----|----------------------|
| ☐ | ☒ Antibodies |
| ☐ | ☒ Eukaryotic cell lines |
| ☒ | ☐ Palaeontology and archaeology |
| ☒ | ☐ Animals and other organisms |
| ☒ | ☐ Clinical data |
| ☒ | ☐ Dual use research of concern |

## Methods

| n/a | Involved in the study |
|-----|----------------------|
| ☒ | ☐ ChIP-seq |
| ☒ | ☐ Flow cytometry |
| ☒ | ☐ MRI-based neuroimaging |

## Antibodies

| | |
|---|---|
| Antibodies used | Refer to Supplementary Table 1 |
| Validation | No additional validation undertaken for this study. Information for each can be found using the Research Resource Identifiers (RRIDs) in Supplementary Table 1 |

## Eukaryotic cell lines

Policy information about cell lines and Sex and Gender in Research

| | |
|---|---|
| Cell line source(s) | HeLa Kyoto (female) cell populations were derived from a single-cell clone (Battich et al., 2015). 184A1 ((ATCC CRL-8798; human female breast epithelial) cell populations were derived from a single-cell clone (Kramer et al., 2022). |
| Authentication | HeLa Kyoto (female) cell populations were tested for identity by karyotyping (Battich et al., 2015). 184A1 (human female breast epithelial) cell populations were not authenticated. |
| Mycoplasma contamination | All cells tested negative for mycoplasma contamination. |
| Commonly misidentified lines (See ICLAC register) | No commonly misidentified lines were used |

