## [Peer Review File · Nature Methods]

Peer Review Information

Manuscript Title: Learning consistent subcellular landmarks to quantify changes in multiplexed protein maps

Corresponding author name(s): Fabian Theis

Editorial Notes: n/a

Reviewer Comments & Decisions:

Decision Letter, initial version:

Dear Fabian,

Your Article entitled "Learning consistent subcellular landmarks to quantify changes in multiplexed protein maps" has now been seen by three reviewers, whose comments are attached. While they find your work of potential interest, they have raised serious concerns which in our view are sufficiently important that they preclude publication of the work in Nature Methods, at least in its present form.

As you will see, the reviewers raise concerns about the performance and benchmarking of the approach and about its presentation.

Should further experimental data allow you to fully address these criticisms we would be willing to look at a revised manuscript (unless, of course, something similar has by then been accepted at Nature Methods or appeared elsewhere). This includes submission or publication of a portion of this work somewhere else. We hope you understand that until we have read the revised paper in its entirety we cannot promise that it will be sent back for peer-review.

If you are interested in revising this manuscript for submission to Nature Methods in the future, please contact me to discuss your appeal before making any revisions (this is the revision plan we discussed earlier). Otherwise, we hope that you find the reviewers' comments helpful when preparing your paper for submission elsewhere.

Sincerely,
Rita

Rita Strack, Ph.D.
Senior Editor
Nature Methods

Reviewers' Comments:

Reviewer #1:

Remarks to the Author:

A. Summary of the key results: In the manuscript "Learning consistent subcellular landmarks to quantify changes in multiplexed protein maps" the authors develop a computational framework to learn consistent molecular landmarks from highly multiplexed quantitative subcellular landmarks. The main challenge that the authors address is the confusion between differences in molecular profiles and their variation across space. To address this the authors introduce CAMP -- a conditional auto-encoder for multiplexed pixel analysis -- and then go on to demonstrate its utility with use cases.

B. Originality and significance: The work is original in that there is not a lot of highly multiplexed data out there yet, and even less algorithms for that data. However, the manuscript could benefit from comparison to a better baseline. Comparing to naive pixel clustering is perhaps quite limited. Another question is what parts of the model used are important and for which reasons, an ablation study could be useful.

C. Data & methodology: The approach is valid, the data appears to be of high quality, and the presentation of the method, data and data is high.

D. Appropriate use of statistics and treatment of uncertainties: Statistics employed seem appropriate.

E. Conclusions: The conclusions seem to be well supported by the data presented.

F. Suggested improvements: Some of the figures are very compact and dense. For example, Fig. 1 deserves to be full page, or split. Paragraphs should be titled to help orient the reader. Overall, the paper is very verbose and lacks a forceful explanation of what is the big idea addressed. The Biological examples go in much detail at the expense of clarity and are difficult to digest. As a methods paper, it should focus a bit more on the method and be a bit more 'to-the-point' when it comes to the demonstrations in my opinion.

G. References: Might be a good thing to cite recent works on deep learning applied to image data of cells, the citation list is a bit light on the topic -- if only to contrast this approach and explain its difference not only in terms of method, but also in terms of the ingested data.

H. Clarity and context: As explained above, the manuscript would benefit from more explanation on the method, and a better 'situation' of that method in the context of preexisting methods (even if no method as ever been developed for this exact application, there are 'other' classical ways to do this). The examples can be shortened and made more 'punchy', some details can be pushed to supplements. Finally, a bit more of a systematic comparison with baseline(s) approaches would help show that this is not just 'yet another application of some deep learning model' to a new problem, but a real advance that is quantifiable.

Reviewer #2:

Remarks to the Author:

To address the challenges in quantifying the changes in subcellular imaging among perturbations, Spitzer, Berry et al. proposed a deep learning based framework named CAMPA for finding perturbation-independent landmarks. By training a cVAE on small patches (3x3) and directly providing the perturbation conditions to the encoder/decoder, the cVAE can encode image patches into perturbation-independent latent vectors. The latent vectors were used to perform clustering and manually assigned with biological-relevant labels for facilitating the interpretation of the results. After the method validation, CAMPA was applied in quantifying the changes in molecular composition, spatial organization and scale changes across different conditions and within populations.

By comparing to conventional direct pixel-based clustering, the condition-independent advantages of CAMPA were demonstrated. Importantly, the perturbation-independence property of CAMPA was validated by running UMAP clustering and classification on the generated latent vector.

Overall, the method design, validation and demonstrated applications are convincing, the drawbacks and potential improvements were discussed properly. The method can be used as a powerful tool for generating novel insights on subcellular changes in multiplexing imaging based screenings and is likely to be of broad interest to the community. However, there is a remaining concern regarding potential issues introduced in the manual annotation process of the identified clusters. I therefore suggest a revision on addressing these issues (see major comments and code review) before sharing the method with the community.

Major comments

Since CAMPA is an unsupervised approach, it relies on manual annotation and merging of the identified clusters, it may pose a major limitation in its applicability in a broader range of applications. Annotating the identified clusters is necessary to link to direct biological components, wrong or inaccurate annotation will mislead or create bias in the interpretation and further analysis. The manuscript provides validations in Supplementary Fig. 6 to justify the annotation, however, the validation of a subset of CSLs does not provide guarantee for the correctness of other unverified CSLs. As to my understanding, since there is no guarantee that each cluster will correspond to a biologically interpretable structure or component, the annotation of some markers can easily become challenging (e.g. the classes that were merged into nucleoplasm in Supplementary Figure 5) which can further cause issues in the interpretation.

More specifically, in Figure 2, the latent clusters were identified and were labeled with biologically meaningful labels such as Nucleolus and Nuclear speckles, and the author mentioned the criterion for the annotation were “the relative molecular abundance of each 4i-marker in each cluster as well as their spatial distributions within the cell”. However, the exact rules for the annotation were not clear. It will be helpful if the authors can clarify the following questions:

- In Figure 2a, the UMAP visualization of the latent vectors does not show clear separation between clusters. I am therefore wondering whether the leiden clustering will suffer from instability, at least for the samples that are near the border of two clusters.
- Figure 2c and Supplementary Figure 5 show that each cluster often corresponds to more than one marker, and in many cases, a spectrum of markers. What are the procedures to obtain the CSL labels?

Correct me if I am wrong but I think the manual annotation step can become a major source of bias in the CAMPA method (e.g. if an inaccurate name was annotated to a CSL), thus compromising the “unbiased” claim of the method. Having in mind that this might be a difficult issue to solve, I would at least tone down the claim about being unbiased, or if possible, provide measures for ensuring objective annotation of the CSL. Ideally, the authors should provide clear instructions for systematically assigning biologically meaningful labels to the identified clusters, as well as the criterion for merging and discarding clusters (if needed). If applicable, the users should be warned about the potential risk of mislabeling of clusters/landmarks.

The authors are encouraged to explore ways that enable automatic or semi-automatic annotation of the clusters, one quick thought is that maybe one can pull localization labels from the human protein atlas (<https://www.proteinatlas.org/>) for the contributing channels, and generate a composed labels or descriptions for each cluster.

Minor comments

Line 133-135: It is unclear why providing the condition labels will result in conditional-independent representation. Could you provide some references or mathematical clues for this? As also mentioned in discussion, it would be more clear if there is an adversarial loss that minimizes the condition predictive power from the latent vector.

Line 157: The sentence suggests there are 34 markers but the figure legend of Supplementary figure 1 said 43 markers. And also are there tables listing the exact 34 markers and 43 markers for the two datasets? It seems missing in the supplementary tables.

Line 168: Is there a benchmark on different patch sizes? Ideally, the authors should justify why 3x3 was chosen, does that depend on the pixel resolution of the image? Providing a description on how different patch sizes affect the results would also be helpful to understand the method.

Line 196-197: "we considered the relative molecular abundance of each 4i-marker in each cluster as well as their spatial distributions within the cell". As mentioned in the major comments, it would be more helpful to provide more details for how the labels were assigned to the clusters. Since the training of cVAE and clustering is completely unsupervised, are the features disentangled and always correspond to a certain biological label? Were there clusters hard to assign labels?

Line 199: Why manually merging the clusters?

Line 219-223: Is it possible to provide some more direct evidence to show that condition-dependent channels were ignored and not used? Since all the channels were fed into the cVAE for training, how can the user find out those condition-dependent channels? Does this mean that we can use this to find out the minimal set of antibodies for obtaining the same set of landmarks? This might be a very interesting direction to explore for reducing the cost of drug screening.

Line 224: It seems that Figure 2a was not mentioned in the main text.

Line 417: Is there evidence showing cell size change is the actual cause of subcellular reorganization? The following paragraph shows the cell size change was induced by reducing the expression of SBF2, but the siRNA treatment may also cause other changes (other than cell size) and lead to subcellular reorganization, right? If not, please provide references.

Line 424-427: To clarify, when applying CAMPA on the 43-channel dataset, is everything (training, clustering, annotation and merging) done from scratch? And also would it be possible to map and compare with the landmarks detected in the 34-channel dataset? Do the two datasets share channels?

Finally, it seems the cVAE should be trained specifically for each collection of immunofluorescence channels, it remains unclear how the antibody channels were selected. In large-scale cost-sensitive drug screening settings, it will be very useful to infer the minimal set of immunofluorescence labels to obtain the landmarks. Could you elaborate on how this could be implemented with CAMPA?

Code Review

The manuscript comes with a Python library (campa) which is a great way for increasing the adoption of the method. It is also appreciated that the software package was released under a permissive license and comes with proper documentation to ease the adoption of the software. I spent some time running the package but didn't manage to configure it properly thus cannot provide a thorough review on the code bases. As a general comment, it is clear that efforts have been made for other users to adopt the campa tool but it would be more helpful if the tutorials can be made more friendly for the first-time users.

There is also a dedicated code repository (campa_ana) for reproducing the results in the manuscript which seems complete, but there seems no data provided to enable the actual reproduction of the results. In the manuscript it mentioned that the data will be available upon request, so maybe at least provide more info on how to obtain all the data for readers who are interested in reproducing the results. Meanwhile, if possible, the authors are encouraged to use the open data repositories such as Zenodo and IDR to share the full set of their image data publicly – it will be a valuable resource for the community to evaluate campa and improve it further.

More specifically, the installation of the software through a pip command was easy to follow and the generated documentation about classes, API and CLI are helpful for the users to look up. The use of the campa package heavily depends on the correct configuration of the config file, and the most difficult part during my test is to construct a config file and dataset that works for campa despite the detailed instructions. Similarly, the tutorials were also written based on the assumption that users have already made the correct config file and have already organized data folders in a specific way.

Here are suggestions for making it easier for the first-time user to adopt campa:

- Provide a command to download a complete but minimal example dataset folder with the correct configuration and data. In the documentation, show a complete example of the config file along with a complete zip file with all the data files that work with the config file. (There seems to be a command already)
- The tutorial should be self-contained without too much assumption. Ideally, it should start by telling the user how to prepare the environment for the tutorial (e.g.: create a new conda environment and

install campa and other dependencies, clone the campa repo with notebooks, start Jupyter notebook, and navigate to the notebooks folder).

- All the tutorial notebook should be self-contained without assuming that the user has done the correct configuration. E.g. start by running a command to download the example dataset, then create the config automatically such that the user can just execute the cells one by one. Keep in mind that most users will quit easily if they need to make a lot of effort to get started with campa for the first time.

- It is always helpful to ask someone outside the team to do a blind test of the documentation and tutorials.

Wei Ouyang

Trang Le

Reviewer #3:

Remarks to the Author:

Spitzer and colleagues present a deep-learning method called CAMPA that aims to learn common states from multi-dimensional microscopy measurement across experimental conditions in order to explore the changes in cellular organization and molecular composition. The strategy utilizes a deep learning framework to learn condition independent molecular signatures that can then be compared across conditions. The stated goal is to move away from existing methods that utilize clustering across conditions and therefore may be more sensitive to misassignment if a perturbation leads to a major change in a specific feature. The authors use this approach to analyze a set of images for 44 protein markers across 6 chemical perturbations that disrupt Pol I transcription, Pol II initiation, Histone acetylation, mRNA splicing, and others. They use these 44-markers to define key nuclear structure states and describe several intriguing new observations about nuclear body dynamics upon distinct perturbations. For example, they found that nuclear speckles increase in size upon inhibition of mRNA splicing and that their molecular composition changes, corresponding to an increase in polyA binding protein localization and depletion of Pol II.

Overall, I find the paper, the dataset, and the results described to be quite interesting. Certainly, the dataset alone will be of great interest to many within the nuclear structure field and their observations upon these distinct perturbations add key new insights. Yet, I must admit that I did not fully grasp the importance of the CAMPA method or what specific need it was addressing. While the authors briefly motivate the method by highlighting that other methods learn parameters within samples and therefore perturbations could impact classification, it is not clear why a much simpler solution of simply leaving said variable out in the assignments would not address this very specific issue. Related to this, it is not clear to me that CAMPA was really needed to derive the biological insights highlighted in this paper.

How would the previous approaches perform on these same datasets? Based on the vectors shown, I would imagine just fine, but the authors do not describe this at all.

I do not mean to be a stickler here, since the paper is quite interesting (to me) even without the method. However, given that the paper was submitted to Nature Methods, I suspect the method is far more important than is obvious to me from the description. If this is indeed the case, the authors really should rewrite a bit to more clearly articulate the nature of the issue with existing methods and their limitations, what CAMPA enables and why this is specifically required for addressing the goals in the experimental datasets they presented. I would also encourage them to consider comparing the results obtained from CAMPA to alternative approaches on their dataset in order to highlight the nature of information you can derive using this approach. I think that the authors will find that this effort will make their method more useful and widely adopted.

Author Rebuttal to Initial comments

Point-by-point response to the reviewers' comments**Learning consistent subcellular landmarks to quantify changes in multiplexed protein maps**Hannah Spitzer^{1,*}, Scott Berry^{2,3,*}, Mark Donoghoe⁴, Lucas Pelkmans², Fabian J. Theis^{1,5,6}¹ Institute of Computational Biology, Helmholtz Center Munich, Germany.² Department of Molecular Life Sciences, University of Zurich, Zurich, Switzerland.³ EMBL Australia Node in Single Molecule Science, School of Medical Sciences, University of New South Wales, Australia.⁴ State Central, Mark Wainwright Analytical Centre, University of New South Wales, Australia.⁵ Department of Mathematics, Technical University of Munich, Germany.⁶ TUM School of Life Sciences Weihenstephan, Technical University of Munich, Germany.

In the following, we present our response to the reviewers comments. We state **reviewers' comments (black)**, **point-by-point answers (green)** to the questions and in parts **copy parts of the text or specific panels (blue)**, which directly correspond to comments or reference to them.

Editor comments:

Thank you for your letter asking us to reconsider our decision on your Article, "Learning consistent subcellular landmarks to quantify changes in multiplexed protein maps". After careful consideration we have decided that we are willing to consider a revised version of your manuscript that is updated as you've outlined. We ask that in your revision you place extra emphasis on clarifying the details requested by referee 2 to make it clear that the approach is unbiased.

Thank you for the chance to address the reviewers comments and revise our manuscript. This has given us the opportunity to further strengthen our claims, ensuring that CAMPAs will be a valuable contribution to the community in providing quantitative analysis of subcellular changes across conditions.

The main changes and additions to the manuscript in response to the reviews are as follows:

- We clarified the **purpose and procedure of the manual annotation step**, which does not impact the quantitative results of our analysis, but rather aids interpretability. To ensure that the criteria for the manual annotation are clear, we have added a description to the methods section, and supplementary figures to further explain our decision to merge several consistent subcellular landmarks (CSLs) and validate that this is appropriate. We have also added a tutorial to the CAMPAs documentation (comments 2.1 and 2.2). As proposed by reviewer #2, we have added an **automatic annotation** of CSLs with subcellular locations provided by the Human Protein Atlas to CAMPAs (comment 2.4).
- We have revised the manuscript to place a larger focus on the **unique contributions of CAMPAs, in particular the introduction of consistent subcellular landmarks, the quantitative differential analysis enabled by them and the downstream biological findings**. We now place CAMPAs more firmly in the context of previous literature (comment 1.G), and have redrafted the abstract, introduction and individual introductions to the biological results sections by putting a larger emphasis on CAMPAs unique contributions (comments 1.B and 1.F). We have also added an extensive analysis of our dataset using previous methods to show how insights that can be gained with CAMPAs are difficult to obtain with previous methods (comment 3.1). We have also restructured Figure 1 to show more clearly how CAMPAs differs from previous approaches, and have moved several figure panels from the main figures to the supplements to make figures less dense (comment 1.F).

- We have added **ablation studies** for better understanding of CAMPA. We now show that explicit use of conditioning is necessary for CAMPA and the effect of changing the neighbourhood size (comment 1.B). To deepen the reader's understanding of the cVAE model, we have extended the explanation of the cVAE model in the methods (comment 2.5) and use saliency scores to show what the model is focussing on when predicting individual channels (comment 2.10)
- To enable users to easily use CAMPA and reproduce results, we followed reviewer #2's suggestions and made **CAMPA's tutorials self-contained**, not requiring any specific config files, and will make the full dataset and analysis results publicly available via Zenodo upon acceptance of the manuscript.

We believe these changes greatly clarify the CAMPA workflow and make the advantages of CAMPA over previous methods clear, to make it a strong contribution to the field of subcellular multiplexed image analysis.

Reviewer #1 (Remarks to the Author):

A. Summary of the key results: In the manuscript "Learning consistent subcellular landmarks to quantify changes in multiplexed protein maps" the authors develop a computational framework to learn consistent molecular landmarks from highly multiplexed quantitative subcellular landmarks. The main challenge that the authors address is the confusion between differences in molecular profiles and their variation across space. To address this the authors introduce CAMP -- a conditional auto-encoder for multiplexed pixel analysis -- and then go on to demonstrate its utility with use cases.

Thank you for this accurate summary of CAMPA.

B. Originality and significance: The work is original in that there is not a lot of highly multiplexed data out there yet, and even less algorithms for that data. However, the manuscript could benefit from comparison to a better baseline. Comparing to naive pixel clustering is perhaps quite limited. Another question is what parts of the model used are important and for which reasons, an ablation study could be useful.

Thank you for raising the question of better baselines. As the reviewer mentions, there are very few algorithms for analysis of highly multiplexed imaging data. The goal of CAMPA is to identify consistent subcellular landmarks in an unsupervised manner and use these to quantify subcellular changes across conditions. As such, CAMPA allows qualitatively different analyses than previous methods developed for similar data. It thereby enables novel biological insights, and provides a general framework for how to do these analyses on future spatially resolved multiplexed datasets. After discussions across the involved labs, we have not been able to identify a better baseline than pixel clustering. Also, we are not aware of alternative existing methods that aim to achieve the same goal, and none of the reviewers have mentioned a comparable method. For example, MCMICRO¹ (a recently developed toolkit to package several analysis tools for multiplexed tissue images) provides only methods for cellular segmentation using nuclear stains, and does not leverage the multiplexed nature of images for segmentation. Naive (direct) pixel clustering is the method that has, so far, been used to analyse highly multiplexed images in an automated manner at the pixel-scale^{2,3}. This method therefore provides an appropriate comparison for this study. In response to reviewer comment 3.1, we have extended this comparison to show in detail how the direct pixel clustering performs on this dataset when the goal is to identify and quantify subcellular changes that occur in different perturbations (Supplementary Fig. 7, Supplementary Note 1, see response to reviewer #3, comment 3.1). We now also clearly distinguish the pixel clustering approach from CAMPA in the results:

[lines 270-275] (Results)

Unlike the direct pixel clustering approaches used previously^{2,3}, in which conditions are compared by identifying pixel classes that change abundance between conditions (Supplementary Fig 7, Supplementary Note 1), the CSL-based approach used in CAMPA compares molecular abundances across landmarks that are consistently found in both conditions (CSLs). This naturally extends traditional analysis of cellular abundance changes (Fig. 1d) to the subcellular scale.

As suggested in comment G, we have also now made changes to clearly contrast CAMPA from related deep-learning based approaches for segmentation and representation learning.

As suggested by the reviewer, we have now added ablation studies to further explore which parts of the model are important. We have evaluated the influence of the conditioning and the input patch size on the learned latent space. To summarise the results, the condition input to the VAE is necessary to obtain latent spaces that do not contain condition information (Figure 1g,i, see below). On the other hand, the 3x3 input patch size was optimal for reducing the susceptibility of the latent representation to single-pixel noise (Supplementary Fig. 2d, see below). To further interrogate the cVAE model, we also investigated saliencies showing that the condition input was preferentially used for condition-dependent channels (see response to reviewer #2, comment 2.10).

We summarise the results of the ablation studies in the first results section:
 [lines 180-186] (Results)

We also found that a VAE model without conditioning was not able to generate condition-independent latent spaces (median accuracy 0.72; min=0.51; max=0.97), validating the explicit use of conditioning in CAMPA (Fig. 1g, Supplementary Fig. 3a). Investigating the importance that the cVAE places on the condition input using integrated gradients⁴ revealed that for channels with perturbation-specific differences in intensity, the cVAE places increased importance on the condition input (as opposed to the latent representation) for modelling these channels (Supplementary Fig. 5d,e). We found that for our data, using a 3x3 neighbourhood of each pixel as cVAE input was optimal for improving cVAE latent space robustness to single-pixel noise which often occurs in microscopy imaging (Supplementary Fig. 2d), representing another advantage of using an autoencoder over direct pixel clustering.

We have added a comparison to a VAE to panels g and i in Figure 1:

Figure 1: CAMPA enables unsupervised learning of consistent subcellular landmarks using a conditional variational autoencoder.

[...] **g**: Comparison of perturbation-specificity of 4i pixel profiles, VAE, and cVAE latent space coordinates. Plots show balanced accuracy scores of binary logistic regression classifiers predicting perturbation from normalised 4i pixel profiles or latent representations of pixels. Accuracy values of 0.5 indicate random chance (perturbation information is not present in the data).

[...] **i**: Comparison of perturbation-specificity of direct clustering at different Leiden resolutions (0.2, 0.4, 0.6, 0.8, 1.2, 1.6, 2.0), VAE latent space clustering and cVAE latent space clustering / CSLs. Plots show coefficient of variation of the fraction of pixels in each perturbation assigned to each cluster. Boxplot summarises results for all clusters with resulting number of clusters n shown above. Center line, median; box limits, upper and lower quartiles; whiskers, 1.5x interquartile range; points, all data points.

Extended results for the VAE are presented in Supplementary Fig. 3a:

Supplementary Figure 3: Pixel clustering and VAE latent space clustering are perturbation-dependent at different Leiden clustering resolutions.

a: Left: UMAP representation of VAE latent space colored by perturbation. Middle: Example cells from each perturbation colored by VAE latent space clustering. Right: Fraction of pixels assigned to each cluster per perturbation colored by VAE latent space clustering. [...]

We have extended Supplementary Fig. 3d to show the results of noise-robustness analysis with input patch sizes from 1x1 to 7x7:

Supplementary Figure 2: Robustness to noise and randomness in subsampling and clustering using the 184A1 dataset.

[...] d: Robustness of cVAE latent space when adding Gaussian noise to the input. Scale is the standard deviation of the Gaussian distribution. Shown is the MSE and 95% confidence interval of pairwise comparisons with 5 different random seeds. cVAE latent spaces obtained by training with a local 1x1 (no neighbours), 3x3, 5x5, and 7x7 neighbourhood as context for every molecular profile are compared. Using a neighbourhood of 3x3 adds to increased stability of the latent space in the presence of single pixel noise. Using larger neighbourhoods reduces those benefits, as the model is more dependent upon specific configurations of neighbouring pixels for the generation of the latent space representation.

C. Data & methodology: The approach is valid, the data appears to be of high quality, and the presentation of the method, data and data is high.

Thank you.

D. Appropriate use of statistics and treatment of uncertainties: Statistics employed seem appropriate.

Thank you.

E. Conclusions: The conclusions seem to be well supported by the data presented.

Thank you.

F. Suggested improvements: Some of the figures are very compact and dense. For example, Fig. 1 deserves to be full page, or split. Paragraphs should be titled to help orient the reader. Overall, the paper is very verbose and lacks a forceful explanation of what is the big idea addressed. The Biological examples go in much detail at the expense of clarity and are difficult to digest. As a methods paper, it should focus a bit more on the method and be a bit more 'to-the-point' when it comes to the demonstrations in my opinion.

Thank you for the valuable suggestions about how to improve our manuscript. We have made several changes detailed below to emphasise the major contributions as well as the proposed method and to simplify the manuscript.

This paper describes the first method that allows automated identification of consistent subcellular structures across perturbation conditions and/or heterogeneous cell populations. We have edited the abstract to emphasise this more clearly, revised the schematic visualisation of our method in Fig. 1a-c to contrast it from previous methods more clearly (see response to reviewer #3, comment 3.1), and included and extended the comparison to previous methods (which may not be widely known; see response to reviewer #3, comment 3.1):

[lines 20 - 33] (Abstract)

Highly multiplexed imaging holds enormous promise for understanding how spatial context shapes the activity of the genome and its products at multiple length scales. Here, we introduce a deep-learning framework called CAMPAs (Conditional Autoencoder for Multiplexed Pixel Analysis), which uses a conditional variational autoencoder to learn representations of molecular pixel-profiles that are consistent across heterogeneous cell populations and experimental perturbations. Clustering these pixel-level representations identifies consistent subcellular landmarks, which can be quantitatively compared in terms of their sizes, shapes, molecular compositions, and relative spatial organisation. Using high-resolution multiplexed immunofluorescence, this reveals how subcellular organisation changes upon perturbation of RNA production, RNA processing, or cell size, and uncovers links between the molecular composition of membraneless organelles and cell-to-cell variability in bulk RNA synthesis rates. By capturing interpretable cellular phenotypes, we anticipate that CAMPAs will greatly accelerate the systematic mapping of multiscale atlases of biological organisation to identify the rules by which context shapes physiology and disease.

Since reviewers #2 and #3 appreciated the biological impact, we would prefer to not remove particular biological results. However, as suggested, we have sharpened the overall focus of the manuscript on the method and its capabilities by:

- Focussing the last paragraph of the introduction on CAMPAs unique contributions and novel findings that can be gained using CAMPAs (lines 83-94, see below)
- Clearly contrasting CAMPAs to previous work and other related approaches (see point G)
- Editing the biological results sections with focus on clarity, in particular by
 - Changing the paragraph titles to clarify CAMPAs contribution to the results (see below).
 - Ensuring that each biological results section begins with a clear statement of how CAMPAs is used for these particular results
 - Ensuring that each biological results section ends with a summary of CAMPAs contributions to the presented results
 - Shortening the description of the biological experiments (described in detail in the methods)
 - Removing excessive qualitative descriptions of the results and rather focussing on quantitative descriptions
- Revising and shortening all sections for clarity

In order to make the figures less dense, we have:

- Made Figure 1 full-page as suggested, and summarised the size-variation plots of the clusters in panels e and f to pie-charts.
- Moved Figure 3, panel a to Supplementary Figure 8 panel b.
- Removed Figure 4, panel g, and shortened the axis labels of the confusion matrices.
- Moved Figure 6, panels b,c, and e to Supplementary Figure 16 panels a-c.

[lines 83 - 94] (Introduction)

Here, we use CAMPA to derive a detailed map of subnuclear organisation across different perturbations, directly from high-resolution iterative indirect immunofluorescence imaging (4i)² data. This reveals how key proteins and protein states (e.g. phospho-proteins and histone post-translational modifications) involved in transcription, chromatin, mRNA processing and nuclear export, as well as subnuclear organelles change at the cellular and subcellular scale upon perturbation of different stages of mRNA metabolism. We find that the three aspects of cellular phenotypic information captured by CAMPA (cellular intensities, subcellular protein localisations, and subcellular spatial organisation) all contribute unique information to characterise perturbations, indicating that CAMPA will be a powerful approach for cellular phenotypic screening. Finally, by capturing and quantifying interpretable cellular phenotypes at multiple scales, we demonstrate that the combination of 4i and CAMPA can uncover quantitative relationships across scales - from cell populations to subcellular organelles.

To help readers orient themselves in the paper, we already use subsection headings and have improved them as mentioned above to clearly state CAMPAs contribution to the results described in this section. Results sections are now titled:

- CAMPA enables identification of consistent subcellular landmarks across conditions
- CAMPA uncovers perturbation-induced changes in molecular composition and spatial organisation of subcellular landmarks
- CAMPA-derived cell features allow comparisons of spatial and molecular differences across multiple perturbations
- CAMPA reveals subcellular reorganisation upon change in cell size
- Linking cellular heterogeneity and subcellular reorganisation using CAMPA

This will help readers who are not interested in all biological details to find the appropriate sections more easily.

G. References: Might be a good thing to cite recent works on deep learning applied to image data of cells, the citation list is a bit light on the topic – if only to contrast this approach and explain its difference not only in terms of method, but also in terms of the ingested data.

Thank you for this useful suggestion. We agree that it is helpful to place CAMPA more firmly in the context of previous works. We added a short discussion to the introduction on deep-learning based methods for segmenting cells and organelles, and learning cellular representations based on multiplexed imaging to highlight how CAMPA differs from these approaches:

[lines 61 - 70] (Introduction)

Recently, deep learning based segmentation models were used to segment cells and nuclei from multi-channel fluorescence microscopy images^{5,6}. However, using these supervised methods to generate consistent segmentations of subcellular structures would require annotated training data from all conditions. Self-supervised approaches for clustering proteins according to their subcellular localisation patterns alleviates the need for this time-consuming manual labelling^{7,8}, however these do not account for changing protein localisations across perturbations nor do they allow quantification of these changes. To facilitate high-throughput quantitative analysis of subcellular organisation, we therefore need approaches that can reveal consistent subcellular landmarks despite condition-dependent, and possibly unanticipated, changes to protein abundance and/or relative localisation.

H. Clarity and context: As explained above, the manuscript would benefit from more explanation on the method, and a better 'situation' of that method in the context of preexisting methods (even if no method as ever been developed for this exact application, there are 'other' classical ways to do this). The examples can be shortened and made more 'punchy', some details can be pushed to supplements. Finally, a bit more of a systematic comparison with baseline(s) approaches would help show that this is not just 'yet another application of some deep learning model' to a new problem, but a real advance that is quantifiable.

Thank you for these thoughtful suggestions. As detailed in our response to comments B and F, we have followed these suggestions to make our manuscript more clear and place it more firmly in the context of existing methods.

Reviewer #2 (Remarks to the Author):

To address the challenges in quantifying the changes in subcellular imaging among perturbations, Spitzer, Berry et al. proposed a deep learning based framework named CAMPA for finding perturbation-independent landmarks. By training a cVAE on small patches (3x3) and directly providing the perturbation conditions to the

encoder/decoder, the cVAE can encode image patches into perturbation-independent latent vectors. The latent vectors were used to perform clustering and manually assigned with biological-relevant labels for facilitating the interpretation of the results. After the method validation, CAMPA was applied in quantifying the changes in molecular composition, spatial organization and scale changes across different conditions and within populations.

By comparing to conventional direct pixel-based clustering, the condition-independent advantages of CAMPA were demonstrated. Importantly, the perturbation-independence property of CAMPA was validated by running UMAP clustering and classification on the generated latent vector.

Overall, the method design, validation and demonstrated applications are convincing, the drawbacks and potential improvements were discussed properly. The method can be used as a powerful tool for generating novel insights on subcellular changes in multiplexing imaging based screenings and is likely to be of broad interest to the community. However, there is a remaining concern regarding potential issues introduced in the manual annotation process of the identified clusters. I therefore suggest a revision on addressing these issues (see major comments and code review) before sharing the method with the community.

Thank you very much for this positive evaluation. We address the raised concerns about the manual annotation step below.

Major comments:

2.1 Since CAMPA is an unsupervised approach, it relies on manual annotation and merging of the identified clusters, it may pose a major limitation in its applicability in a broader range of applications. Annotating the identified clusters is necessary to link to direct biological components, wrong or inaccurate annotation will mislead or create bias in the interpretation and further analysis. The manuscript provides validations in Supplementary Fig. 6 to justify the annotation, however, the validation of a subset of CSLs does not provide guarantee for the correctness of other unverified CSLs. As to my understanding, since there is no guarantee that each cluster will correspond to a biologically interpretable structure or component, the annotation of some markers can easily become challenging (e.g. the classes that were merged into nucleoplasm in Supplementary Figure 5) which can further cause issues in the interpretation.

Thank you for raising potential issues of the manual annotation step and making us aware that we need to be more clear about describing this annotation process. First, it is important to note that the annotation and CSL merging step is entirely optional for the analysis we present in the manuscript. None of the specific biological insights we highlight in the manuscript are affected by the annotation and merging of CSLs, but interpretability of the results is clearly aided by us giving names to clusters.

In general, CAMPA produces clusters that correspond to a group of pixel locations that have a common intensity profile while accounting for condition-dependent differences between cells. Therefore, each cluster can be seen as representing a specific structure or component of the cell, and we can use the overall intensity profile, shape, size, and spatial location of the structures to compare subcellular structure of cells in different conditions.

We introduce the manual annotation and merging of CSLs to aid interpretability of the results of these comparisons. It is correct that there is no guarantee that each cluster will correspond to a biologically interpretable structure or component, and that incorrect annotations could lead to mis-interpretation of the results. However, we would like to highlight that the analysis of CSLs that we present in the manuscript is not biased, as it is based only on CSLs or merged CSLs, and therefore on the unsupervised clusters CAMPA produces. This is conceptually similar to typical single-cell RNA sequencing analyses where cell-type clusters are discovered in an unbiased manner and subsequently annotated (either semi-automatically using atlases or manually)⁹. It is also conceptually similar to recent deep-learning algorithms for subcellular protein localisations, which rely on annotation after clustering, by applying independent manual annotations⁷ to allow interpretation. In addition, our annotation will not introduce any bias that hinders the comparison of cells across perturbations or other conditions, because all conditions are processed simultaneously using the same model.

We would also like to note that since CSLs as well as canonical organelles represent structures within cells with unique molecular profiles that are present in cells across all conditions, it is very likely that there is a large overlap between these two concepts. If the canonical markers for these structures are present in the antibody panel, we do not expect annotation of most CSLs to biologically meaningful structures to pose a major challenge. However,

when applying CAMPA to even higher-resolution data in the future, or when identifying novel organelles (or different types of the same organelle), this annotation step may not be required or desirable.

To clarify the purpose of the annotation step, we added the following to the first results paragraph:

[lines 217-226] (Results)

To facilitate biological interpretability, we annotated CSLs with the names of known subcellular structures (see methods) (Fig. 2a). This is an optional step in the CAMPA workflow, and none of the specific biological insights that we will present in the following depends on this. The annotation resulted in assignment of the ten original CSLs to seven annotated CSLs (Nucleolus, Nuclear speckles, PML bodies, Cajal bodies, Nucleoplasm, Nuclear periphery, and Extra-nuclear (outside the nucleus)) (Fig. 2d-i), by merging four original CSLs to the Nucleoplasm CSL (Supplementary Fig. 5a). These CSL annotations are consistent with automatic annotation proposals obtained by querying the Human Protein Atlas (HPA)^{10,11} database (Supplementary Fig. 5b). In the following we refer to these annotated CSLs simply as CSLs.

For more details about the CSL annotation and merging process, see our response to comment 2.2b.

As suggested in comment 2.4, we have introduced automatic CSL label proposals by querying the Human Protein Atlas. This is described in our response to comment 2.4.

2.2 More specifically, in Figure 2, the latent clusters were identified and were labeled with biologically meaningful labels such as Nucleolus and Nuclear speckles, and the author mentioned the criterion for the annotation were "the relative molecular abundance of each 4i-marker in each cluster as well as their spatial distributions within the cell". However, the exact rules for the annotation were not clear. It will be helpful if the authors can clarify the following questions:

2.2a - In Figure 2a, the UMAP visualization of the latent vectors does not show clear separation between clusters. I am therefore wondering whether the leiden clustering will suffer from instability, at least for the samples that are near the border of two clusters.

It is correct that the leiden clustering is slightly unstable in the regions of the feature space that are near the border of two clusters. We included a detailed evaluation of this instability in Supplementary Fig. 2a,b in our original submission. This mainly seems to impact the nucleoplasm, nuclear periphery, and Cajal body clusters for the 184A1 cells. Biologically this means that these three structures are not especially well defined with the antibody panel that we used for these experiments. In fact, this cluster instability information could be used to refine the antibody panel for future experiments. For example, the analysis in this case suggests that the inclusion of additional Cajal body markers, or nuclear lamina markers (for the nuclear periphery) in the antibody panel would be advantageous. We added a sentence about this to the methods:

[lines 871-873] (Methods)

This cluster instability analysis could be used to refine the antibody panel for future experiments, by indicating those CSLs for which additional channels might be needed.

Note that the instability of leiden clusters is not unique to CAMPA, but similar issues occur in current state-of-the-art direct pixel clustering^{2,3} and even in segmentation approaches based on a single markers¹². The underlying issues are: i) that protein intensities have a continuous distribution in space; ii) that their levels vary between cells; and iii) that microscope resolution is limited by diffraction, so even point-sources lead to a distribution of intensities spanning several pixels. With these three features of the data, it is therefore not obvious where to set the threshold when grouping pixels. In manual pixel-clustering approaches, it is left to the human who is training the classifier to define which pixels do and do not comprise a certain structure. When comparing CAMPA results with supervised segmentation of two well-defined organelles trained by an expert using single input channels, we saw a high correspondence (Supplementary Fig. 6).

In general, as mentioned in the discussion, the concept of grouping pixels in discrete clusters might be suboptimal, and approaches based on mixture models and fuzzy clustering that do not need a discrete assignment of pixels to clusters could be explored in the future.

Despite these difficulties, the CSLs that our method produces have the advantage that they can be used to quantitatively compare data across conditions, even if they might be slightly different across multiple CAMPA runs. Therefore the instability of the leiden clustering does not hamper unbiased comparison across conditions.

We edited the first results section to include this explanation:

[lines 194-198] (Results)

Cluster stability was not significantly influenced by a different random subsample nor by reducing or increasing the number of samples used for the clustering by a factor of two (Supplementary Fig. 2a-b). Because all conditions are considered together, any cluster instability does not impact the ability to quantitatively compare cells across conditions.

2.2b - Figure 2c and Supplementary Figure 5 show that each cluster often corresponds to more than one marker, and in many cases, a spectrum of markers. What are the procedures to obtain the CSL labels?

Clusters are annotated based on an iterative manual process by labelling clusters based on organelle markers and visualising the annotations on the multiplexed images. Following the reviewers suggestion in comment 2.4, we have also included confirmation of annotations using subcellular localisations proposed by the Human Protein Atlas; see our response to comment 2.4 for more information. For the purposes of clarity, we merge those CSLs that correspond to the same biological structure, as for this particular analysis we are not interested in analysing the identified sub-parcellations of known organelles.

We have added a section to the methods to clarify the CSL annotation and merging procedure:

[lines 883-898] (Methods)

Annotation

To aid interpretability, we manually annotated CSLs with biologically meaningful labels. CSLs corresponding to the same biological structure may be merged to the same annotated CSL. This annotation was done in an iterative fashion and considered the following factors:

- Presence of canonical organelle markers among top enriched channels in each CSL in unperturbed (control) cells. If no canonical markers were present, consider CSL as "background" (i.e. nucleoplasm / cytoplasm).
- Spatial distribution of CSLs compared to spatial distribution of canonical markers of organelles in unperturbed (control) cells
- Human Protein Atlas subcellular localisation^{10,11} of most enriched channels in each CSL, weighted by z-scored channel intensity.

To simplify the presentation of results, we merged those CSLs that with the above criteria correspond to the same biological structure. The merged CSLs either corresponded to the same structure that displayed within-condition variation (e.g. Nucleoplasm CSLs in Fig. 2, see Supplementary Fig. 5c) or to different spatial locations of the same biological structure (e.g. P-body CSL in Fig. 5, see Supplementary Fig. 12b).

We added a panel to Supplementary Figures 5 and 12 each to illustrate the behaviour of the merged CSLs:

Supplementary Figure 5: Details of cVAE clustering (184A1 cells).

[...] c: Coefficient of variation (CV) of CSL sizes per well. Nucleoplasm clusters have a large CV, and are merged to one annotated CSL (shown in a). [...]

Supplementary Figure 12: Details of cVAE clustering (HeLa cells, scrambled siRNA).

[...] b: Cluster 13 and 19, which are merged into the "P-bodies" cluster, and DDX6 visualised in two example cells from scrambled siRNA and SBF2 siRNA. Cluster 19 corresponds to P-body center with very high DDX6 intensity, cluster 13 corresponds to P-body periphery with lower DDX6 intensity. Both P-body clusters show identical behaviour when comparing scrambled to SBF2 (Supplementary Fig. 13b,d) [...]

We also added two panels to Supplementary Figure 13 showing that the two clusters grouped to the P-body cluster show identical behaviour when comparing scrambled to SBF2:

Supplementary Figure 13: Quantification of mean intensity changes of each channel in each CSL in HeLa cells upon SBF2-knockdown

a: Log₂ fold-change of mean intensities for each channel in each CSL, or number of pixels in each CSL, when comparing *SBF2*-knockdown with scrambled siRNA control cells. P-values show significance of *SBF2*-knockdown on channel intensities for each channel/CSL, as determined from the mixed effect model. P-values are corrected for multiple hypothesis testing using the Benjamini-Yuketeli method.

b: As in a, for the non-merged CSLs corresponding to P-bodies.

c: As in a, except normalised by the whole nucleus or whole cytoplasm (not the "Cytoplasm" CSL) changes in intensity, respectively for the upper and lower panels. In this case p-values indicate significance of mean intensity change in CSL compared to the change observed for the nucleus or cytoplasm, respectively (see Methods).

d: As in c, for the non-merged CSLs corresponding to P-bodies.

e: Sum intensity of COIL in the nucleus or Cajal bodies of scrambled and SBF2-knockdown HeLa cells. Boxplots summarise distributions across cells. [...]

Finally, we have extended the tutorial on clustering in CAMPAs documentation to show how annotation works on a practical example: <https://campa.readthedocs.io/en/latest/notebooks/cluster.html>

2.3 Correct me if I am wrong but I think the manual annotation step can become a major source of bias in the CAMPAs method (e.g. if an inaccurate name was annotated to a CSL), thus compromising the “unbiased” claim of the method. Having in mind that this might be a difficult issue to solve, I would at least tone down the claim about being unbiased, or if possible, provide measures for ensuring objective annotation of the CSL. Ideally, the authors should provide clear instructions for systematically assigning biologically meaningful labels to the identified clusters, as well as the criterion for merging and discarding clusters (if needed). If applicable, the users should be warned about the potential risk of mislabeling of clusters/landmarks.

As detailed in our response to comment 2.1, the annotation step is optional, and is not necessary to achieve the quantitative results we present in the manuscript. Therefore we would like to emphasise that the method to generate CSLs that we introduce in this manuscript is unbiased.

As described in the response to comment 2.2b, we have added a methods section to clarify the annotation procedure and extended the clustering tutorial (<https://campa.readthedocs.io/en/latest/notebooks/cluster.html>) to make it clear that annotation is optional and that care should be taken during label assignment.

While any manual annotation method is subject to inaccuracies, the ability to define CSLs using multiple specific markers will help to avoid these errors. Note that many of the markers used in our experiments are the canonical markers of these structures used routinely in cell biology, which facilitates annotation greatly. The new automated annotation functionality (see comment 2.4) will also assist to make annotation more objective.

However, we agree with the reviewer that we should not make any claims about the annotation being completely unbiased, as no manual annotation is ever completely unbiased. Therefore, we have toned down the following two claims about our entire method (including annotation) being unbiased:

[lines 237-238] (Results section 1)

We therefore conclude that CAMPAs allows consistent identification and annotation of subcellular landmarks across perturbations and cell cycle stages.

[lines 396-398] (End of results section 4)

Nonetheless, it [the analysis] shows that CAMPAs can rapidly identify that the nucleolus is the primary site of activity of this compound, despite the antibody panel not having a marker for the directly targeted protein (RNA Polymerase I).

2.4 The authors are encouraged to explore ways that enable automatic or semi-automatic annotation of the clusters, one quick thought is that maybe one can pull localization labels from the human protein atlas (<https://www.proteinatlas.org/>) for the contributing channels, and generate a composed labels or descriptions for each cluster.

Thank you for this interesting and very useful suggestion. We have added such a semi-automatic annotation functionality to CAMPAs. This function uses the most enriched channels (top three channels with highest z-scored intensity) for each cluster to query the Human Protein Atlas API for subcellular location labels. This will allow users to form initial hypotheses regarding the correct annotation of each cluster, and also to validate previous manual annotations. The annotations currently available on the Human Protein Atlas are quite limited. For example, there is just one annotation for “nuclear bodies” which encompasses several distinct subnuclear organelles (in our case both Cajal bodies and PML bodies) so our manual annotations are higher resolution than those in the HPA. We anticipate that additional validated atlases of subcellular protein localisation will become available over the coming years, and this annotation functionality in CAMPAs could be expanded to encompass these. We have verified that our manual annotations are consistent with the HPA subcellular location labels and now mention this in the manuscript:

[lines 223-225] (Results, annotation of nuclear CSLs from 184A1 cells)

These CSL annotations are consistent with automatic annotation proposals obtained by querying the Human Protein Atlas (HPA)^{10,11} database (Supplementary Fig. 5b).

[lines 452-453] (Results, annotation of cellular CSLs from HeLa cells)

Again, our manual annotation is consistent with the automated annotation, but is more detailed (Supplementary Fig. 12c).

In addition, we extended Supplementary Figures 5 and 12 with the HPA annotation proposals for the two clusterings presented in the manuscript:

Supplementary Figure 5: Details of cVAE clustering (184A1 cells).

[...] **b:** Automated annotations from the Human Protein Atlas. For each CSL a maximum of 3 channels with a z-score > 1 were used to query HPA for subcellular localisation of these channels. Shown is the result of these queries for each CSL, weighted by the mean z-score of the channel (shown in a). [...]

Supplementary Figure 12: Details of cVAE clustering (HeLa cells, scrambled siRNA).

[...] **c:** Automated annotations from the Human Protein Atlas (HPA). For each CSL a maximum of 3 channels with a z-score > 1 were used to query HPA for subcellular localisation of these channels. Shown is the result of these queries for each CSL, weighted by the mean z-score of the channel (shown in a). [...]

Finally, we extended CAMPAs tutorial on clustering

(<https://campa.readthedocs.io/en/latest/notebooks/cluster.html>) to showcase this functionality in more detail.

Minor comments:

2.5 Line 133-135: It is unclear why providing the condition labels will result in conditional-independent representation. Could you provide some references or mathematical clues for this? As also mentioned in discussion, it would be more clear if there is an adversarial loss that minimizes the condition predictive power from the latent vector.

Conditional VAEs were employed for similar purposes for batch-correction of scRNAseq data, where a cVAE was used to learn a batch-independent latent space from scRNAseq data and batch labels as input¹³. Briefly, the intuition for these models is as follows: During training, the model learns the probabilistic encoder $q(z|x,c)$ for latent space z conditioned on input x and condition c . This distribution is pushed towards $p(z)$ using the KL divergence. The jointly optimised probabilistic decoder $p(x|z,c)$ models pixel profiles x conditioned on latent space z and condition c . Therefore it is not necessary for z to contain information about condition c . As $q(z|x,c)$ only approximates $p(z)$, there is no guarantee that the learned latent space z will not contain information about c , however the training objective clearly pushes the probabilistic encoder in this direction. As mentioned in the discussion, additional adversarial losses could be employed to make the removal of condition information from the latent space more explicit, however for the data used in this manuscript we found that a simple cVAE worked well to remove condition information from the latent space (Fig. 1f). Also note that a VAE without conditioning was not successful in removing condition information, showing that the use of a cVAE here was necessary (see response to reviewer #1, point B).

We have included a more detailed description of the cVAE framework in the methods and added a short summary of the above explanation to the main text:

[lines 141-145] (Results)

Because the cVAE model learns a conditional generative distribution $p_\theta(x|z,c)$ for the pixel profiles, the model is optimised to encode variation like subcellular differences in intensity that occur across all conditions (and omit condition-specific information) in the latent representation z ^{13,14}, which results in less condition-dependent clustering of z (cf. Fig. 1g,i).

[lines 801-825] (Methods)

The cVAE models the pixel profiles as samples generated by a generative conditional distribution $p_\theta(x|z,c)$ (also named the probabilistic decoder), where z is a latent variable generated from a prior distribution $p_\theta(z|c)$, and c are the condition labels (e.g. perturbation and cell cycle state of the cell that the current pixel profile is coming from). For a given x , the latent variable z is inferred using a probabilistic encoder $q_\phi(z|x,c)$, which approximates the intractable true posterior $p_\theta(z|x,c)$. Using variational inference, parameters θ and ϕ are jointly tuned by maximising the evidence lower bound of the marginal log-likelihood $\log(p_\theta(x|c))$ ^{14,15}:

$$L(x, c; \theta, \phi) = E_{q_\phi(z|x,c)}[\log p_\theta(x|z,c)] - D_{KL}(q_\phi(z|x,c) || p_\theta(z|c)) \leq \log p_\theta(x|c)$$

With this formulation, the pixel profiles x are modelled by latent distribution z and condition labels, which encourages the model to encode non-condition specific variation like subcellular differences in intensity that occur across all conditions in the latent distribution.

To improve training stability and samples from the decoder, we use σ -VAE¹⁶ to learn the variance of the decoder resulting in calibrated decoders:

$$L(x, c; \theta, \phi) = D \ln \sigma + \frac{D}{2\sigma^2} \text{MSE}(\hat{x}, x) + D_{KL}(q_\phi(z|x,c) || p_\theta(z|c))$$

with $x \in R^D$ the centre pixel of the input, $\hat{x} = p_\theta(x|z,c)$ the VAE reconstruction of x . We use the analytical solution for the variance¹⁶ which minimises the (weighted) MSE loss while also minimising the logarithm of the variance:

$$\sigma^{*2} = \text{MSE}(x, \mu)$$

where μ is the estimated latent mean for x . As prior distribution $p_\theta(z|c)$ we choose:

$$p_\theta(z|c) = p(z) = N(0, 1).$$

2.6 Line 157: The sentence suggests there are 34 markers but the figure legend of Supplementary figure 1 said 43 markers. And also are there tables listing the exact 34 markers and 43 markers for the two datasets? It seems missing in the supplementary tables.

Thank you for catching that. 34 of the 43 channels are nuclear-localised, so these were used for the first set of analyses focused on the nucleus (Figures 1-4). All 43 channels were used from Figure 5 onwards. We have edited the dataset description in the methods to clarify which channels were used for which evaluations:

[lines 780-789] (Methods)

The first dataset consisted of 184A1 cells across six chemical treatments (Supplementary Table 3), using 34 channels localising (at least partially) to the nucleus (ALYREF, CCNT1, CDK7, CDK9, COIL, DAPI, GTF2B, H2B, H3, H3K27ac, H3K4me3, HDAC3, KPNA1, KPNA2, NCL, NONO, PABPC1, PABPN1, pCDK9, PCNA, PML, POLR2A, POLR2A-S2P, POLR2A-S5P, pRB1, RPS6, SETD1A, Sm antigen, SON, SP100, SRRM2, SRSF2, U2SNRNPB, YAP1). [...] The second dataset consisted of control and SBF2-knockdown HeLa cells (Supplementary Table 5), using 43 channels (including all those used in the first dataset together with pRPS6, pMAPK1, CALR, CTNNB1, PXN, HSPD1, GOLGA2, TUBA1A, DDX6).

2.7 Line 168: Is there a benchmark on different patch sizes? Ideally, the authors should justify why 3x3 was choosing, does that depend on the pixel resolution of the image? Providing a description on how different patch sizes affect the results would also be helpful to understand the method.

Thank you for this suggestion. We use a small local neighbourhood as input to reduce the dependency of the latent representation to single pixel noise. To show the impact of different patch sizes for this purpose, we extended Supplementary Fig. 2d (see response to reviewer #1, comment B) to show the impact of noise on latent representations learned from different patch sizes of 1x1 - 7x7. This shows that a patch size of 3x3 was ideal for reducing the dependency of noise. See our response to reviewer #1, comment B for more details and the changes that we introduced.

In general, for our specific data it is advisable to use relatively small patch sizes to avoid overfitting on specific neighbourhoods of pixels. However, if the data contained larger scale noise, or for example registration errors between individual fluorescence channels, a larger neighbourhood might be advisable to ensure that all relevant information is captured. In this case, more training data might be required as well.

2.8 Line 196-197: "we considered the relative molecular abundance of each 4i-marker in each cluster as well as their spatial distributions within the cell". As mentioned in the major comments, it would be more helpful to provide more details for how the labels were assigned to the clusters. Since the training of cVAE and clustering is completely unsupervised, are the features disentangled and always correspond to a certain biological label? Were there clusters hard to assign labels?

See our response to comment 2.1 and 2.2b for a detailed explanation of the purpose and workflow of the annotation step. In the datasets generated here, clusters were very easy to identify because the antibody panel contains well-characterised antibodies corresponding to known structural components of the cell. These were most often the highest-enriched channel in the "loading matrix" (e.g. Fig. 2c). In general, where the subcellular organisation being investigated is less well characterised, assigning labels will be more challenging and CAMPA users may want to label none, or only some, of the CSLs, leaving the unlabeled CSLs simply as cluster numbers from the Leiden algorithm.

2.9 Line 199: Why manually merging the clusters?

Briefly, for the purposes of clarity we manually merge some clusters that represent the same subcellular structure or organelle (as determined by our manual annotation). These clusters are either heterogeneous between cells within a condition, or represent different spatial locations of the same structure (i.e. areas with higher/lower abundance of the same markers). As shown in Supplementary Figure 5 and Supplementary Figure 12, these were predominantly the nucleoplasm and cytoplasm clusters, which lack specific markers, and represent a 'background' cluster. We also merged the two P-body clusters into one CSL. These corresponded to very high-intensity central P-body pixels and peripheral P-body pixels, respectively, as shown in Supplementary Fig. 12b.

See our response to comment 2.2b for an explanation of the CSL annotation and merging step and our changes to the manuscript to clarify this procedure.

2.10 Line 219-223: Is it possible to provide some more direct evidence to show that condition-dependent channels were ignored and not used? Since all the channels were fed into the cVAE for training, how can the user find out those condition-dependent channels? Does this mean that we can use this to find out the minimal set of antibodies for obtaining the same set of landmarks? This might be a very interesting direction to explore for reducing the cost of drug screening.

Thank you for the comments.

First, we would like to clarify that the initial phrasing of the sentence "Importantly, channels that show characteristic changes in intensity between conditions are ignored by the cVAE when generating the latent space [...]" was overly simplistic. In many cases the cVAE will not simply 'ignore' these perturbation-dependent channels, but might, for example, use both the channel intensity and the condition label to remove the condition-specific effect on the intensity. The result is that the learned latent representation does not contain condition-specific information, but other information like subcellular differences in intensity. We have edited the sentence to reflect this:

[line 245-247] (Results)

Importantly, the cVAE learns to remove condition-specific information from channels that show characteristic changes in intensity between conditions when generating the latent space and the CSLs.

A direct way of showing that conditional information is removed in the latent representation is presented in Fig. 1f which reports that latent representation has low predictive power for condition (close to that of random chance). The condition-dependent channels can be seen in the dot plots presented throughout in the manuscript (Fig. 1d, Fig. 3a, Supplementary Fig. 13).

To provide an additional way of investigating which channels depend on the condition input, we have now calculated saliency scores using integrated gradients. With this analysis we can directly see the importance that the cVAE assigns to the condition input for reconstructing every output channel. We added a short description of the results of this analysis to the first results section, and two new panels to Supplementary Fig. 5:

[line 183-186] (Results)

Investigating the importance that the cVAE places on the condition input using integrated gradients⁴ revealed that for channels with perturbation-specific differences in intensity, the cVAE places increased importance on the condition input (as opposed to the latent representation) for modelling these channels (Supplementary Fig. 5d,e).

Supplementary Figure 5: Details of cVAE clustering (184A1 cells).

[...] **d:** Importance of condition input for reconstructing each channel. Saliency scores are calculated with integrated gradients⁴ for every output channel with respect to the latent representation and condition input. The importance of the condition for each output channel is calculated as the fraction of absolute scores for the condition normalised by the absolute sum of scores for the entire input (condition and latent representation). Shown is the maximum condition importance of each channel, the median condition importance for each perturbation and channel, and the median condition importance for each cell cycle stage and channel. Large values indicate that the condition information was important for correctly reconstructing this channel, and the cVAE focused less on the latent representation.

e: Scatter plot of condition importance (shown in d) vs log₂fc in overall cellular intensity for each channel and perturbation (shown in Fig. 1d). Channels that show an overall perturbation-dependent effect in intensity also have high condition importances, meaning that the cVAE preferentially uses the condition label and not the latent representation to model the pixel intensities for these channels.

Identifying a minimal set of antibodies for obtaining the same landmarks is an interesting research direction. In general, CAMPA was not built to directly output such a minimal set, but it would be possible to infer a minimal set after training CAMPA. In addition to using prior knowledge to select these antibodies (e.g. those against known molecular markers for subcellular organelles), this could be achieved by identifying proteins that are consistently enriched in a specific subset of CSLs, or have high saliency scores – indicating that channels do not depend on the condition input. It may also be advantageous to consider automatic feature selection methods¹⁷. Once a selection has been made, this could be rigorously validated in silico, by retraining CAMPA with a set of input channels omitted and comparing the learned CSLs.

2.11 Line 224: It seems that Figure 2a was not mentioned in the main text.

Thank you for catching that, we corrected this and added a mention in the first results section:

[line 217-218] (Results)

To facilitate biological interpretability, we annotated CSLs with the names of known subcellular structures (see methods) (Fig. 2a).

2.12 Line 417: Is there evidence showing cell size change is the actual cause of subcellular reorganization? The following paragraph shows the cell size change was induced by reducing the expression of SBF2, but the siRNA treatment may also cause other changes (other than cell size) and lead to subcellular reorganization, right? If not, please provide references.

In general, cell size control is poorly understood in mammalian cells. There is no evidence that we are aware of that the cell size changes induced by SBF2 knockdown are the actual cause of the subcellular reorganisation or whether SBF2 knockdown has other effects on subcellular organisation independent of cell size changes. We have edited the text to reflect this:

[line 457-460] (Results)

This indicates that the doubling of cell volume induced by *SBF2* knockdown is associated with disproportionate changes in size of different subcellular compartments, however we cannot exclude other effects of *SBF2* knockdown that are independent of cell size changes.

2.13 Line 424-427: To clarify, when applying CAMPA on the 43-channel dataset, is everything (training, clustering, annotation and merging) done from scratch? And also would it be possible to map and compare with the landmarks detected in the 34-channel dataset? Do the two dataset share channels?

Yes, CAMPA is re-trained on the 43-channel dataset described in Figures 5 and 6. For these analyses we were interested in studying cytoplasmic compartments, and therefore needed to include the additional channels localising to cytoplasm in the CAMPA workflow. As the neural network models are quite small this takes little time and effort and lets us study both nucleoplasmic and cytoplasmic subcellular structures.

As the reviewer correctly remarks, the 34 channels of the 34-channel dataset are a subset of the 43 channels in the 43-channel dataset. It is therefore possible to use the model trained on the 34-channel dataset on the 43-channel dataset to compare the two CSL annotations. The caveat of this comparison is that the model trained on the 34 channel data was trained on 184A1 cells, and the 43 channel data is on HeLa cells, so there might be cell-line specific differences. In addition, the model trained on 43 channels is able to use additional information from the channels not present in the 34 channel dataset and therefore might result in a more precise allocation of CSLs.

We have now done this cross-comparison of the two different networks and added panels d and e to Supplementary Figure 12 showing these results, and reference these results in the main text. Our comparison shows that Nucleolus, Cajal bodies and Nucleoplasm have a high overlap between the two models. Nuclear speckles, PML bodies and Nuclear periphery show lower overlap scores, potentially due to the previously mentioned cell-line specific differences and additional markers in the 43-channel dataset resulting in slightly different definitions of these CSLs.

[lines 448-452] (Results)

These [CSLs] comprise all major compartments marked by the antibodies in the panel, including all previously identified nuclear CSLs (for comparison see Supplementary Fig. 12d,e) as well as cytoplasmic landmarks such as perinuclear and peripheral ER and mitochondria (HSPD1/CALR), Golgi apparatus (GOLGA2) cell-cell contacts (CTNNB1), focal adhesions (PXN) and P-bodies (DDX6) (Fig. 5b-c).

Supplementary Figure 12: Details of cVAE clustering (HeLa cells, scrambled siRNA).

[...] d: Example HeLa cells colored by CSLs obtained from 43-plex HeLa cells (cf. Fig. 5) and CSLs obtained from 34-plex 184A1 cells (nucleus only, cf. Fig. 2).

e: Comparison of CSLs obtained from 43-plex HeLa cells and CSLs obtained from 34-plex 184A1 cells on HeLa cells. Shown is the intersection over union (IOU) of the common CSLs from both clusterings. Nucleolus, Cajal bodies, Nucleoplasm and Nuclear speckles have high IOU scores. Nuclear speckles, PML bodies, and Nuclear periphery have lower IOU scores, indicating cell-line specific differences or that information from the additional channels in the 43-plex HeLa dataset results in a slightly different definition of these CSLs.

2.14 Finally, it seems the cVAE should be trained specifically for each collection of immunofluorescence channels, it remains unclear how the antibody channels were selected. In large-scale cost-sensitive drug screening settings, it will be very useful to infer the minimal set of immunofluorescence labels to obtain the landmarks. Could you elaborate on how this could be implemented with CAMPA?

Correct, CAMPA is supposed to be trained for every dataset individually. The markers in this experiment were selected for several reasons. We now include the following description in the first results section:

[lines 157-160] (Results)

The proteins and post-translational modifications imaged (Supplementary Table 1) either play roles in RNA metabolism, or are molecular markers characteristic of certain subcellular organelles (e.g. nuclear speckles) or cellular states (e.g. cell cycle stage, cell crowding).

See our response to comment 2.10 for a short elaboration of how to infer a minimal set of antibodies to obtain the same landmarks.

Code review:

2.15 The manuscript comes with a Python library (campa) which is a great way for increasing the adoption of the method. It is also appreciated that the software package was released under a permissive license and comes with proper documentation to ease the adoption of the software. I spent some time running the package but didn't

manage to configure it properly thus cannot provide a thorough review on the code bases. As a general comment, it is clear that efforts have been made for other users to adopt the campa tool but it would be more helpful if the tutorials can be made more friendly for the first-time users.

Thank you for the helpful evaluation. Following this feedback, we have improved the CAMPAs configuration and documentation, revised the tutorials to be self-contained, and had a first-time user test the package and tutorials.

2.16 There is also a dedicated code repository (campa_ana) for reproducing the results in the manuscript which seems complete, but there seems no data provided to enable the actual reproduction of the results. In the manuscript it mentioned that the data will be available upon request, so maybe at least provide more info on how to obtain all the data for readers who are interested in reproducing the results. Meanwhile, if possible, the authors are encouraged to use the open data repositories such as Zenodo and IDR to share the full set of their image data publicly – it will be a valuable resource for the community to evaluate campa and improve it further.

Thank you for bringing up the important point of data sharing. Public sharing of image data is not yet as common as for genomic data, primarily due to large file sizes. Our preprocessed and compressed image data in the file format used by CAMPAs is 60GB large, and initially we were not able to find an open data repository that allowed sharing of such large files. Following this comment, we have contacted Zenodo who were able to increase the repository limit (usually 50GB) for our dataset. Therefore, we will now share the preprocessed data in the file format used by CAMPAs on Zenodo upon acceptance of our manuscript. To enable full reproducibility of our results we will now additionally share the pretrained models and the CSL clusterings based on these models in a second Zenodo repository upon acceptance of our manuscript. We have edited the data availability statement to reflect this:

[lines 616-621]

The data used to generate all results and figures reported in this manuscript will be available at <https://doi.org/10.5281/zenodo.7299516> upon acceptance¹⁸. Pre-trained models and clusterings reported in the manuscript will be available at <https://doi.org/10.5281/zenodo.7299750> upon acceptance¹⁹. All scripts necessary for reproducing the results and figures (except schematic figure panels Fig. 1a,b,c, Fig. 3f) can be found at https://github.com/theislab/campa_ana.

In addition, we added an explanatory notebook to the reproducibility repository (campa_ana), explaining how to setup CAMPAs and download the data in order to fully reproduce the results and figures presented in the manuscript: https://github.com/theislab/campa_ana/blob/develop/workflow/00_setup_and_download_data.ipynb

2.17 More specifically, the installation of the software through a pip command was easy to follow and the generated documentation about classes, API and CLI are helpful for the users to look up. The use of the campa package heavily depends on the correct configuration of the config file, and the most difficult part during my test is to construct a config file and dataset that works for campa despite the detailed instructions. Similarly, the tutorials were also written based on the assumption that users have already made the correct config file and have already organized data folders in a specific way.

Thank you. As suggested, we have improved the documentation of the config file. In addition to having a permanent config file, it is also possible to define a temporary config, without the need of saving a file to disk. We have revised the documentation to more clearly explain these two different ways to generate a config file, and made the tutorials self-contained by configuring the correct config file in the first tutorial (<https://campa.readthedocs.io/en/latest/notebooks/setup.html>). This change makes configuring and testing CAMPAs easier, because tutorials don't depend on the user correctly configuring the config file.

2.18 Here are suggestions for making it easier for the first-time user to adopt campa:

Thank you for these valuable suggestions on how to improve CAMPAs tutorials. We have implemented all of them in the revised version of the code as described below.

- Provide a command to download a complete but minimal example dataset folder with the correct configuration and data. In the documentation, show a complete example of the config file along with a complete zip file with all the data files that work with the config file. (There seems to be a command already)

The initial version of CAMPA already included an example dataset that could be downloaded and that all tutorials were based on. We improved the setup tutorial to ensure that all necessary files for running the following tutorials are downloaded.

- The tutorial should be self-contained without too much assumption. Ideally, it should start by telling the user how to prepare the environment for the tutorial (e.g.: create a new conda environment and install campa and other dependencies, clone the campa repo with notebooks, start Jupyter notebook, and navigate to the notebooks folder).

Thank you for this suggestion. We have edited the setup tutorial to include a short description of how to set up the environment before running this tutorial.

- All the tutorial notebook should be self-contained without assuming that the user has done the correct configuration. E.g. start by running a command to download the example dataset, then create the config automatically such that the user can just execute the cells one by one. Keep in mind that most users will quit easily if they need to make a lot of effort to get started with campa for the first time.

As described in our response to 2.17 we made the tutorials self-contained by improving the setup tutorial to set up and download all relevant data and by adding reminders at the top of the other tutorials to run the setup tutorial first.

- It is always helpful to ask someone outside the team to do a blind test of the documentation and tutorials.

As suggested, we have asked a first-time user to test the documentation and tutorials, and following their feedback were able to further clarify the descriptions in the tutorials.

Reviewer #3 (Remarks to the Author):

Spitzer and colleagues present a deep-learning method called CAMPA that aims to learn common states from multi-dimensional microscopy measurement across experimental conditions in order to explore the changes in cellular organization and molecular composition. The strategy utilizes a deep learning framework to learn condition independent molecular signatures that can then be compared across conditions. The stated goal is to move away from existing methods that utilize clustering across conditions and therefore may be more sensitive to misassignment if a perturbation leads to a major change in a specific feature. The authors use this approach to analyze a set of images for 44 protein markers across 6 chemical perturbations that disrupt Pol I transcription, Pol II initiation, Histone acetylation, mRNA splicing, and others. They use these 44-markers to define key nuclear structure states and describe several intriguing new observations about nuclear body dynamics upon distinct perturbations. For example, they found that nuclear speckles increase in size upon inhibition of mRNA splicing and that their molecular composition changes, corresponding to an increase in polyA binding protein localization and depletion of Pol II.

Thank you for this positive evaluation of CAMPA and sharing our enthusiasm for the biological findings that we can gain by applying CAMPA.

3.1 Overall, I find the paper, the dataset, and the results described to be quite interesting. Certainly, the dataset alone will be of great interest to many within the nuclear structure field and their observations upon these distinct perturbations add key new insights. Yet, I must admit that I did not fully grasp the importance of the CAMPA method or what specific need it was addressing. While the authors briefly motivate the method by highlighting that other methods learn parameters within samples and therefore perturbations could impact classification, it is not clear why a much simpler solution of simply leaving said variable out in the assignments would not address this very specific issue. Related to this, it is not clear to me that CAMPA was really needed to derive the biological insights highlighted in this paper. How would the previous approaches perform on these same datasets? Based on the vectors shown, I would imagine just fine, but the authors do not describe this at all.

Thank you for this comment. Direct identification of the same subcellular structures across conditions was not possible using the previous (direct) pixel clustering approaches because the intensity changes in some channels induced by perturbations often resulted in reassignment of specific organelles to different clusters in different perturbations. In response to this comment, and those of the other reviewers, we have simplified Fig. 1 and edited the description of the method to sharpen the focus on this point (see response to reviewer #1, comment F).

Figure 1: CAMPA enables unsupervised learning of consistent subcellular landmarks using a conditional variational autoencoder.

a: Schematic showing perturbation-induced changes in channel intensity.

b: Schematic of direct pixel clustering across experimental conditions leading to condition-dependent clusters.

c: Schematic of CAMPA, showing how a cVAE conditioned on perturbation can learn a perturbation-independent latent space. Clustering this latent space results in consistent subcellular landmarks (CSLs), enabling quantitative comparisons at the subcellular scale. [...]

We have also more fully expanded on how this data would be analysed using direct pixel clustering and how this compares to CAMPA (Supplementary Figure 7, Supplementary Note 1, see below). This now shows in detail how the direct pixel clustering approach performs on this dataset, when the goal is to identify and quantify subcellular changes that occur in different perturbations. In the Supplementary Note, we focus on TSA-treatment, which expands on one of the examples of perturbation-dependent clustering shown in Fig. 1h. The direct pixel clustering does not perform at all well because many of the identified clusters change in relative abundance, so it is difficult to interpret what effects, if any, the TSA treatment has on subcellular organisation.

One intuitive solution you propose is to leave the variables out of the clustering. Indeed, this a reasonable approach, which we have previously used and which we initially tried on this dataset. The problem is that this does not scale well to a large number of conditions because too many channels change in overall intensity in different conditions. Moreover, there are also markers that change in intensity across cell-cycle stages, so these would also have to be omitted, or cell-cycle stages considered separately. To describe this problem, we have added a paragraph to Supplementary Note 1.

We added a short summary of how the direct pixel clustering differs from CAMPA to the main text in the first and second results section:

[lines 237-240] (Results)

We therefore conclude that CAMPA allows consistent identification and annotation of subcellular landmarks across perturbations and cell cycle stages. This contrasts with previous direct pixel clustering approaches, which often identify different clusters for the same subcellular organelle (in different perturbations or cell cycle stages).

[lines 270-275] (Results)

Unlike the direct pixel clustering approaches used previously^{2,3}, in which conditions are compared by identifying pixel classes that change abundance between conditions (Supplementary Fig 7, Supplementary Note 1), the CSL-based approach used in CAMPA compares molecular abundances across landmarks that are consistently found in both conditions (CSLs). This naturally extends traditional analysis of cellular abundance changes (Fig. 1d) to the subcellular scale.

[Supplements, lines 23-85] (Supplementary Note 1)

Detailed comparison of CAMPA and direct pixel clustering in perturbed 184A1 nuclei

Analysis of multiplexed image datasets at the subcellular scale across conditions has previously been performed using direct pixel clustering². In this case, pixel profiles are clustered across the entire dataset of perturbed and non-perturbed cells to identify the predominant pixel 'types' (previously called multiplexed cellular units - MCUs), corresponding to a certain set of colocalising markers (Supplementary Fig 7a). Because perturbations result in changes to molecular pixel profiles, the abundance of the identified pixel clusters in each cell (or condition) contains information about both the overall and subcellular changes in protein abundance and colocalization. This vector of cluster abundances can be used as a cellular "phenotype" which can be used to cluster cells according to experimental perturbations². Quantitative comparisons of these direct pixel clusters can then be achieved by determining which clusters change in abundance in the different conditions or cell states (Supplementary Fig. 7b). The average molecular profiles which define the clusters are assumed to be independent of the condition, however the extent to which this is a reasonable approximation depends on the clustering resolution. In summary, direct pixel clustering aims to identify pixel combinations that are unique to different conditions, but whose molecular composition is consistent across conditions.

In contrast to this approach, CAMPA identifies CSLs that are found across conditions (and correspond to specific subcellular structures) but whose molecular composition differs between conditions. Using CAMPA, one can therefore make quantitative comparisons of subcellular molecular changes by comparing the molecular compositions of the same CSLs across conditions, rather than by comparing how the abundances of clusters change (in the direct clustering approach).

As an example to compare these two different approaches, we focus on 184A1 nuclei treated with the histone deacetylase inhibitor TSA. At a Leiden resolution of 1.2, direct pixel clustering across all data identifies 20 pixel clusters (Supplementary Fig. 7a). In TSA-treated cells, clusters 10 and 19 are strongly increased in abundance, and clusters 2, 3, 4, 5, 7, 8, 11, 14, 16 are reduced in abundance (Supplementary Fig. 7b). To extract biological insights from this data, one must examine all of these clusters to identify what they correspond to, and then determine why they change in abundance in the clustering. Focussing on cluster 10 (which increases in abundance upon TSA-treatment), it can be seen that this is strongly enriched for histone H3 lysine 27 acetylation (H3K27ac), however there are also numerous other channels enriched in cluster 10, such as POLR2A-S2P, NONO and CDK9. Plotting the clusters out across example cells, we can see that cluster 10 represents a specific subset of nucleoplasmic pixels that predominantly occur in TSA-treated cells (Supplementary Fig. 7c). This enrichment for a pixel cluster that is defined by high H3K27ac is expected because TSA treatment results in increased global histone acetylation. Using direct pixel clustering, however, it is difficult to obtain detailed insights into whether TSA has other effects on subcellular organisation because changes in H3K27ac result in reassignment of a large fraction of nuclear pixels to different clusters. For example, how should the reductions in abundance of clusters 2, 3, 4, 5, 7, 8, 11, 14, 16 be interpreted? Using CAMPA to analyse the same data, we instead learn a common set of seven nuclear CSLs across all conditions (Supplementary Fig. 7d), and can then examine how each 4i marker intensity changes within these identified CSLs (Supplementary Fig. 7e). Doing so reveals that there is very little change to the size or the molecular composition of any of the CSLs, with the exception of increased H3K27ac that occurs uniformly across all nuclear subcompartments (Supplementary Fig. 7e). CAMPA therefore greatly simplifies the analysis of this perturbation condition compared to the direct pixel clustering approach.

An alternative method to 'save' the direct pixel clustering approach in this case would be to omit H3K27ac from the clustering because it varies globally between conditions, and thereby may allow identification of consistent clusters across conditions in TSA-treated and unperturbed cells. However, this would then require any measured protein that varies in intensity between conditions to be omitted from the clustering, which would rapidly limit the scalability to a large number of conditions. For example, at least eight of the 43 channels vary significantly in at least one of the six perturbations studied here (Fig. 1d), and at least two more vary across different cell-cycle stages (Supplementary Fig. 4a). Moreover, if the omitted channels include structural markers, then this approach may fail to recognize certain important subcellular landmarks. By instead using a conditional model, CAMPA removes this requirement to choose input channels *a priori* and allows the model to learn which proteins change in intensity between conditions during model training.

Supplementary Fig. 7: Detailed comparison of CAMPA and direct pixel clustering in perturbed 184A1 nuclei.

a: Mean intensities of each channel across different clusters. Clusters obtained from direct Leiden clustering (resolution 1.2) of pixel profiles for all 34 nuclear channels across all six experimental conditions. Clusters are those shown in Fig. 1h. Values z-scored by channel.

b: Log₂ fold-change in relative cluster abundance in perturbation conditions, compared to unperturbed cells.

c: Example cells treated with TSA or unperturbed. Direct pixel clusters and H3K27ac levels shown.

d: CAMPA-derived CSLs for the cells shown in c.

e: Log₂ fold-change of mean intensities for each channel in each CSL, or number of pixels in each CSL, when comparing TSA-treated with unperturbed control cells. P-values show significance of TSA treatment on levels for each channel/CSL, as determined from a mixed effect model. P-values are corrected for multiple hypothesis testing using the Benjamini-Yuketeli method.

3.2 I do not mean to be a stickler here, since the paper is quite interesting (to me) even without the method. However, given that the paper was submitted to Nature Methods, I suspect the method is far more important than is obvious to me from the description. If this is indeed the case, the authors really should rewrite a bit to more clearly articulate the nature of the issue with existing methods and their limitations, what CAMPAs enables and why this is specifically required for addressing the goals in the experimental datasets they presented. I would also encourage them to consider comparing the results obtained from CAMPAs to alternative approaches on their dataset in order to highlight the nature of information you can derive using this approach. I think that the authors will find that this effort will make their method more useful and widely adopted.

Thank you for this suggestion on how to improve the manuscript, this was indeed not ideal and your suggestion is in line with reviewer #1. We have now sharpened the focus of the manuscript on the method and its capabilities by re-focussing on CAMPAs unique contributions in the introduction (see response to reviewer #1, comment B). We have also extended discussions of how CAMPAs differs from previous work, and performed ablation studies To discuss how CAMPAs differs from previous work, we have added a Supplementary Note and Figure to perform a more detailed comparison to previous methods, using the data acquired for this work (see our response to comment 3.1), and we have added a discussion of related deep-learning based approaches developed for similar data or purposes (see response to reviewer #1, comment G). We added ablation studies to further explore which parts of the model are important. For this, we have evaluated the influence of the conditioning and the input patch size on the learned latent space, and discuss the results in the first results section and show them in Fig. 1 (see response to reviewer #1, comment B).

Finally, we revised the entire manuscript to make the descriptions of the biological results more concise and put a larger focus on CAMPAs contributions (see response to reviewer #1, comment F).

List of references

1. Schapiro, D. *et al.* MCMICRO: a scalable, modular image-processing pipeline for multiplexed tissue imaging. *Nat. Methods* **19**, 311–315 (2022).
2. Gut, G., Herrmann, M. D. & Pelkmans, L. Multiplexed protein maps link subcellular organization to cellular states. *Science* **361**, (2018).
3. Takei, Y. *et al.* Integrated spatial genomics reveals global architecture of single nuclei. *Nature* **590**, 344–350 (2021).
4. Sundararajan, M., Taly, A. & Yan, Q. Axiomatic Attribution for Deep Networks. in *Proceedings of the 34th International Conference on Machine Learning* (eds. Precup, D. & Teh, Y. W.) vol. 70 3319–3328 (PMLR, 06–11 Aug 2017).
5. Greenwald, N. F. *et al.* Whole-cell segmentation of tissue images with human-level performance using large-scale data annotation and deep learning. *Nat. Biotechnol.* **40**, 555–565 (2022).
6. Stringer, C., Wang, T., Michaelos, M. & Pachitariu, M. Cellpose: a generalist algorithm for cellular segmentation. *Nat. Methods* **18**, 100–106 (2021).

7. Kobayashi, H., Cheveralls, K. C., Leonetti, M. D. & Royer, L. A. Self-supervised deep learning encodes high-resolution features of protein subcellular localization. *Nat. Methods* **19**, 995–1003 (2022).
8. Lu, A. X., Kraus, O. Z., Cooper, S. & Moses, A. M. Learning unsupervised feature representations for single cell microscopy images with paired cell inpainting. *PLoS Comput. Biol.* **15**, e1007348 (2019).
9. Clarke, Z. A. *et al.* Tutorial: guidelines for annotating single-cell transcriptomic maps using automated and manual methods. *Nat. Protoc.* **16**, 2749–2764 (2021).
10. The Human Protein Atlas. <https://www.proteinatlas.org/>.
11. Thul, P. J. *et al.* A subcellular map of the human proteome. *Science* **356**, (2017).
12. Berg, S. *et al.* ilastik: interactive machine learning for (bio)image analysis. *Nat. Methods* **16**, 1226–1232 (2019).
13. Lopez, R., Regier, J., Cole, M. B., Jordan, M. I. & Yosef, N. Deep generative modeling for single-cell transcriptomics. *Nat. Methods* **15**, 1053–1058 (2018).
14. Sohn, K., Lee, H. & Yan, X. Learning Structured Output Representation using Deep Conditional Generative Models. in *Advances in Neural Information Processing Systems* (eds. Cortes, C., Lawrence, N., Lee, D., Sugiyama, M. & Garnett, R.) vol. 28 (Curran Associates, Inc., 2015).
15. Kingma, D. P. & Welling, M. Auto-Encoding Variational Bayes. *arXiv [stat.ML]* (2013).
16. Rybkin, O., Daniilidis, K. & Levine, S. Simple and Effective VAE Training with Calibrated Decoders. in *Proceedings of the 38th International Conference on Machine Learning* (eds. Meila, M. & Zhang, T.) vol. 139 9179–9189 (PMLR, 2021).
17. Tang, J., Alelyani, S. & Liu, H. Feature selection for classification: A review. *Data classification: Algorithms and applications* **37** (2014).
18. Spitzer, H., Berry, S., Pelkmans, L. & Theis, F. J. 4i dataset for ‘Learning consistent subcellular landmarks to quantify changes in multiplexed protein maps’. doi:10.5281/zenodo.7299516.
19. Spitzer, H., Berry, S., Pelkmans, L. & Theis, F. J. Analysis results reported in ‘Learning

consistent subcellular landmarks to quantify changes in multiplexed protein maps'.

doi:10.5281/zenodo.7299750 .

Decision Letter, first revision:

Dear Fabian,

Thank you for submitting your revised manuscript "Learning consistent subcellular landmarks to quantify changes in multiplexed protein maps" (NMEMH-A49213B). It has now been seen by the original referees and their comments are below. The reviewers find that the paper has improved in revision, and therefore we'll be happy in principle to publish it in Nature Methods, pending minor revisions to satisfy the referees' final requests and to comply with our editorial and formatting guidelines.

We ask that you please address the remaining concerns of referee 2 and clarify in your revision how the choice of annotations impacts the biological implications of findings generated with CAMPA.

TRANSPARENT PEER REVIEW

Nature Methods offers a transparent peer review option for new original research manuscripts submitted from 17th February 2021. We encourage increased transparency in peer review by publishing the reviewer comments, author rebuttal letters and editorial decision letters if the authors agree. Such peer review material is made available as a supplementary peer review file. Please state in the cover letter 'I wish to participate in transparent peer review' if you want to opt in, or 'I do not wish to participate in transparent peer review' if you don't. Failure to state your preference will result in delays in accepting your manuscript for publication.

ORCID

IMPORTANT: Non-corresponding authors do not have to link their ORCIDs but are encouraged to do so. Please note that it will not be possible to add/modify ORCIDs at proof. Thus, please let your co-authors

know that if they wish to have their ORCID added to the paper they must follow the procedure described in the following link prior to acceptance:

Sincerely,
Rita

Rita Strack, Ph.D.
Senior Editor
Nature Methods

Reviewer #1 (Remarks to the Author):

The Authors have put a convincing work in improving the manuscript and improve on the points that the reviewers raised. The text and figures are improved in clarity. Some of the missing background has been added. In particular the abstract is definitely better.

At this point I have no objections to the publication of this very nice paper.

Reviewer #2 (Remarks to the Author):

In the revised version, Spitzer et. al improved the manuscript by adding clarifying the purpose and procedure of the annotation step. The newly added automatic annotation further increases the usability of the tool. For the software package, the tutorial and documentation are greatly improved. The effort for sharing the full dataset and trained models with the community is appreciated. Overall, I think the improved manuscript is ready to be shared with a broader audience.

However, regarding the following statement about the annotation step:

“First, it is important to note that the annotation and CSL merging step is entirely optional for the analysis we present in the manuscript. None of the specific biological insights we highlight in the manuscript are affected by the annotation and merging of CSLs, but interpretability of the results is clearly aided by us giving names to clusters.

”

I think the annotation step is crucial for assigning meaning for each cluster, otherwise we won't be able to say much about the underlying biology and wrong or inaccurate annotation will lead to wrong biological insights.

For example, in the following statement (line 294-295): "our analysis shows that the relative abundance of CDK9 (the kinase predominantly responsible for POLR2A-S2P) actually increases within nuclear speckles at the same time". Considering two cases: 1) if we skip the annotation step, we can only say that CDK9 increases within "cluster 3", which won't mean much for the user; and 2) if we label the cluster "nuclear speckles" wrongly as "nucleolus", it will be an incorrect conclusion.

In my opinion, the interpretability aided via the annotation step is a crucial step to make CAMPA actually useful. Generating biological insights largely depends on the correct annotation of CSLs.

Wei Ouyang

Reviewer #3 (Remarks to the Author):

My concerns with the initial submission primarily related to the description of the CAMPA method and specifically its importance relative to existing and naive approaches. The authors did a thorough job describing these aspects and directly establishing the importance of this approach relative to the previous alternatives. Overall, I am happy with this revision and would be delighted to see this published in Nature Methods.

Author Rebuttal, first revision:

Point-by-point response to the reviewers' comments

Learning consistent subcellular landmarks to quantify changes in multiplexed protein maps

Hannah Spitzer^{1,*}, Scott Berry^{2,3,*}, Mark Donoghoe⁴, Lucas Pelkmans², Fabian J. Theis^{1,5,6}

¹ Institute of Computational Biology, Helmholtz Center Munich, Germany.

² Department of Molecular Life Sciences, University of Zurich, Zurich, Switzerland.

³ EMBL Australia Node in Single Molecule Science, School of Medical Sciences, University of New South Wales, Australia.

⁴ Stats Central, Mark Wainwright Analytical Centre, University of New South Wales, Australia.

⁵ Department of Mathematics, Technical University of Munich, Germany.

⁶ TUM School of Life Sciences Weihenstephan, Technical University of Munich, Germany.

In the following, we present our response to the reviewers comments. We state **reviewers' comments (black)**, **point-by-point answers (green)** to the questions and in parts **copy parts of the text or specific panels (blue)**, which directly correspond to comments or reference to them.

Editor comments:

Thank you for submitting your revised manuscript "Learning consistent subcellular landmarks to quantify changes in multiplexed protein maps" (NMETH-A49213B). It has now been seen by the original referees and their comments are below. The reviewers find that the paper has improved in revision, and therefore we'll be happy in principle to publish it in Nature Methods, pending minor revisions to satisfy the referees' final requests and to comply with our editorial and formatting guidelines.

We ask that you please address the remaining concerns of referee 2 and clarify in your revision how the choice of annotations impacts the biological implications of findings generated with CAMPA.

Thank you for this positive assessment. Reviewer #2 rightly pointed out that biological interpretation of results generated with CAMPA is impacted by annotations, which we address in our revision by clarifying the section introducing the annotation, putting a larger focus on our automated annotation proposals that guard against mis-annotations.

Reviewer #1 (Remarks to the Author):

The Authors have put a convincing work in improving the manuscript and improve on the points that the reviewers raised. The text and figures are improved in clarity. Some of the missing background has been added. In particular the abstract is definitely better. At this point I have no objections to the publication of this very nice paper.

Thank you.

Reviewer #2 (Remarks to the Author):

In the revised version, Spitzer et. al improved the manuscript by adding clarifying the purpose and procedure of the annotation step. The newly added automatic annotation further increases the usability of the tool. For the software package, the tutorial and documentation are greatly improved. The effort for sharing the full dataset and trained models with the community is appreciated. Overall, I think the improved manuscript is ready to be shared with a broader audience.

However, regarding the following statement about the annotation step:

"First, it is important to note that the annotation and CSL merging step is entirely optional for the analysis we present in the manuscript. None of the specific biological insights we highlight in the manuscript are affected by the annotation and merging of CSLs, but interpretability of the results is clearly aided by us giving names to clusters."

I think the annotation step is crucial for assigning meaning for each cluster, otherwise we won't be able to say much about the underlying biology and wrong or inaccurate annotation will lead to wrong biological insights.

For example, in the following statement (line 294-295): "our analysis shows that the relative abundance of CDK9 (the kinase predominantly responsible for POLR2A-S2P) actually increases within nuclear speckles at the same time". Considering two cases: 1) if we skip the annotation step, we can only say that CDK9 increases within "cluster 3", which won't mean much for the user; and 2) if we label the cluster "nuclear speckles" wrongly as "nucleolus", it will be an incorrect conclusion.

In my opinion, the interpretability aided via the annotation step is a crucial step to make CAMPA actually useful. Generating biological insights largely depends on the correct annotation of CSLs.

Wei Ouyang

Thank you for the positive assessment. We agree with you that in order to biologically interpret the results, correct annotation of CSLs is absolutely necessary and that mis-annotation of CSLs might lead to incorrect conclusions.

The point that were trying to make in the rebuttal is that our proposed workflow for generating consistent subcellular landmarks (training of a conditional variational autoencoder and subsequent clustering of the learned latent representation) does not depend on specific subjective user choices

like manual merging or annotation of clusters. Instead, even without annotation, the resulting CSLs will be consistent across different conditions and represent common landmarks in cells and quantitative comparisons are possible.

For example, without annotation, we can still compare multiple perturbations with respect to intensity or spatial changes as presented in Figure 4. However, to assess how perturbations are similar or different to each other, or to interpret the results of the quantitative comparisons presented throughout the manuscript, annotating CSLs with biologically meaningful names is necessary. This means that the *results* of the method itself do not depend on the annotation of CSLs, but the *interpretation* of the results (arguably as important as the results themselves) does. This might not have been entirely clear in our previous revision.

We have revised the statement you cite above to make the purpose of the annotation more clear and to put a stronger focus on the automated annotations:

[Results]

To enable biological interpretability of quantitative comparisons between cells, we annotated CSLs with the names of known subcellular structures (see methods) (Fig. 2a). To facilitate this optional step in the CAMPA workflow, and to avoid mis-annotations, automated annotation proposals can be obtained by querying the Human Protein Atlas (HPA)^{23,24} database. The annotation resulted in assignment of the ten original CSLs to seven annotated CSLs (Nucleolus, Nuclear speckles, PML bodies, Cajal bodies, Nucleoplasm, Nuclear periphery, and Extra-nuclear (outside the nucleus)) (Fig. 2d-i), by merging four original CSLs to the Nucleoplasm CSL (Extended Data Fig. 4a). These annotations are consistent with automatic annotations proposed by HPA (Extended Data Fig. 4b).

Reviewer #3 (Remarks to the Author):

My concerns with the initial submission primarily related to the description of the CAMPA method and specifically its importance relative to existing and naive approaches. The authors did a thorough job describing these aspects and directly establishing the importance of this approach relative to the previous alternatives. Overall, I am happy with this revision and would be delighted to see this published in Nature Methods.

Thank you.

Final Decision Letter:

Dear Fabian,

I am pleased to inform you that your Article, "Learning consistent subcellular landmarks to quantify changes in multiplexed protein maps", has now been accepted for publication in Nature Methods. Your paper is tentatively scheduled for publication in our July print issue, and will be published online prior to that. The received and accepted dates will be May 12, 2022 and April 25, 2023. This note is intended to let you know what to expect from us over the next month or so, and to let you know where to address any further questions.

Once your paper is typeset, you will receive an email with a link to choose the appropriate publishing options for your paper and our Author Services team will be in touch regarding any additional information that may be required.

Please note that *Nature Methods* is a Transformative Journal (TJ). Authors may publish their research with us through the traditional subscription access route or make their paper immediately open access through payment of an article-processing charge (APC). Authors will not be required to make a final decision about access to their article until it has been accepted. [Find out more about Transformative Journals](https://www.springernature.com/gp/open-research/transformative-journals)

Your paper will now be copyedited to ensure that it conforms to Nature Methods style. Once proofs are generated, they will be sent to you electronically and you will be asked to send a corrected version within 24 hours. It is extremely important that you let us know now whether you will be difficult to contact over the next month. If this is the case, we ask that you send us the contact information (email, phone and fax) of someone who will be able to check the proofs and deal with any last-minute problems.

If, when you receive your proof, you cannot meet the deadline, please inform us at rjsproduction@springernature.com immediately.

Once your manuscript is typeset and you have completed the appropriate grant of rights, you will receive a link to your electronic proof via email with a request to make any corrections within 48 hours. If, when you receive your proof, you cannot meet this deadline, please inform us at rjsproduction@springernature.com immediately.

Once your paper has been scheduled for online publication, the Nature press office will be in touch to confirm the details.

Once your paper has been scheduled for online publication, the Nature press office will be in touch to confirm the details.

Content is published online weekly on Mondays and Thursdays, and the embargo is set at 16:00 London time (GMT)/11:00 am US Eastern time (EST) on the day of publication. If you need to know the exact publication date or when the news embargo will be lifted, please contact our press office after you have submitted your proof corrections. Now is the time to inform your Public Relations or Press Office about your paper, as they might be interested in promoting its publication. This will allow them time to prepare an accurate and satisfactory press release. Include your manuscript tracking number NMETH-A49213C and the name of the journal, which they will need when they contact our office.

About one week before your paper is published online, we shall be distributing a press release to news organizations worldwide, which may include details of your work. We are happy for your institution or

funding agency to prepare its own press release, but it must mention the embargo date and Nature Methods. Our Press Office will contact you closer to the time of publication, but if you or your Press Office have any inquiries in the meantime, please contact press@nature.com.

Nature Portfolio journals [encourage authors to share their step-by-step experimental protocols](https://www.nature.com/nature-research/editorial-policies/reporting-standards#protocols) on a protocol sharing platform of their choice. Nature Portfolio 's Protocol Exchange is a free-to-use and open resource for protocols; protocols deposited in Protocol Exchange are citable and can be linked from the published article. More details can found at www.nature.com/protocolexchange/about.

Please note that you and any of your coauthors will be able to order reprints and single copies of the issue containing your article through Nature Portfolio 's reprint website, which is located at <http://www.nature.com/reprints/author-reprints.html>. If there are any questions about reprints please send an email to author-reprints@nature.com and someone will assist you.

Best regards,
Rita

Rita Strack, Ph.D.
Senior Editor
Nature Methods